# Technical note: Using Long Short-term Memory Models to Fill Data Gaps in Hydrological Monitoring Networks

Huiying Ren[1], Erol Cromwell[2], Ben Kravitz[3,4], and Xingyuan Chen[4]

[1]Earth Systems Science Division, Pacific Northwest National Laboratory, WA, USA
[2]Advanced Computing, Mathematics, and Data Division, Pacific Northwest National Laboratory, WA, USA
[3]Department of Earth and Atmospheric Sciences, Indiana University, Bloomington, IN, USA
[4]Atmospheric Sciences and Global Change Division, Pacific Northwest National Laboratory, WA, USA

**Correspondence:** Xingyuan Chen (Xingyuan.Chen@pnnl.gov)

**Abstract.** Quantifying the spatio-temporal dynamics in subsurface hydrological flows over a long time window usually employs a network of monitoring wells. However, such observations often are spatially sparse with potential temporal gaps from poor quality or instrument failure. In this study, we explore the ability of recurrent neural networks to fill gaps in a spatially distributed time-series dataset. We use a well network that monitors the dynamic and heterogeneous hydrologic exchanges between the Columbia River and its adjacent groundwater aquifer at the U.S. Department of Energy's Hanford site. This 10-year-long dataset contains hourly temperature, specific conductance, and groundwater table elevation measurements from 42 wells with various lengths of gaps. We employ a long short-term memory (LSTM) model to capture the temporal variations in the observed system behaviors needed for gap filling. The performance of the LSTM-based gap filling method was evaluated against a traditional autoregressive integrated moving average (ARIMA) method in terms of error statistics and accuracy in capturing the temporal patterns of river corridor wells with various dynamics signatures. Our study demonstrates that the ARIMA models yield better average error statistics, yet tend to have larger errors during time windows with abrupt changes or high-frequency (daily and subdaily) variations. The LSTM-based models excel in capturing both high-frequency and low-frequency (monthly and seasonal) dynamics. But, the inclusion of high-frequency fluctuations may also lead to overly dynamic predictions in time windows that lack such fluctuations. The LSTM can take advantage of the spatial information from neighboring wells to improve the gap filling accuracy, especially for long gaps in system states that vary at subdaily scales. While LSTM models require substantial training data and have limited extrapolation power beyond the conditions represented in the training data, they afford great flexibility to account for the spatial correlations, temporal correlations, and nonlinearity in data without a-priori assumptions. Thus, LSTMs provide effective alternatives to fill in data gaps in spatially distributed time-series observations characterized by multiple dominant frequencies of variability, which are essential for advancing our understanding of dynamic complex systems.

## 1 Introduction

Long-term hydrological monitoring using distributed well networks is of critical importance for understanding how ecosystems respond to chronic or extreme perturbations, as well as for informing policies and decisions related to natural resources and

environmental issues (Wett et al., 2002; Taylor and Alley, 2002; Grant and Dietrich, 2017). One of the most common methods for collecting hydrologic and chemistry data in groundwater is through wells (Güler and Thyne, 2004; Strobl and Robillard, 2008; Lin et al., 2012). However, most well data have temporal gaps due to instrument failure or poor measurement quality. These data gaps degrade the quality of the dataset and increase the uncertainty in the spatial and temporal patterns derived from the data. Gap filling is essential for developing an understanding of dynamic system behaviours and for use in creating continuous, internally consistent boundary conditions for numerical models. One outstanding challenge in gap filling is capturing nonstationarity in data.

Various statistical methods have been developed to fill gaps in spatio-temporal datasets, with the autoregressive integrated moving average (ARIMA) method being the most commonly used (Han et al., 2010; Zhang, 2003). For any given spatial location, ARIMA uses temporal autocorrelations to predict unobserved data points in a time series. Spatio-temporal autocorrelations can be considered by using multivariate ARIMA and space-time autoregressive models (Kamarianakis and Prastacos, 2003; Wikle et al., 1998; Kamarianakis and Prastacos, 2005). However, ARIMA cannot capture nonlinear trends because it assumes a linear dependence between adjacent observations (Faruk, 2010; Valenzuela et al., 2008; Ho et al., 2002). In addition, all existing space-time ARIMA models assume fixed global autoregressive and moving average terms, which fail to capture evolving dynamics in highly dynamic systems (Pfeifer and Deutrch, 1980; Griffith, 2010; Cheng et al., 2012, 2014). Spectral-based methods, such as singular spectrum analysis, the maximum entropy method, and Lomb-Scargle periodogram, have been used to account for nonlinear trends while filling in gaps in spatio-temporal datasets (Ghil et al., 2002; Hocke and Kämpfer, 2008; Kondrashov and Ghil, 2006). These methods use a few optimal spatial or temporal modes occurring at low frequencies to predict the missing values, with the other higher frequency components discarded as noise. This can lead to a reduced accuracy of the statistical models when fitting observations and predicting missing values (Kondrashov et al., 2010; Wang et al., 2012). Kriging and maximum likelihood estimation used in spatial and spatio-temporal gap filling often face challenges computing the covariance matrix of the data vector, as it can be quite large (Katzfuss and Cressie, 2012; Eidsvik et al., 2014). Other nonlinear methods have been explored with some success, including expectation-maximization or Bayesian probabilistic inference including hierarchical models, Gaussian process, and Markov chain Monte Carlo. Using models that build dependencies in different stages or hierarchies most effectively captures spatial and temporal correlations (Calculli et al., 2015; Banerjee et al., 2014; Datta et al., 2016; Finley et al., 2013; Stroud et al., 2017). In general, the expectation-maximization algorithm and Bayesian-based methods are sensitive to the choice of initial values and prior distributions in parameter space (Katzfuss and Cressie, 2011, 2012). Moreover, prior distributions and their associated parameters all need to be specified in both the spatial and temporal domains, which becomes increasingly difficult in more complex systems. Empirical Orthogonal Functions (EOF) related interpolation methods, such as least squares EOF (LSEOF), data interpolation EOF (DINEOF), and recursively subtracted EOF (REEOF), are widely used to fill in missing data such as clouds in sea surface temperature datasets or other satellite-based images with regular gridded domains (Beckers and Rixen, 2003; Beckers et al., 2006; Alvera-Azcárate et al., 2016). However, EOF methods the require gridded data which limits their use in filling data gaps in irregularly spaced monitoring networks.

Deep neural networks (DNNs) (Schmidhuber, 2015) are data-driven tools that, in principle, could provide a powerful way of extracting the nonlinear spatio-temporal patterns hidden in distributed time-series data without knowing their explicit forms (Längkvist et al., 2014). They are increasingly being used in geoscience domains to extract patterns and insights from streams of geospatial data and to transform our understanding of complex systems (Reichstein et al., 2019; Shen, 2018; Sun, 2018; Sun et al., 2019; Gentine et al., 2018). The umbrella term of DNN contains numerous architecture categories which can be selected for the problem at hand. Recurrent neural networks (RNNs) are a natural choice of architecture for the analyses in this paper, which focus on filling gaps in time-series data (JORDAN, 1986). RNNs take sequences (e.g., time series) as input and output either the single values or sequences that follow. They are designed to use information about previous events to make predictions about future events, by essentially letting the model "remember." However, RNNs have been shown to lose memory from previously trained data for longer sequences of data, i.e., they "forget" (Hochreiter et al., 2001). The earlier information becomes exponentially less impactful for prediction as sequence size increases. Long short-term memory (LSTM) networks are variations of RNNs explicitly designed to avoid this problem by using memory cells to retain information about relevant past events (Hochreiter and Schmidhuber, 1997). RNNs and LSTMs have been successfully applied to text prediction (Graves, 2013), text translation (Wu et al., 2016), speech recognition (Graves et al., 2013), and image captioning (You et al., 2016). Recently, hydrology applications of RNNs and LSTMs have emerged. For example, Kratzert et al. (2018) used LSTMs to predict watershed runoff from meteorological observations, Zhang et al. (2018) used LSTMs for predicting sewer overflow events from rainfall intensity and sewer water level measurements, and Fang et al. (2017) used LSTMs to predict soil moisture with high fidelity.

Our study aims to evaluate the potential of using LSTM models for filling gaps in spatio-temporal time series data collected from a distributed network. We tested the LSTM-based gap filling method using datasets collected to understand the interactions between a regulated river and a contaminated groundwater aquifer. We treat the gap filling as a forecasting problem, i.e., we use the historical data as input to predict the missing values in the data gaps. The future information relative to the gap is implicitly used in training the LSTM models. Framing gap filling as a predictive problem is a common practice when machine learning methods are used for filling gaps in time series data (see examples in Kandasamy et al. (2013), Körner et al. (2018), Chen et al. (2020), Zhao et al. (2020), Sarafanov et al. (2020), Contractor and Roughan (2021)). The performance of the LSTM-based gap filling method is compared with traditional time series approaches (i.e., ARIMA) to identify situations in which LSTM models out- or under-perform the ARIMA models.

## 2   Study Site and Data Description

A 10-year (2008–2018) spatio-temporal dataset was collected from a network of groundwater wells that monitor temperature (Water conductivity and temperature probe CS547A by Campbell Scientific), specific conductance (SpC) (Water conductivity and temperature probe CS547A by Campbell Scientific), and water-table elevation (stainless-steel pressure transducer CS451 by the Campbell Scientific) at the 300 Area of the U.S. Department of Energy Hanford site, located in southeastern Washington State. The groundwater well network was originally built to monitor the attenuation of legacy contaminants. The groundwater

aquifer at the study site is composed of two distinct geologic formations: a highly permeable formation (Hanford formation, consisting of coarse gravelly sand and sandy gravel) underlain by a much less permeable formation (the Ringold Formation, consisting of silt and fine sand). The dominant hydrogeologic features of the aquifer are defined by the interface between the Hanford and Ringold formations and the heterogeneity within the Hanford formation (Chen et al., 2012, 2013).

The intrusion of river water into the adjacent groundwater aquifer causes two water bodies with distinct geochemistry to mix and stimulates biogeochemical reactions at the interface. The river water has lower SpC (0.1–0.12 $mS/cm$) than the groundwater (averaging $\sim 0.4\, mS/cm$). Groundwater has a nearly constant temperature (16–17°C) as opposed to the seasonal variations of river temperature (3–22°C). The highly heterogeneous coarse-textured aquifer (Zachara et al., 2013) interacts with dynamic river stages to create complex river intrusion and retreat pathways and dynamics. The time series of multi-year

SpC and temperature observations at the selected wells show these complicated processes of river water intrusion into our study site (Figure 1). Wells near the river shoreline (e.g., wells 1-1, 1-10A, 2-2, and 2-3) tend to be strongly affected by river water intrusion in spring and summer. As such, the dynamic patterns of SpC and temperature correspond well to river stage fluctuations, specifically that SpC decreases and temperature increases with increasing river stage. Fluctuations of SpC in well 2-2 appear to be stronger and at higher frequency than in other wells, likely indicating its higher connectivity with the river.

For further inland wells (e.g., well 1-15) temperatures consistently remain within the groundwater temperature range and SpC has three noticeable dips (dropping from 0.5 to 0.4 $mS/cm$ range). The SpC dips coincide with the high river stages in years 2011, 2012, and 2017, which featured higher peak river stages than other years that enabled river water to intrude further into the groundwater aquifer. In wells located an intermediate distance from the river, such as well 2-5, the intrusion of river water is evident in most years. It is absent in low-flow years, such as 2009 and 2015, during which both SpC and temperature remain

nearly unchanged.

Earlier studies demonstrated that physical heterogeneity contributes to the different response behaviors of different locations while the river stage dynamics lead to multi-frequency dynamics in those responses. Natural climatic forcing drives seasonal and annual variations (Amaranto et al., 2019, 2018), whereas the operations of upstream hydroelectric dams to meet human societal needs primarily induce the higher-frequency (i.e., daily and sub-daily) fluctuations (Song et al., 2018). Our

system is representative of many dam-regulated gravel-bed rivers across the world, where anthropogenic dam operations have significantly altered the hydrologic exchanges between river water and groundwater, as well as the associated thermal and biogeochemical processes (Song et al., 2018; Shuai et al., 2019; Zachara et al., 2020). Note that the multi-frequency variations in data characterize the dynamic features of data, which could exist in both short-term and long-term time series data as a result of short-term or long-term monitoring efforts.

To understand the multi-frequency variations of the river water and groundwater mixing in each well at the study site, we perform spectral analysis on multi-year SpC observations at each selected well using a discrete wavelet transform (DWT). The DWT is widely used for time–frequency analysis of time series and relies on a "mother wavelet", which is chosen to be the Morlet wavelet (Grossmann and Morlet, 1984) to deal with the time-varying frequency and amplitude in this site's time-series data (Stockwell et al., 1996; Grinsted et al., 2004). We illustrate the Wavelet Power Spectrum (WPS) in log scale and

its normalized global power spectrum (average WPS over the time domain) for the multi-year SpC time series in the first two

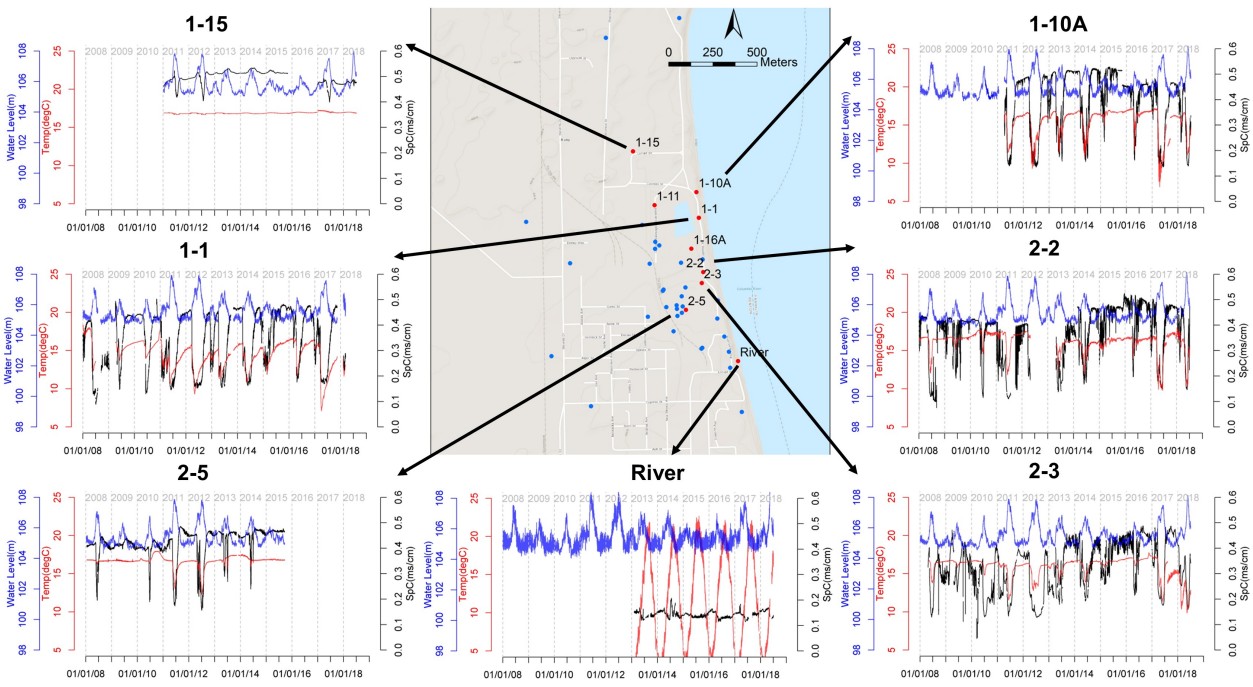

**Figure 1.** Groundwater monitoring well network in the 300 Area of the Hanford site and monitoring data at select wells. Each dot represents a well instrumented to measure groundwater elevation, temperature, and SpC. The wells selected for this study are marked with red dots with well names. The three monitored variables monitored are shown in time-series plots with blue (water elevation), black (SpC), and red (temperature) lines. Base map @Google Maps.

columns of Figure 2. Data gaps are shown as blank regions in Figure 2; examples include early year 2009 at well 1-1, the beginning of year 2011 at well 1-10A, and the later part of year 2012 at well 2-2. The WPS amplitude represents the relative importance of variation at a given frequency compared to the variations at other frequencies across the spectrum. At wells 1-1, 1-10A, 2-3, 2-5, and 2-2 the strongest intensities of their SpC signals appear at the half-year and yearly frequencies; however, well 1-15 has a different pattern with most of its high intensities below the 256-hour frequency. The averaged WPS further shows the behavior contrast: wells 1-1, 1-10A, 2-3, and 2-5 have a dominant frequency at half a year; well 2-2 has multiple dominant frequencies at daily, monthly, and seasonal scales; while well 1-15 has similar intensities at the half-year and hourly scales. Using this information to inform our approach, we hypothesize that gap filling at well 2-2 could be more challenging due to such mixture of dynamics signatures.

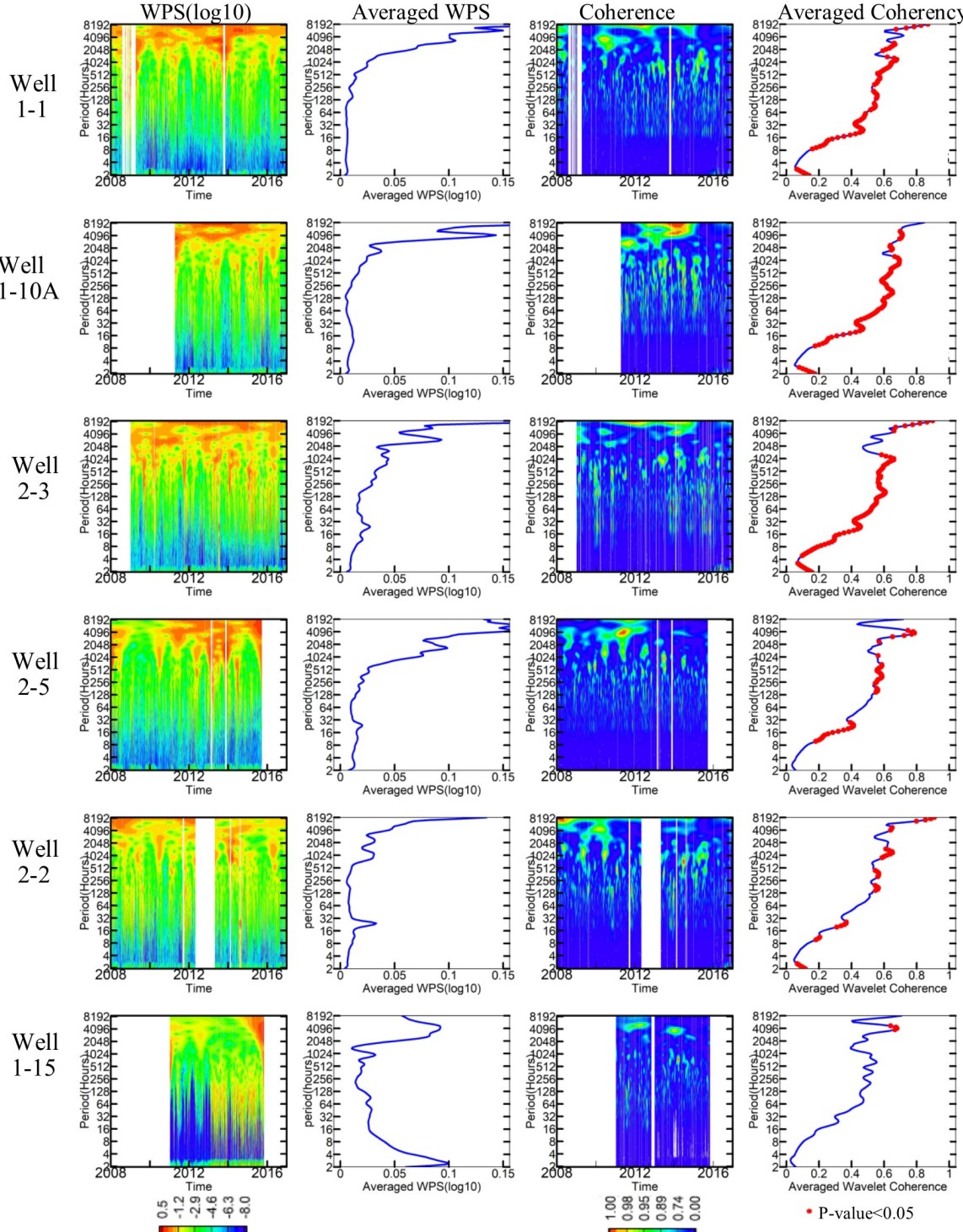

**Figure 2.** WPS analysis of the SpC at each well from 2008 to 2018. The first column is the spectrogram (in log10 scale) of SpC in each well; the second column is the averaged WPS; the third column is the coherence between SpC in each well and the river stage; and the fourth column is the averaged coherence with $p < 0.05$ values indicated in red.

Since the system dynamics are driven by the river stage, we perform magnitude-squared wavelet coherence analysis via the Morlet wavelet to reveal the dynamic correlations between the SpC and river stage time series (Grinsted et al., 2004; Vacha and Barunik, 2012). Wavelet coherence in the time-frequency domain is plotted in the third column in Figure 2 and the average coherence is plotted in the fourth column; red points indicate statistically significant values at the 95th percent confidence interval. A larger coherence at a given frequency indicates a stronger correlation at that frequency between the SpC at a well and the river stage. We consider these two variables highly correlated when the coherence is larger than 0.7 (shown with colors ranging from green to red colors in the Coherence plots). We found that such high correlations exist at multiple frequencies, from subdaily to daily to yearly, at all wells close to the river (e.g., 1-1, 1-10A, 2-2, and 2-3). Wells farther from the river (e.g., 1-15 and 2-5) have higher correlation regimes shifted towards longer periods at semi-annual and annual frequencies and are less persistent in time.

As can be seen in Figure 2, many of the wells have long data gaps which have unknown effects on our ability to estimate dynamics from the wavelet spectra. As such, gap filling is needed to infer observations and guide our modeling of the underlying system. Figure 3 provides a summary of gap lengths for the overall network of monitoring wells. The majority of the gap lengths of the three monitored variables are less than 50 hours. Therefore, our investigations explore the ability of different methods to fill gaps of 1-, 6-, 12-, 24-, 48-, and 72-hour lengths to capture the multi-frequency fluctuations using hourly data input.

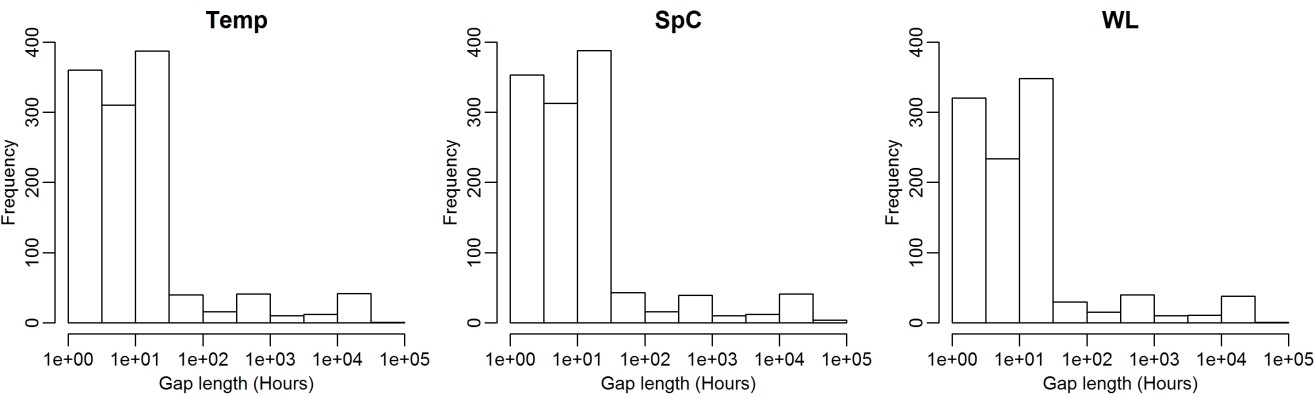

**Figure 3.** Histograms of gap lengths for each monitored variable, aggregated across all wells in the monitoring network during 2008-2018.

## 3  Gap-Filling Methods

In this section, we describe two methods we implemented to fill gaps of various lengths in SpC measurements at selected wells: an LSTM model and the traditional ARIMA model. We focused our analyses on filling gaps in SpC due to its importance in

revealing river water and groundwater mixing. The same set of analyses can be performed on water level and temperature. An input with $M$ time steps (input window length) is provided to both the LSTM and ARIMA models for predicting the next time step immediately following the input window. For gaps larger than one hour, the gap-filling models are applied to fill in one missing value at a time. The entire gap is filled by sliding the input window forward hour by hour and treating the gap-filled

values of the previous missing hours as observed values.

The input window may contain multiple variables relevant to the prediction from a single well or multiple wells. After experimenting with different sets of input variables (SpC only, SpC and water level, SpC plus water level and temperature), we found that including SpC and water level measurements in the input window yielded the most robust performance. Therefore, we used historic water level (m) and SpC ($mS/cm$) observations to fill gaps in the SpC time series of a single well. Using

measurements from multiple wells as input allows the models to account for both the temporal and spatial correlations in the data, which impacts gap-filling performance. Wells were selected based on adequate data availability and distance from the target well. Assuming the observations from $W$ ($W \geq 1$) wells are used to fill in data gaps, the input size of the model is then $M \times 2W$.

### 3.1 LSTM Models for Gap Filling

We designed an LSTM architecture, shown in Figure 4, to train models of an input size of $M$ time steps and an output size of one time step. The LSTM model contains a single LSTM layer followed by an output dense layer. The supplemental materials show the detailed structure of the LSTM layer (Figures S1 and S2).

Training data for the LSTM models were created by finding data segments of $M + 1$ hours with no missing values, i.e., no gaps in the data, for both water level and SpC measurements over a specified monitoring window. The well data were then

pre-processed by normalizing all measurements to zero mean and unit variance for each variable, as SpC is on a scale of $10^{-1}$, and water level is on a scale of $10^2$. Validation datasets were used to select the best model hyperparameters and optimal input window size $M$ (3.1.1) for gap filling at each well. Another independent testing period was selected at each well, depending on data availability, to compare the gap filling performance using the LSTM and ARIMA methods. The complete set of alternatives we considered for each LSTM model configuration is shown in Table 1. We used the Adam optimizer (Kingma and Ba, 2014)

for training and the mean-squared error as the loss function. The models were trained for 50 iterations (i.e., epochs) over the training data.

To evaluate the accuracy of the trained LSTM models in filling SpC data gaps during the validation and testing processes, we assumed that synthetic gaps of various lengths (e.g., 1, 6, 12, 24, 48, and 72 hours, referred to as gap scenarios hereafter) exist in the validation or testing dataset of a well. We assume that only the SpC measurements are missing while the water level

measurements are available. Then, an LSTM model configured with an input of $M$ is given $M$ hours of data from the time series preceding the occurrence of a gap (assuming no missing values in these $M$ hours) to fill in hour by hour. The accuracy of the gap-filling model is evaluated by calculating the mean absolute percentage error (MAPE; %) between the filled in (i.e.,

**(a) LSTM architecture**

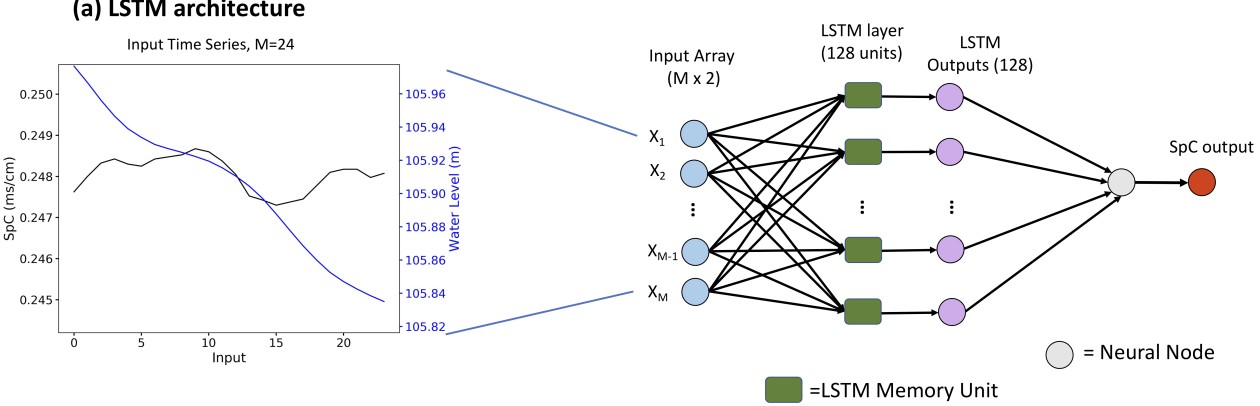

**(b) Example LSTM memory unit**

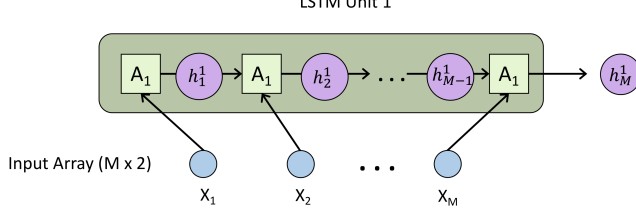

**Figure 4.** Illustration of LSTM models for gap filling. (a) Architecture of the LSTM models, where $M$ is the input window size. Includes example input with $M = 24$ and example LSTM layer with $128$ units; (b) Example of an LSTM unit, where $A$ is the repeating module of the LSTM unit and $h$ is the output.

predicted) and observed SpC values:

$$MAPE = 100 \times \frac{1}{n} \sum_{t=1}^{n} \left| \frac{\text{Prediction} - \text{Observation}}{\text{Observation}} \right|, \tag{1}$$

where $n$ is the total number of synthetic missing data points during the evaluation period.

In addition to MAPE, the models are evaluated using the Nash–Sutcliffe model efficiency coefficient (NSE) (Nash and Sutcliffe, 1970) and the Kling-Gupta Efficiency (KGE) (Gupta et al., 2009) metric. NSE is a metric used to assess the predictive skills and accuracy of hydrological models. Values range from $-\infty$ to 1, where 1 indicates a perfect model fit, 0 indicates that the model has the same predictive power as the observational mean, and a value less than 0 indicates that the model is a worse predictor than the observational mean. The NSE of the SpC predictions is calculated using the following equation:

$$NSE = 1 - \frac{\sum_{t=1}^{n} (P_t - O_t)^2}{\sum_{t=1}^{n} (O_t - \mu(O))^2}, \tag{2}$$

where $n$ is the total number of synthetic missing data points during the evaluation period, $P_t$ and $O_t$ are the predicted and observed SpC values at time $t$, and $\mu(O)$ is the mean observed SpC value.

KGE is another goodness-of-fit metric used to evaluate hydrological models by combining the three components of the NSE model errors (i.e. correlation, bias, ratio of variances or coefficients of variation) in a more balanced way. It has the same range of values as NSE, where 1 indicates a perfect model fit. KGE is calculated on a model's SpC predictions by the following equations:

$$KGE = 1 - \sqrt{(r-1)^2 + (\alpha-1)^2 + (\beta-1)^2} \qquad (3)$$

$$r = \frac{\text{cov}(O, P)}{\sigma(O) * \sigma(P)} \qquad (4)$$

$$\alpha = \frac{\sigma(P)}{\sigma(O)} \qquad (5)$$

$$\beta = \frac{\mu(P)}{\mu(O)}, \qquad (6)$$

where cov is the covariance, $\sigma$ is the standard deviation, and $\mu$ is the arithmetic mean.

**Table 1.** Parameters used in training single-well LSTM models.

| Parameter | Values |
|---|---|
| **Training wells** | 1-1, 1-10A, 1-15, 2-2, 2-3, 2-5 |
| **Synthetic gap length (hours)** | 1, 6, 12, 24, 48, 72 |
| **Model input window ($M$ hours)** | 24, 48, 72, 96, 120, 144, 168 |
| **LSTM Units ($U$ units)** | 32, 64, 128 |
| **Learning Rate ($L$)** | 1e-3, 1e-4, 1e-5 |
| **Training period** | 2012-2015 |
| **Validation period**[a] | 2011 |
| **Testing Period**[b] | 2008 for well 2-5; 2017 for well 1-15; 2016 for all other wells |

[a] used to select the best LSTM model configurations and hyperparameters.

[b] used to evaluate performance of LSTM vs ARIMA.

In addition to the LSTM models trained for the single-well setup, we also trained multi-well LSTM models that used observations from wells 1-1, 1-10A, and 1-16A to fill in data gaps for well 1-1. We explored the same set of configuration parameters in the multi-well LSTM models as shown in Table 1 for the single-well models. We then compared the gap filling performance

of the multi-well LSTM with the single-well LSTM model for well 1-1. The multi-well models were not explored for other wells due to lack of close proximity neighboring wells.

### 3.1.1 Optimizing LSTM model configuration

We used a grid-search approach to explore different LSTM model hyperparameter configurations to find the best model for a
given gap length at each well. This involved iterating over all combinations of the input time window size ($M$), the number of units ($U$) in the LSTM layer, and the learning rate ($L$) listed in Table 1 for each well. We chose the optimal LSTM configuration using model performance on the validation data set (see Table 1) based on the MAPE defined in Eq. (1). The combinations that yielded the lowest SpC MAPEs were selected as the best configuration for a given gap length at each well. These configurations are shown in Table S1 in the supplemental material. The best model configurations were then used to evaluate the LSTM-based
gap filling method against the ARIMA-based method (3.2) using relative errors (similar to MAPE by setting n=1 in Eq. (1) and removing the absolute value operation) calculated for each data point in the testing period (Table 1). These varied among the wells due to availability of continuous data required for testing.

### 3.2 ARIMA Models for Gap Filling

ARIMA is one of the most general model classes for extrapolating time series to produce forecasts. We used it as a baseline to
compare and assess the LSTM gap-filling method. ARIMA can be applied to nonstationary time series data using a combination of differencing, autoregressive, and moving average components. A nonseasonal ARIMA($p, d, q$) model is given by:

$$Y_t = c + \phi_1 Y_{t-1}^d + \phi_p Y_{t-p}^d + ... + \theta_1 e_{t-1} + \theta_q e_{t-q} + e_t, \tag{7}$$

where $\phi$s and $\theta$s are polynomials of orders p and q, respectively, each containing no roots inside the unit circle. $e$s are the error terms, $Y^d$ is $Y$ differenced $d$ times, and $c$ is a constant. Note that only non-seasonal terms $(p, d, q)$ are included in Eq. (2). Sea-
sonal structure can be added with parameters $(P, D, Q)_m$ to the base ARIMA model to convert it to ARIMA$(p, d, q)(P, D, Q)_m$, with a periodic component containing $m$ periods. $c \neq 0$ implies a polynomial of order $d + D$ in the forecast function. The detailed mathematical equations for the seasonal ARIMA are provided in the online supplemental materials.

The main task in ARIMA-based forecasting is to select appropriate model orders, i.e., the values of $p, q, d, P, Q,$ and $D$. If $d$ and $D$ are known, we can select the orders $p, q, P,$ and $Q$ via an information criterion such as the Akaike Information Criterion
(AIC):

$$AIC = -2\log(L) + 2(p + q + P + Q + k), \tag{8}$$

where $k = 1$ if $c \neq 0$ and 0 otherwise, and $L$ is the likelihood of the model fitted to the differenced data. The ARIMA models were built using the `auto.arima` function from the R package `forecast` (Hyndman et al., 2007), which applies the Hyndman-Khandakar algorithm (Hyndman and Khandakar, 2008) that minimizes the AIC to obtain the best-fit parameters of
the ARIMA model.

Similar to the LSTM-based gap filling, we explored various input window sizes, from 24 to 504 hours in increments of 24 hours, for the ARIMA model at each well to identify the optimal input windows within the search range. An optimal input window size is chosen for each gap length of each well using the same MAPE metric (i.e., Eq. (1)) on the validation dataset.

## 4 Results and Discussion

### 4.1 Performance of single-well LSTM models

We selected the best combination of LSTM units ($U$) and learning rate (L) for each input time window ($M$) under each gap length at each well using the MAPE metric. The validation MAPEs of those selected models were summarized in boxplots under different grouping, examples of which are shown in Figure 5. Each MAPE boxplot was drawn from a group of models with one parameter (corresponding to each x-axis) fixed at a given value while the other parameters cycle through their possible combinations.

As shown in Figure 5 (a), model performance deteriorates as the gap length increases, indicating that the LSTM-based method tends to lose ground truth information from its input to inform prediction. In comparing MAPEs across the various input window sizes shown in Figure 5 (b), we observe that models with all input windows have comparable MAPE summary statistics, with larger input windows ($>$ 96 hours) leading to slightly smaller MAPE quartiles. The larger input windows also yield fewer outliers on the larger MAPE end, indicating that the memory units in the LSTM layers are capturing important daily to weekly signatures for some wells (evident in WPS plots in Figure 2 for all wells except for well 1-15).

The performance of single-well LSTM models varied among the wells as shown in Figure 5 (c). The LSTM models for well 1-15 lead performance with the smallest MAPEs, while the model of well 2-2 yields the worst performance. The LSTM models for wells 1-1,1-10A, 2-3 and 2-5 performed comparably overall, with slightly more large MAPE outliers for well 2-3.

### 4.2 Single-well LSTM and ARIMA comparisons

The single-well LSTM gap filling approach was compared to the ARIMA approach using the relative errors calculated for each data point assumed to be missing in the testing data by setting n=1 in Eq. (1) for MAPE. Relative errors were used to show the overestimations or underestimations of both approaches. Their respective best model configurations, determined on the validation dataset (i.e., data from year 2011), were used to compare the two approaches. Figure 6 illustrates the optimal input windows for the LSTM and ARIMA methods. We observe that the LSTM models require much less input information than the ARIMA method under all gap lengths for all the wells.

Figure 7 shows the boxplots of relative errors under different gap lengths for all testing wells. As expected, the relative errors increase as the gap length increases for both approaches. The ARIMA models tend to perform better than the LSTM models in terms of interquartile range. However, the ARIMA models produce more outliers of large positive or negative relative errors than the LSTM models in general and particularly for larger gap lengths (48 and 72 hours). For well 1-15, the relative errors of both approaches are small for all gap lengths. Both approaches appear to have larger error outliers at well 2-3.

For each well, we performed a T-Test to calculate the T-Score and P-Value between the relative errors of the two models to determine the significance in the performance difference of the models. As seen in Table 2, each well has a high T-Score and P-Value significantly less than 0.05. Thus, the differences between their relative errors are significant and meaningful.

In addition to the relative errors, we calculated the MAPE, Root Mean Squared Error (RMSE), NSE, and KGE for the best LSTM and ARIMA model for each gap length. Table 2 compares the performance of the LSTM and ARIMA models filling in gap lengths of 24 hours. The table for all gap lengths is in the online supplemental material (Table S2).

The LSTM and ARIMA models yielded comparable average metrics at all wells for the 24 hour gap length, as can be seen in Table 2. The NSE and KGE resulting from both models are close to 1 for all the wells with negligible differences between the two models. The difference in MAPE and RMSE is also small, with more notable differences for wells 2-2 and 2-3, where the ARIMA models resulted in lower MAPE and RMSE.

**Table 2.** Comparison of single-well LSTM and ARIMA models for a 24-hour synthetic gap in the SpC data of the test set for each well. The models are the same ones used in Figure 8. The calculated statistics are: MAPE, Root Mean Squared Error (RMSE), Nash–Sutcliffe model efficiency coefficient (NSE), and Kling-Gupta Efficiency (KGE). The T-Score and P-Value are calculated for the relative errors of the two models per well.

| Well | Model Type | MAPE | RMSE | NSE | KGE | T-Score | P-Value |
|------|-----------|------|------|-----|-----|---------|---------|
| 1-1 | LSTM | 1.38 | $8.33 \times 10^{-3}$ | 0.991 | 0.988 | 19.1 | $1.00 \times 10^{-80}$ |
| | ARIMA | 1.36 | $8.98 \times 10^{-3}$ | 0.989 | 0.994 | | |
| 1-10A | LSTM | 1.37 | $8.07 \times 10^{-3}$ | 0.986 | 0.968 | $-24.6$ | $1.48 \times 10^{-131}$ |
| | ARIMA | 1.5 | $9.60 \times 10^{-3}$ | 0.98 | 0.987 | | |
| 1-15 | LSTM | 0.259 | $1.88 \times 10^{-3}$ | 0.989 | 0.982 | $-48.9$ | 0.00 |
| | ARIMA | 0.119 | $1.18 \times 10^{-3}$ | 0.996 | 0.997 | | |
| 2-2 | LSTM | 2.97 | $1.87 \times 10^{-2}$ | 0.922 | 0.962 | 48.1 | 0.00 |
| | ARIMA | 2.23 | $1.64 \times 10^{-2}$ | 0.939 | 0.967 | | |
| 2-3 | LSTM | 2.15 | $1.63 \times 10^{-2}$ | 0.945 | 0.965 | 21.6 | $4.69 \times 10^{-102}$ |
| | ARIMA | 1.72 | $1.48 \times 10^{-2}$ | 0.954 | 0.971 | | |
| 2-5 | LSTM | 0.929 | $6.86 \times 10^{-3}$ | 0.976 | 0.988 | $-9.6$ | $9.22 \times 10^{-22}$ |
| | ARIMA | 0.866 | $7.45 \times 10^{-3}$ | 0.971 | 0.977 | | |

In addition to the error statistics, it is also important to examine how well a gap-filling method captures the desired dynamics patterns in the gap-filled time series. Therefore, the SpC time series reproduced by the gap-filling methods during the testing period (2016 for wells 1-1, 1-10A, 2-2, 2-3; 2017 for well 1-15; 2008 for well 2-5) with 24-hour synthetic gaps are evaluated against the real time series. The model configurations are the same as those used in the error statistics comparison (Figure 6). A gap length of 24 hours is the selected example as we consider it a reasonably challenging case to fill the gaps in time series

data exhibiting significant nonstationarity, such as the SpC data at well 2-3. Moreover, the relative performance of the two approaches is similar at other gap lengths with varying error magnitudes.

As shown in the first column of Figure 8, both approaches capture the general dynamic patterns in the data fairly well. For more detail, the time series of relative errors for both methods are provided in Figure S3 in the online supplemental materials. The ARIMA approach (blue lines in column 1) missed some abrupt changes that occur over a short time window (i.e., at higher frequency), leading to more error spikes in all wells. This is consistent with the relative error outliers in Figure 7 and is an indication that the ARIMA models lack mechanisms to represent such high-frequency changes. The LSTM approach (red lines in column 1 plots) is able to better capture such dynamics in all the wells. However, the inclusion of the high-frequency fluctuations may also lead to overly dynamic predictions in time windows dominated by lower-frequency fluctuations. This contributed to less desirable relative errors distributed between the first and third quartiles in some wells (i.e., wells 1-1, 2-3, 2-2, and 1-15), as shown in Figure S4 in the supplemental materials. The errors are likely caused by the variability in dynamics signatures among the training, validation, and testing periods, as well as the selection of the training loss functions and validation metric that balance between the occurrence of small vs large errors to achieve optimal solutions.

To further investigate the dependence of the relative performance of the two gap-filling methods on the inherent dynamics in each time series, spectral analyses for the testing SpC datasets were performed using the same wavelet decomposition method for the multi-year analyses (shown earlier in Figure 2). As shown in Figure 8, the time windows of high relative errors are found to approximately co-locate with the time when the high-frequency (daily and subdaily) signals gain more power. The differences between the LSTM and ARIMA models tend to be amplified during those time windows. Wells 1-1, 1-10A, and 2-2 share similar seasonal patterns in WPS, with the highest intensity bin above 1024 hours across February to July. Their average WPSs all show peaks around daily and subdaily frequencies. Well 2-3 has its greatest energy between 16 and 256 hours from January to July. Well 2-5 has low variability intensities at daily and subdaily frequencies with low-frequency variations (monthly and seasonal) dominating the Jan to March time frame. For well 1-15, one of its strongest intensities is above 2048 hours across the entire year and the other strong intensities are narrow bands between 16 and 256 hours. In general, both LSTM and ARIMA are effective at capturing low-frequency variability (monthly and seasonal). Although LSTM is more effective at capturing high-frequency (daily and subdaily) fluctuations and nonlinearities in the datasets, it may also lead to overly dynamic predictions in time windows with no considerable high-frequency fluctuations. However, the errors during these time windows are small and can be improved by smoothing if such fluctuations are undesirable.

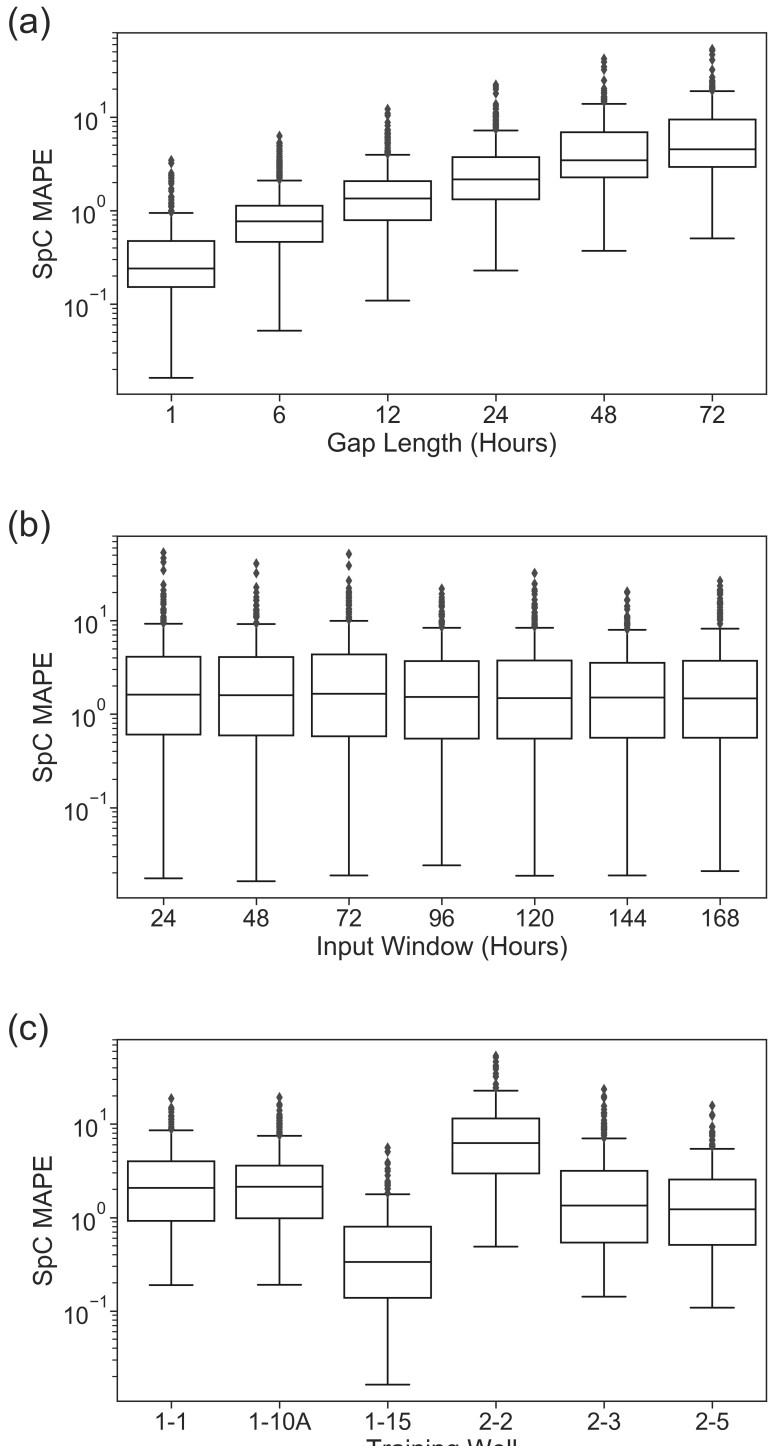

**Figure 5.** Gap filling performance for SpC evaluated against the validation datasets, grouped by gap lengths (a), model input window size $M$ (b), and training wells (c).

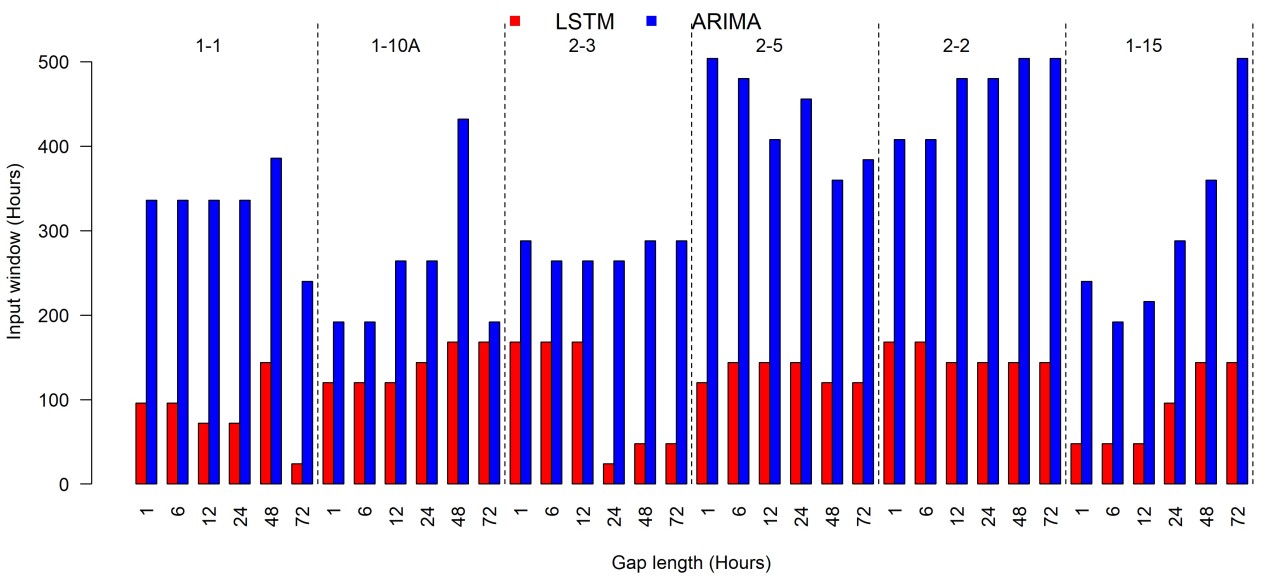

**Figure 6.** Optimal input windows for the LSTM and ARIMA models to fill gaps of various lengths at each well.

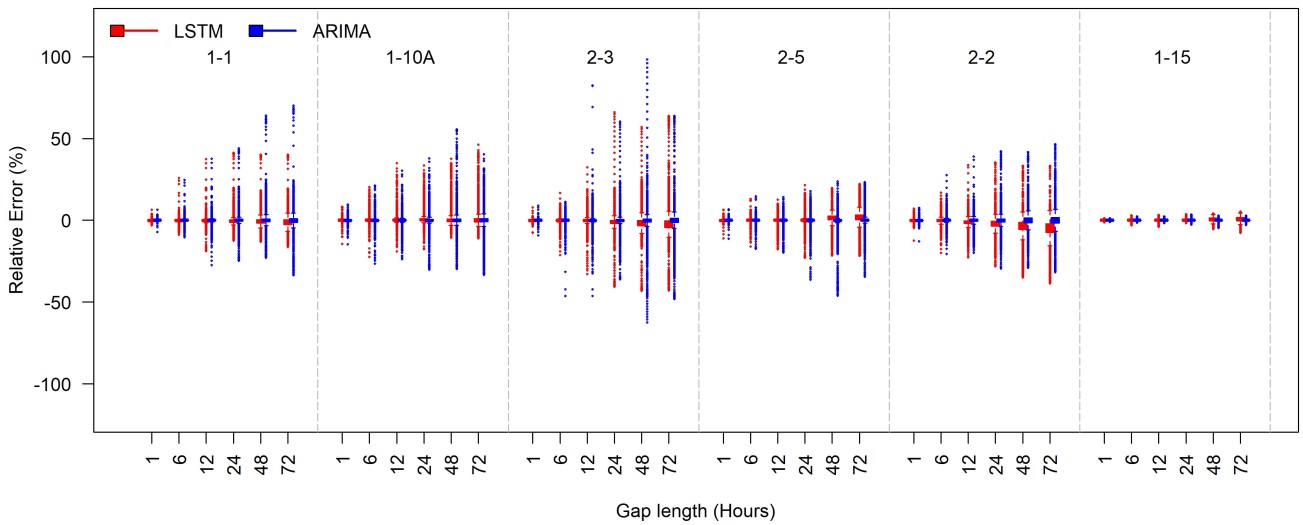

**Figure 7.** Boxplots of the relative errors for filling SpC gaps of various lengths (1, 6, 12, 24, 48, and 72 hours) at each well during the test periods. The best LSTM and ARIMA models were used for evaluation. The LSTM and ARIMA models are represented by red bars and blue bars, respectively.

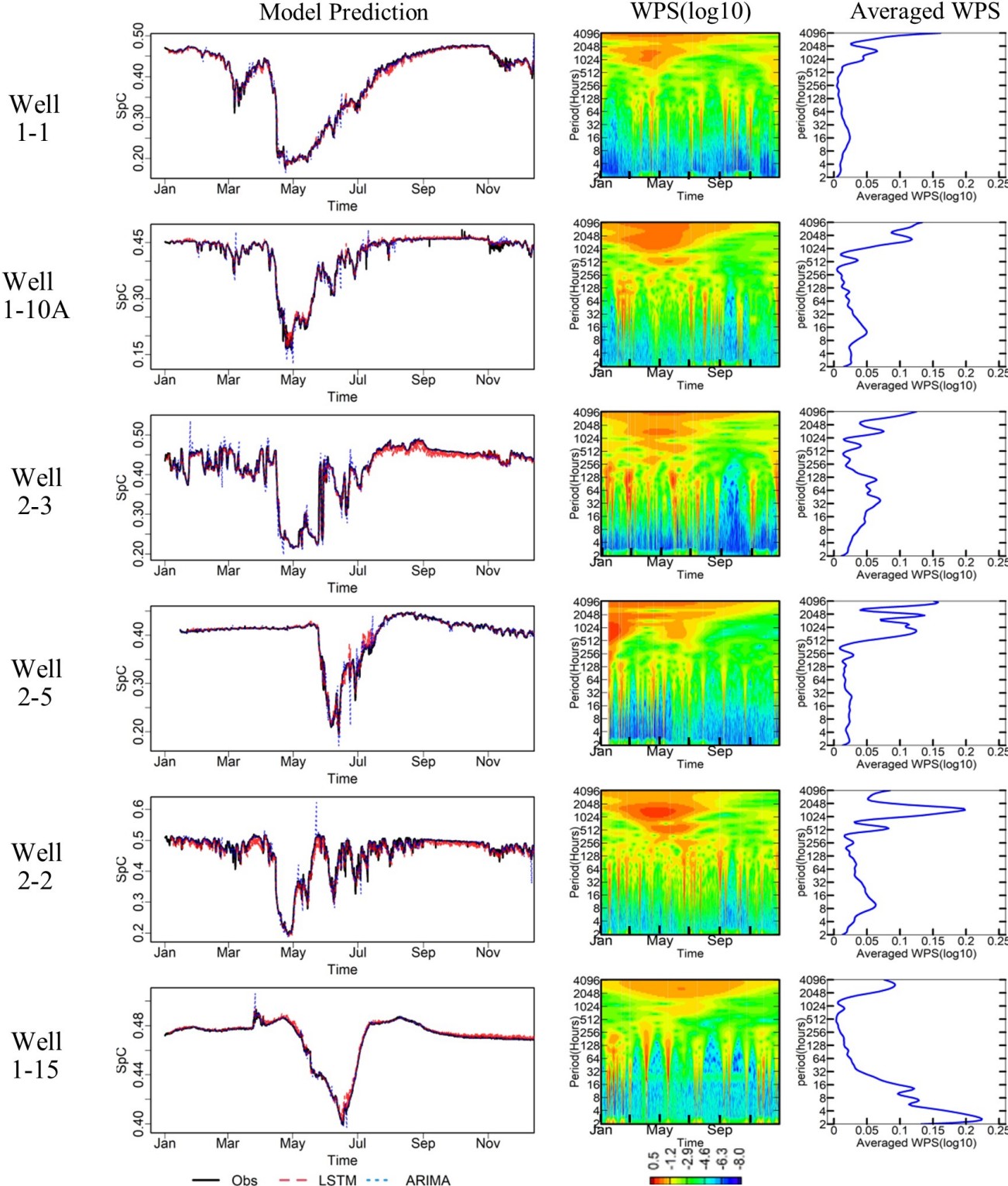

**Figure 8.** Columns 1 shows time series of model predictions from the LSTM (in red) and ARIMA (in blue) methods, respectively, assuming a 24-hour synthetic gap in the SpC data compared with observations (in black). The best model configurations were used for all models. The testing data come from year 2016 for wells 1-1, 1-10A, 2-2, and 2-3; from year 2017 for well 1-15; and from 2008 for well 2-5. Column 2 is the spectrogram of each well and column 3 is the averaged WPS for the corresponding year.

There is also a significant difference in the computational cost for the LSTM and ARIMA methods. ARIMA requires very few computational resources: the `auto.arima` function in R requires approximately 40 seconds for fit and validate a model for each prediction segments on a personal computer with a 3.00 GHz CPU. Conversely, training and validating a single LSTM model takes approximately 20-30 minutes on dual NVIDIA P100 12GB PCI-e based GPUs.

## 4.3 Performance of multi-well models

We evaluated the predictive ability of multi-well models using both approaches for filling gaps of various lengths in the SpC data at well 1-1 by comparing their performance against their single-well counterparts. Well 1-1 was chosen because of the data availability for nearby wells (wells 1-10A and 1-16A). Similar to the single-well ARIMA and LSTM model for well 1-1, the multi-well models also predict the SpC measurement using water level and SpC from three wells. We adopted the same LSTM architecture from the single-well LSTM model and trained the same set of alternatives considering input window sizes, LSTM units, and learning rates for various gap lengths as listed in Table 1. The same training and validation periods were adopted to select the optimal combination of $M$, $U$, and $L$. The optimal combinations are shown in Table S3 in the online supplemental material. For the multi-well ARIMA models, we included additional variables as regression terms when building and fitting models using the `auto.arima` function. The optimal input window sizes of ARIMA are 216, 240, 288, 288, 288, and 192 hours for gap lengths 1, 6, 12, 24, 48, and 72 hours, respectively. These are smaller than the optimal window sizes of the single-well models.

The boxplots of relative errors yielded from the single-well and multi-well models using both approaches are provided in Figure 9 for comparison. Additionally, we include performance metrics for comparing the single and multi-well models in Table 3. Additional spatial information seems to exacerbate the relative errors of the ARIMA models, except in large gaps (e.g., 72 hours). The LSTM approach, on the contrary, benefits from the information carried by the neighboring wells when filling in the larger gaps, while their performance for small gaps stay unchanged. The aggregated performance metrics in Table 3 show slightly improved metrics for multi-well ARIMA models for gaps smaller than 24 hours compared to the single-well models, while the turning point in relative performance is at 12 hours for the LSTM models. The deteriorated performance metrics of the multi-well LSTM models at the larger gap lengths are consistent with the larger inter-quartile ranges as shown in the boxplots of relative errors in Figure 9. However, the multi-well LSTM and ARIMA models can reduce the occurrence of large relative errors for larger gaps and provide more robust gap-filling under those circumstances.

These comparisons show that, although the information from a single well may be sufficient to fill in gaps smaller than a day, including spatial information from neighbouring wells in the LSTM and ARIMA models could potentially increase the chance of successes in filling data gaps under more challenging circumstances, such as capturing more complex dynamic patterns with

longer data gaps. While the aggregated metrics provide an overall assessment of model performance, examining the distribution of relative errors could provide complementary information on large error spikes while selecting optimal model configurations.

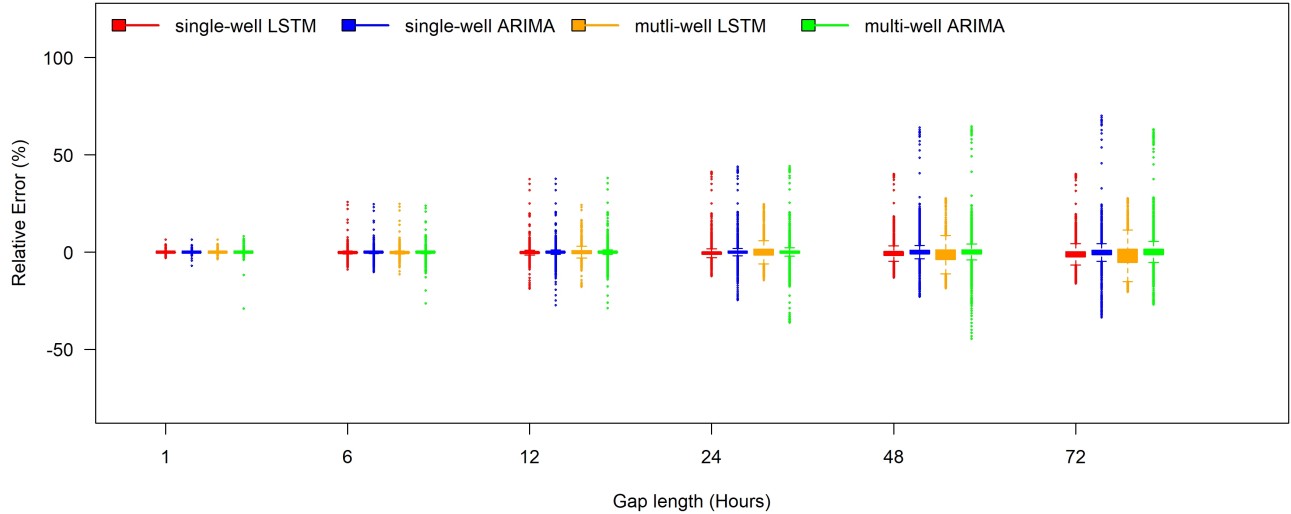

**Figure 9.** Comparing relative error performance between the best single-well LSTM models (well 1-1, red), multi-well LSTM models (wells 1-1, 1-10A, and 1-16A, yellow), single-well ARIMA model (blue), and multi-well ARIMA model (green) for filling in various SpC gap lengths for well 1-1 during the test period (year 2016).

## 5   Conclusion

In this study, we implemented an LSTM-based gap filling method to account for spatio-temporal correlations in monitoring
data. We extensively evaluated the method on its ability to fill data gaps in the groundwater SpC measurements that are often used to indicate groundwater and river water interactions along river corridors. We took advantage of a 10-year, spatially distributed, multi-variable time series dataset collected by a groundwater monitoring well network and optimized an LSTM architecture for filling SpC data gaps. A primary advantage of using LSTM is its ability to incorporate spatio-temporal correlations and nonlinearity in model states without a priori assuming an explicit form of correlations or nonlinear functions in
advancing system states. We compared the performance of a single-well LSTM-based gap-filling method with a traditional gap-filling method, ARIMA, to evaluate how well an LSTM model can capture multi-frequency dynamics. We also trained LSTM and ARIMA models that take input from multiple wells to predict responses at one well. The multi-well models were compared with single-well models to identify and assess improvements in gap filling performance from including additional spatial correlation from neighboring wells.
In general, both LSTM and ARIMA methods were highly accurate in filling smaller data gaps (i.e., 1 and 6 hours). They were reasonably effective at filling in medium gaps between 12 and 48 hours, while more work is needed for gaps larger than 48

**Table 3.** Comparison of single-well and multi-well LSTM and ARIMA models for all synthetic gap lengths in the SpC data. The models are the same as those used in Figure 9. Calculations are performed on the test data set for well 1-1 (year 2016). The calculated statistics are: MAPE, Root Mean Squared Error (RMSE), Nash–Sutcliffe model efficiency coefficient (NSE), and Kling-Gupta Efficiency (KGE). T-Scores and P-Values are calculated on the relative errors of the two single-well models for each gap length and calculated on the relative errors of the two multi-well models.

| Gap Length | Model Type | MAPE | RMSE | NSE | KGE | T-Score | P-Value |
|---|---|---|---|---|---|---|---|
| 1 | Single-Well LSTM | 0.117 | $7.76 \times 10^{-4}$ | 1.0 | 0.999 | 11.6 | $6.14 \times 10^{-31}$ |
| | Single-Well ARIMA | 0.183 | $1.23 \times 10^{-3}$ | 1.0 | 1.0 | | |
| | Multi-Well LSTM | 0.117 | $7.94 \times 10^{-4}$ | 1.0 | 1.0 | 9.37 | $8.54 \times 10^{-21}$ |
| | Multi-Well ARIMA | 0.134 | $1.22 \times 10^{-3}$ | 1.0 | 1.0 | | |
| 6 | Single-Well LSTM | 0.435 | $2.92 \times 10^{-3}$ | 0.999 | 0.995 | 13.8 | $4.46 \times 10^{-43}$ |
| | Single-Well ARIMA | 0.461 | $3.29 \times 10^{-3}$ | 0.999 | 0.999 | | |
| | Multi-Well LSTM | 0.435 | $2.98 \times 10^{-3}$ | 0.999 | 0.998 | 12.2 | $5.82 \times 10^{-34}$ |
| | Multi-Well ARIMA | 0.405 | $3.16 \times 10^{-3}$ | 0.999 | 0.999 | | |
| 12 | Single-Well LSTM | 0.75 | $5.07 \times 10^{-3}$ | 0.997 | 0.996 | 16.0 | $2.05 \times 10^{-57}$ |
| | Single-Well ARIMA | 0.781 | $5.47 \times 10^{-3}$ | 0.996 | 0.996 | | |
| | Multi-Well LSTM | 1.19 | $6.48 \times 10^{-3}$ | 0.994 | 0.985 | 5.25 | $1.55 \times 10^{-7}$ |
| | Multi-Well ARIMA | 0.77 | $5.40 \times 10^{-3}$ | 0.996 | 0.997 | | |
| 24 | Single-Well LSTM | 1.38 | $8.33 \times 10^{-3}$ | 0.991 | 0.988 | 19.1 | $1.00 \times 10^{-80}$ |
| | Single-Well ARIMA | 1.36 | $8.98 \times 10^{-3}$ | 0.989 | 0.994 | | |
| | Multi-Well LSTM | 2.26 | $1.17 \times 10^{-2}$ | 0.982 | 0.968 | 7.77 | $8.48 \times 10^{-15}$ |
| | Multi-Well ARIMA | 1.47 | $9.55 \times 10^{-3}$ | 0.988 | 0.99 | | |
| 48 | Single-Well LSTM | 2.13 | $1.21 \times 10^{-2}$ | 0.98 | 0.988 | 17.8 | $3.24 \times 10^{-70}$ |
| | Single-Well ARIMA | 2.15 | $1.34 \times 10^{-2}$ | 0.976 | 0.988 | | |
| | Multi-Well LSTM | 3.49 | $1.76 \times 10^{-2}$ | 0.958 | 0.969 | 28.4 | $7.83 \times 10^{-174}$ |
| | Multi-Well ARIMA | 2.35 | $1.40 \times 10^{-2}$ | 0.974 | 0.981 | | |
| 72 | Single-Well LSTM | 2.56 | $1.40 \times 10^{-2}$ | 0.974 | 0.983 | 26.7 | $4.91 \times 10^{-154}$ |
| | Single-Well ARIMA | 2.57 | $1.46 \times 10^{-2}$ | 0.971 | 0.985 | | |
| | Multi-Well LSTM | 4.41 | $2.19 \times 10^{-2}$ | 0.936 | 0.955 | 31.7 | $4.78 \times 10^{-214}$ |
| | Multi-Well ARIMA | 3.02 | $1.78 \times 10^{-2}$ | 0.958 | 0.972 | | |

hours. Both models captured the long-term trends in data (i.e., low-frequency variations at monthly or seasonal time scales). The ARIMA method was found to have difficulty capturing abrupt changes and is thus more suitable for time series with less dynamic behavior. Compared with the ARIMA models, the LSTM models excel in dealing with high-frequency dynamics (daily and subdaily) and nonlinearities, although they require more training data and computational resources. As a side effect

of including high-frequency (daily and subdaily) fluctuations in the model, the LSTM approach may produce overly dynamic predictions in time windows that lacks dynamics. As with any deep learning method, the availability of sufficient training data that cover a wide range of conditions is critical for the success of LSTM methods. Extrapolating the LSTM models to conditions beyond those in the training data remains a major challenge.

Wavelet analysis could provide useful insights into the dynamic signatures of the data and changes in the composition of their important frequencies over time, which can serve as a basis for selecting an appropriate gap-filling method. For example, the ARIMA method would work well if the dynamics are dominated by seasonal cycles, while more sophisticated approaches like LSTM-based methods could work better if there is evidence of weekly, daily, and subdaily fluctuations. Depending on the mixture of high- and low-frequency variability inherent in the time series, different LSTM architecture and configurations can
be explored and evaluated through hyperparameter searches with respect to LSTM layers, dense layers, and activation functions to achieve better performance in capturing complex dynamics. The optimal LSTM model configuration and achievable performance would vary case by case.

We also demonstrated that incorporating spatial information from neighboring stations in LSTM models could contribute to performance improvements under challenging scenarios with dynamic system behaviours and longer data gaps of up to 2
15 days. However, other alternatives need to be explored for gaps beyond 2 days. Bidirectional and convolutional LSTMs are two promising methods to leverage information from the future time window and spatially distributed networks, respectively. While we introduced a new method that can be broadly applied to fill in gaps in an irregularly spaced network for monitoring groundwater and surface water interactions, the transferrability of this method to other monitoring systems could be more extensively evaluated by community participation. When applying LSTM or other DNN-based methods for gap filling, it is
20 important to rigorously evaluate the model performance by asking the following questions: (1) what is the dynamics signature of the data to be filled? (2) how is gap filling performance impacted by the length of gaps? (3) how does the amount of training data impact the model performance? (4) how does the choice of the input time window impact gap filling performance? and (5) How much value can measurements at neighboring add to the performance improvement? Better capturing spatio-temporal dynamics in system states is essential for generating the most valuable insights to advance our understanding of dynamic
complex systems.

*Code and data availability.* The well observations have been made accessible at https://sbrsfa.velo.pnnl.gov/datasets/?UUID=14febd81-05b6-47fb-be52 -439c4382decd

*Author contributions.* HR and EC developed scripts and performed the analyses and they have equal contribution to the paper. BK contributed on interpretation of the results. XC conceived and designed the study. All authors contributed to writing the manuscript.

*Competing interests.* The authors declare that they have no conflicts of interest.

*Acknowledgements.* This research was supported by the U.S. Department of Energy (DOE), Office of Biological and Environmental Research (BER), as part of BER's Environmental System Science (ESS) program. A portion of methodology development was supported by the Laboratory Directed Research and Development Program at Pacific Northwest National Laboratory, a multiprogram national laboratory operated by Battelle for DOE under contract DE-AC05-76RL01830. This research was performed using PNNL Institutional Computing at Pacific Northwest National Laboratory. This research was also supported in part by the Indiana University Environmental Resilience Institute and the *Prepared for Environmental Change* grand challenge initiative. This paper describes objective technical results and analysis. Any subjective views or opinions that might be expressed in the paper do not necessarily represent the views of the U.S. Department of Energy or the United States Government.

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
