# Peer review of "Technical note: Using Long Short-term Memory Models to Fill Data Gaps in Hydrological Monitoring Networks"

_Hydrology and Earth System Sciences, 2019_

## Short Comment (SC1) · 21 Jun 2019

Short Reviews of HESS-2019-196: This paper uses long short-term memory (LSTM) neural networks to fill in gaps in spatially distributed time-series data. The performance of the LSTM-based gap-filling method is compared to that of a traditional, popular gap-filling method: autoregressive integrated moving average (ARIMA). Overall, this paper is well written, structured and results seem sufficiently justified and useful. However, this paper is very technical and there is no physical insight beyond just feeding data into a standard code. I think this paper should be published as technical note (not as research article). Several aspects could be further improved in order to having it published in this journal.

[Figure]

(1)Would you guarantee the LSTM method in your paper can achieve the same excellent performance in other areas of the whole world? Is it possible that the good performance of the LSTM model is just applicable for the case given by the manuscript? The authors should include one more test for another area (maybe not in the text but in the supporting materials).

(2) LSTM model is only compared to ARIMA. Why not compare LSTM with other widely-used methods (such as Kriging interpolation and Gaussian process)? Furthermore, are the authors familiar with DIEOF (Data Interpolating Empirical Orthogonal Functions) which are proposed by Beckers and Rixen (2003)? I think that DIEFOF is powerful and useful for filling temporal and spatial gaps in geophysical datasets. Maybe the authors can compare LSTM with DIEOF.

(3) The present title "Using Deep Learning to Fill Spatio-Temporal Data Gaps in Hydrological Monitoring Networks" are inaccurate. I suggest new title like "Using Long Short Term Memory Neural Network Model to Fill Spatio-Temporal Data Gaps in Hydrological Monitoring Networks"

Reference: Beckers, J. M., and M. Rixen (2003), EOF Calculations and Data Filling from Incomplete Oceanographic Datasets, J. Atmos. Ocean. Technol., 20(12), 1839–1856.

---

## Referee Comment (RC1) · Gerald A Corzo P (Referee) · 30 Aug 2019

The paper presents an interesting use of deep learning with LSTM Networks for infilling groundwater data. The article is timely and tries to make a comprehensive description and explanation of how the Deep learning technique is implemented using statistical and machine learning techniques. The paper is a welcome contribution to the field of groundwater and hydrological earth sciences. However, I cannot recommend publication in the present form due to the comments and questions raised. The paper needs major revision.

1. The paper states that long-term spatiotemporal changes in subsurface hydrological flow is usually quantified using a network of wells. However this paper does not deal

with the long-term trend or analysis. Hourly data is hardly interpreted or used for the long term. Hourly information for sure contains noise that would be advisable to remove for the long term analysis.

2. Observations are mentioned to be spatially sparse, and temporal gaps exist. Many papers have solved the same type of problem, without using the term spatiotemporal. Almost every course in hydrology deals in one chapter with the issue of using spatial correlation and temporal correlation to fill in data. So in this respect, the authors are invited to clearly indicate what innovation is brought by this work to spatiotemporal analysis.

3. Following point two, it is known that in most of the cases, aquifers with little or no human intervention have low variability. Conventional guidelines and measures in hydrogeological science are typically based on monthly data.

4. In the present paper the idea of nonlinear dynamics is mentioned almost everywhere in the introduction and justification of the work. This is somewhat surprising and needs better justification, since groundwater dynamics, in many cases, can be represented with linear models. As it is concluded in this paper results, ARIMA can approximate the system quite well.

5. The particular case study presented here shows a relative complex dynamic nature indeed, but it seems it is due to human intervention (however I could be wrong). Can you comment on this and the uncertainties associated?.

- The human intervention might affect your calculation and therefore, extractions might not be following a random but more human induced behaviour. So data understanding or replicability used in one year might not be the same in another. It would be advisable first to check how much and when extraction took place. Is this data filled in for a long term analysis, or short-term? This question arises since the hourly step is used. - If indeed human intervention influence the dynamics of the groundwater system, the logical approach would be to find a variable to represent direct or indirect measurement of extractions. - It is suggested to read the paper by Amaranto et al. (2018) "Semi-seasonal groundwater forecast using multiple data-driven models in an irrigated cropland". J Hydroinformatics, 20 (6): 1227–1246. DOI: https://doi.org/10.2166/hydro.2018.002 and - Amaranto et al. ( 2019). A spatially enhanced data‐driven multimodel to improve semiseasonal groundwater forecasts in the High Plains aquifer, USA. Water Resources Research, 55, 5941– 5961. https://doi.org/10.1029/2018WR024301

6. The regional aquifer and geology might play a more significant role in the study, since not only the river but the size and other interventions and hydrometeorological recharges might be correlated.

7. The stations are so close, and the hourly variation appears to be periodic with an amplitude of 4 or 5cm, according to Figure 1 (and on other graphs). It is intriguing, the question I would have is what happens every hour? and if this hourly variation is noise on the measurement device or data? What is the precision of the measurement device? What is the volume of water extracted to reach the variation of 1 cm? Where the recharge water comes from(has this been studied in the past)? Is this 5 cm recharge volume feasible in one hour? Could be the water from the river affecting your measurements (interflow)? It is advisable to present the time series of the river flow. It would be also useful to have a few hydrological balances (note that this is a hydrological journal). The problematic still can be questioned due to its apparent complex dynamics with the river and human intervention (not a typical, natural aquifer).

8. On the model setup, Please explain why you use Mx128.

9. Page 7, line 10, mentions the supplemental material, but I cannot find it in the paper.

10. Important: choice of (a very complex model) LSTM has to be justified, since it seems AR-type models is enough. Frankly, I don't see the need for complex models like LSTM, but if you have arguments to defend your position, please present them to convince the readers.

11. On page 14, it states that other configurations of LSTM can be further explored; however, it is not clear why this was not done before. Not sure why the selected configuration was just tried to see if it works or not, without any analysis what is the best structure. This relates to comment 8 and 9.

12. I am a bit in confusion how to interpret the statements made in conclusion. The ARIMA is not suited or less suited for filling high frequency (hourly, or short gaps) and more suitable for a long term period (24, 48 and 74 hours). It is suggested we need deep learning for filling high-frequency gaps (of one hour)?. Maybe is good to elaborate on the simplicity of what this translates to, I am not sure if the meaning is right.

13. Not sure if there is an idea of how high is the overall error; in the figure 8, with well 1-15 it seems almost perfect representation (zero error in the validation data for many points). Also in the same well, it appears like high negative correlation up to 128 hours.

---

## Referee Comment (RC2) · Anonymous Referee #2 · 4 Dec 2019

General note: I was asked by the editor to review this manuscript, although groundwater hydrology is not my area of expertise. However, machine learning is and therefore most of my review will be around the methods and experimental setting used in this manuscript.

This manuscript presents an approach for filling gaps in time series of ground water well measurements. Specifically, the authors compare two different methods (LSTM-based and ARIMA) for different gap lengths for six different wells.

Although I generally welcome publications that try to make use of deep learning based methods for various applications in earth science, I see various major concerns with the manuscript at hand. Overall, it seems like the authors are not too familiar with the methods they apply (especially the LSTM-based model) and many decisions made seem questionable and lack any justification or explanation. Because of these concerns, I'm not sure if I can recommend this manuscript for publication. If it should be published at all, major revisions are required.

Major Concerns:

1. **Model architecture**: Coming from the field of machine learning, I was surprised by the creativity of the authors in finding their model architecture. To be honest, I have never seen such a combination of LSTM layers, dense layers and convolutional layers for a time series task and I wonder if the authors know what they are doing. Here is a list of sub points to this major comment:
    a. First: Did you perform any hyperparameter search at all to find this architecture? If yes, please give details on the model configurations (in terms of layers) you tried, if not, why not? To propose such an excotic architecture, it is required to see quantitative evidence that this is required and not a much simple LSTM-based model would be better (e.g. single LSTM layer with single dense + dropout layer)
    b. Why do you stack 3 LSTM layers? In theory, a single LSTM layer is turing-complete. Besides probably natural language processing, where the training data consists of million/billion of samples, there is almost always no need to use more than a single LSTM layer. Additionally, since you have very limited training data (2 years of hourly data are just 17520 data points), the size of your LSTMs seem to be exorbitantly large. Especially with 3 LSTM layers.
    c. Why the combination of convolutional layers and dense layers after the LSTM? Probably the standard is to have a single dense layer that uses the hidden output of the LSTM to map to your desired target shape. Why do you think so much complexity is needed after the LSTM, since the LSTM should capture the complex temporal dependencies already?
    d. Why do you have the convolutional layer at all? If I understand your setting correctly, the convolutional layer can again look at the entire sequence (M x 64, with M the input sequence length). Why is this necessary? The task of the LSTM is to summarize the input sequence and store all the information necessary for

predicting the M+1 time step (first step of your N time step long gap) in it's cell state.

e. Another point related to the convolutional layer. I see that the filter size was solely chosen to be able to map from a sequence length of M to an output of N (filter size M-N+1). However, are the authors aware of what that means? For example, for predicting the first of the N time steps, the convolutional filter will only look at the first M-N+1 input sequence elements, effectively ignoring what has happened at the time steps preceding the current time step. Why do you want this? It makes absolutely no sense to not include the most informative information (the previous time steps) necessary to predict the next time step.

2. **Related work:** Since (correct me if I'm wrong) this is not a forecast task, but just filling gaps in historic data records, I wonder if the authors have done some research, which approaches are currently used in the field of deep learning, before proposing their own method. E.g. for gap filling in historic time series, Bi-directional LSTMs are commonly used over normal LSTMs, since they do two sided gap filling (closer to interpolation), compared to the standard LSTM, which basically extrapolates into the future. I would also advise to add some related work section of LSTM-based gap filling into the introduction.

3. **Training setup:** There are various points around the model training setup that I see problematic. Some of them might overlap to other points mentioned above or below.
    a. Input features for any neural network should be normalized to zero mean, unit variance and not to the range of 0 to 1. This will basically bias your network during the start of the training in a wrong way. Maybe as some intuition: Most (all?) activation functions are centered around zero, e.g. the sigmoid function in all gates of the LSTM. With randomly initialized weights (which are normally initialized around 0), using your normalization would bias the entire network to always have pre-activations of larger than zero, and thus sigmoid values close to one. However, what you want is in expectancy to be undecided in the beginning (pre-activation of 0, equals to sigmoid of 0.5). Long story short, you should re-run all experiments with different normalizations, at least for the LSTM.
    b. Results of neural networks are generally affected by some stochasticity, because of the random weight initialization and the randomness of stochastic gradient descent. This requires almost always to train multiple models for the exact same setting with different random initialization (seeds) and to report the average model performance and variations across those repetitions. Otherwise, results might not be reproducible, since you might only be lucky (or unlucky) with your single initialization.
    c. In general, you have very few data points for such a large deep learning model, as already stated above. You could either think of ways, how to combine the data of all wells in a single model, or reduce your model size drastically, which is what I would propose here.

d.  I found it very hard to follow your training and testing setup, until late in the paper. E.g. around the number of possible model configurations, and total train-test combinations. I would advise to a sentence at the very beginning of the methods like "We train one model for a single well and evaluate this model on the same well and all other wells."

e.  Furthermore, why are models tested out-of-sample, meaning being trained on different wells than evaluated? Is there any idea behind it? Is the idea to learn a model that should be able to fill gaps in time series of any well at any location? If yes, you should probably re-think your entire training setup. If not, I don't see the need for this evaluation, since this is also not done for the ARIMA model.

4.  **LSTM vs ARIMA comparison:**
    a.  Why did you perform Hyperparameter search for the ARIMA method and not for the LSTM-based model?
    b.  Why is ARIMA not tested on wells that are not the training well, while the LSTM is?
    c.  P12 L6f: How was the best model decided? On training or test period? As of P13 Line 2f it seems like you picked the best model based on the test period results. If this is true, your results are biased and do not represent the true expected results of your methods. You either chose the best model by the training period, or better, have a third independent period (called validation split in machine learning) and pick your model based on the performance in this third data split, which is neither used for training nor for the final model evaluation.

5.  **SpC:** Later in the results section, you state that only SpC is of interest and no results for any of the other two variables are presented in this manuscript. This is totally okay, but my question is, why then do you model all three variables? Why not train the model using three inputs (temp, level and SpC) and predict only SpC?

6.  **P 11 L 20:** "We also observe that models with a daily 24-hour input window outperform other models with longer input windows as shown in Figure 6 (c)." This statement, figure 6(c) and thus your conclusion in the following sentences and the rest of the paper are misleading. It is completely logical, that the averaged MAPE over all settings for the input sequence length of 24h is the lowest, since this only includes models, where you predicted N=1h, 6h, 12h or 24h (as of table 1: N <= M). And as you have seen from all other experiments, filling only small gaps is easier for any model than filling large gaps. So the fact that the 24h input sequence has the smallest error is not due to the 24h input sequence, but due to the short output sequence for M=24h inputs. I would bet that if you train a model with input length 168h and only evaluate for 1h, 6h, 12h and 24h performance should be similar/better than for a 24h input window. It is probably better to remove figure 6(c) or rethink how you can fairly compare the average results over different input sequence length, since the different input sequence length also mean you evaluate them for different gap filling length.

Minor Comments:

- Title: At no point of this manuscript I see the term "spatio-temporal" justified. You are only filling temporal gaps in time gaps of a single well, without any spatial input information (e.g. the input features of the neighboring wells). So I would strongly advise to change all occurrences of the spatio-temporal framing to temporal only or clearly justify what in your work is the spatial component.
- P3 L4: Connor et al. (1994) is not the citation you should cite here for the RNN. Jordan (1986) would be more appropriate. Also the blog post from Olah (2015) is probably misleading here.
- P3 L11 Ma et al (2015) is definitely not the correct reference here and you should cite the original LSTM paper by Hochreiter & Schmidhuber (1997).
- P3 L11f. Beside text prediction, text translation, speech recognition and image captioning, LSTMs have also already been applied to earth science and even in hydrology, which might be also/more relevant to mention here.
- P 4 L 2 "select" -> "selected"
- P5 L15ff: In this entire discussion you mention "highly correlated" (L19), "lower correlations" (L20), "correlates well" (L20) and many more of these statements. Such statements usually required some quantitative measures (e.g. correlation coefficient). Otherwise, what is a high correlation and what low?
- P5 L27 here you state you only investigate 24-, 48-, 72-h gaps. In table 1 you have much longer periods listed as well as in figure 6, while then in figure 7 again only 24, 48, 72. This is a bit inconsistent.
- P5 L23 delete "clearly"
- P6 L3 What you mean is not a dropout layer, but the combination of a dense layer with additional dropout. Two consecutive dropout layer would mean simply applying dropout again to the result of your previous dropout output. Correctly it would state "followed by dense layer with dropout".
- P6 L3f: "This model architecture is generally described as a stacked LSTM model, given that the LSTM layers are "stacked" on top of each other." This is a tautology. Maybe simply remove this sentence or rephrase it.
- P7 L7 "select" -> "selected"
- P7 L17 This is not called a "sigmoid neural net layer". You could say "A linear layer with sigmoid activation function". At least call it "neural network" not "neural net".
- P7 L17: The pointwise multiplication is not part of the gate it-self, but how the gate is combined with the cell state.
- P7 L18ff and Fig5: all gates (f,i,o) and the cell and hidden state are vectors and should be written in lower, bold, italics letter and not capital letters.
- P7 L 23ff: "Finally, an output gate ($O_t$) decides what to output based on the input and previous memory state. The sigmoid layer of the output gate decides what parts of the memory state will be output..." The second sentence is basically a repetition of the first. Consider rephrasing.

- Table 1: Any particular reason, why you excluded 96h from the list of possible output window length, since otherwise possible input and output window length seems to be equal?
- P10 L 22 How are the terms (P, D, Q)m combined into equation 2. This needs more explanation.
- P11 L 19: In your setting, you always extrapolate. So this statement is not correct.
- P11 L 32: delete "very"
- LSTM results in general: It would be good to see only insample results at some point. How good does the LSTM perform for the same well it was trained for (as average over the 6 wells or for each well independently).
- Figure 7: Missing the information that results are only for SpC.
- The point above applies to the entire section here.
- P12 L15f: "It is noted that the optimal…" I would be cautious with such statements, unless you perform similar hyperparameter search for LSTMs as you did for ARIMA.
- P13 L 8f I do not see this in Figure 8. For me, there is no visible difference (or very hard to detect) in the Arima and LSTM error at any special frequencies. Maybe a better visualization or some quantitative measures would help.
- Figure 8. Why are the results now with the ARIMA model and 72 hour inputs and not 168 as in Figure 7?
- P14 L 1 Again, I don't see the LSTM outperforming ARIMA from Figure 8 column 3. Not sure how these (also column 4) help here. Maybe it is due to my lack of understanding of the data itself, but I think some quantitative measures are better than these figures. (e.g. a table with some metrics)
- "In general, both LSTM and ARIMA are effective at capturing longer term variability, but LSTM is more effective at capturing high-frequency fluctuations and nonlinearities in the dataset." I don't see any (quantitative) evidence for such a statement.

- Conclusion: As of everything written above, I think the conclusions need to be entirely rewritten, including possible new results of different model configurations etc. I will not go into more detail here, since I raised many concerns above, that apply similarly to the same statements in the conclusion (e.g. LSTM and ARIMA comparisons etc). Furthermore, you miss to say for which variable you are doing gap filling (SpC only)

References:

Hochreiter, S., & Schmidhuber, J. (1997). Long short-term memory. *Neural computation*, *9*(8), 1735-1780.

Jordan, M. I. (1986).  Attractor dynamics and parallelism in a connectionist sequential machine. In Proceedings of Ninth Annual Conference of the Cognitive Science Society, Amherst, pages 531–546

---

## Author Comment (AC1) · 4 Feb 2020

**Response to the short comments**

**Reviewer 1**

**Reviewer Comment 1.1** — This paper uses long short-term memory (LSTM) neural networks to fill in gaps in spatially distributed time-series data. The performance of the LSTM-based gapfilling method is compared to that of a traditional, popular gapfilling method: autoregressive integrated moving average (ARIMA). Overall, this paper is well written, structured and results seem sufficiently justified and useful. However, this paper is very technical and there is no physical insight beyond just feeding data into a standard code. I think this paper should be published as technical note (not as research article). Several aspects could be further improved in order to having it published in this journal.

**Response**: We thank the reviewer for overall positive assessment of our manuscript. While we emphazised the scientific importance of gap-filling the spatio-temporal data to capture and understand dynamic behaviors of complex systems, we also agree with the reviewer that our primary focus was to introduce a technical method that can fill data gaps by capturing the dynamic features using LSTM. We will resubmit this manuscript as a technical note as suggested.

**Reviewer Comment 1.2** — Would you guarantee the LSTM method in your paper can achieve the same excellent performance in other areas of the whole world? Is it possible that the good performance of the LSTM model is just applicable for the case given by the manuscript? The authors should include one more test for another area (maybe not in the text but in the supporting materials).

**Response**: There is unfortunately no guarantee for any model to have the same performance in other applications, which is the case for all data-driven and physics-based models. The performance of data-driven models can be optimized by iterating on various model configurations based on data types and characteristics of the relations between predictors and desired responses, as we have demonstrated in our study case when comparing ARIMA with LSTM. Sometimes the ARIMA works better, and sometimes the LSTM works better, and we believe that we have started down the path of predicting which model will work better for a given case. The LSTM model we adopted in our study has the same chain-like nature as other recurrent neural networks, meaning that this architecture lends itself well to sequences, so it will often be a useful (if not the best) approach for dynamic system behaviors [Karim et al., 2017] [Malhotra et al., 2015] [Kratzert et al., 2018] [Malhotra et al., 2016] [Wang et al., 2017] [Reddy and Prasad, 2018][Lipton et al., 2015]. The optimal model configuration and performance we can achieve would be case by case, and our focus of this technical note in to introduce a general method that can be broadly applied to other systems and be evaluated similarly. We will emphasize this aspect of transferability in our discussion and conclusion sections.

**Reviewer Comment 1.3** — LSTM model is only compared to ARIMA. Why not compare LSTM with other widely-used methods (such as Kriging interpolation and Gaussian process)? Furthermore, are the authors familiar with DIEOF (Data Interpolating Empirical Orthogonal Functions) which are proposed by Beckers and Rixen (2003)? I think that DIEFOF is powerful and

useful for filling temporal and spatial gaps in geophysical datasets. Maybe the authors can compare LSTM with DIEOF.

**Response**: There are many interpolation approaches that are commonly used, including the EOF-based approaches and kriging (based on Gaussian processes) that the reviewer mentions. We did discuss that kriging and the Gaussian processes are mostly used for spatial interpolation rather than spatio-temporal interpolation. We will add more discussions on the EOF related interpolation methods, such as least squares EOF (LSEOF), data interpolation EOF (DINEOF), and recursively subtracted EOF (REEOF), which are widely used to fill in missing data from geophysical fields such as clouds in sea surface temperature datasets or other satellite-based images with regular gridded domains. However, as discussed by Beckers and Rixen (2003), "For the method to have a chance to work, one needs, for each moment, at least a sufficiently large number of data points (otherwise one should drop the whole picture) and for each spatial point a sufficient amount of data in time (otherwise one should discard the point from the analysis)", which is a challenge for most of the monitoring networks that are sparsely distributed. Therefore, we would keep using ARIMA as the benchmark since it is the most commonly used method in time series analysis (a conclusion based on reviewing the hydrological literature) and gap filling. While we acknowledge that we did not (and could not) explore every possible interpolation method, we feel that by choosing such a representative approach, our study would not suffer greatly from loss of generality. We will also acknowledge this aspect in our discussion and conclusion sections.

**Reviewer Comment 1.4** — The present title "Using Deep Learning to Fill Spatio-Temporal Data Gaps in Hydrological Monitoring Networks" are inaccurate. I suggest new title like "Using Long Short Term Memory Neural Network Model to Fill Spatio-Temporal Data Gaps in Hydrological Monitoring Networks"

**Response**: We agree that specifying LSTM in the title could be more accurate, so we will change the title as suggested.

**References**

Fazle Karim, Somshubra Majumdar, Houshang Darabi, and Shun Chen. Lstm fully convolutional networks for time series classification. *IEEE access*, 6:1662–1669, 2017.

F. Kratzert, D. Klotz, C. Brenner, K. Schulz, and M. Herrnegger. Rainfall–runoff modelling using long short-term memory (lstm) networks. *Hydrology and Earth System Sciences*, 22(11):6005–6022, 2018. doi: 10.5194/hess-22-6005-2018. URL https://www.hydrol-earth-syst-sci.net/22/6005/2018/.

Zachary C Lipton, David C Kale, and Randall C Wetzel. Phenotyping of clinical time series with lstm recurrent neural networks. *arXiv preprint arXiv:1510.07641*, 2015.

Pankaj Malhotra, Lovekesh Vig, Gautam Shroff, and Puneet Agarwal. Long short term memory networks for anomaly detection in time series. In *Proceedings*, volume 89. Presses universitaires de Louvain, 2015.

Pankaj Malhotra, Anusha Ramakrishnan, Gaurangi Anand, Lovekesh Vig, Puneet Agarwal, and Gautam Shroff. Lstm-based encoder-decoder for multi-sensor anomaly detection. *arXiv preprint arXiv:1607.00148*, 2016.

D Sushma Reddy and P Rama Chandra Prasad. Prediction of vegetation dynamics using ndvi time series data and lstm. *Modeling Earth Systems and Environment*, 4(1):409–419, 2018.

Qianlong Wang, Yifan Guo, Lixing Yu, and Pan Li. Earthquake prediction based on spatio-temporal data mining: an lstm network approach. *IEEE Transactions on Emerging Topics in Computing*, 2017.

---

## Author Comment (AC2) · 4 Feb 2020

**Response to referee comments**

**Reviewer 2**

**General Remarks:**The paper presents an interesting use of deep learning with LSTM Networks for infilling groundwater data. The article is timely and tries to make a comprehensive description and explanation of how the Deep learning technique is implemented using statistical and machine learning techniques. The paper is a welcome contribution to the field of groundwater and hydrological earth sciences. However, I cannot recommend publication in the present form due to the comments and questions raised. The paper needs major revision.

 **Response**:  Thanks for the reviewer's careful consideration of our manuscript and positive assessment. We address individual comment below to improve our manuscript for possible publication.

**Reviewer   Comment   2.1**  —  The paper states that long-term spatiotemporal changes in subsurface hydrological flow is usually quantified using a network of wells. However this paper does not deal with the long-term trend or analysis. Hourly data is hardly interpreted or used for the long term. Hourly information for sure contains noise that would be advisable to remove for the long term analysis.

 **Response**:  There seems to be some confusion between long-term changes and long-term trends. We will rephrase to clarify that monitoring networks capture the dynamic system behaviors over long time windows, which allows the discovery of signals over a spectrum of time scales as revealed by the spectral analysis we presented in the manuscript. While long-term trends, e.g., low-frequency variations, are usually smoother and could be captured by existing time series analysis method like ARIMA (see our comparison analyses between LSTM- and ARIMA-based gap filling results), our focus is on capturing high-frequency dynamics that are important signatures for understanding managed systems. We will clarify those points in our objectives and reiterate in conclusions to avoid confusion.

**Reviewer   Comment   2.2**  —  Observations are mentioned to be spatially sparse, and temporal gaps exist. Many papers have solved the same type of problem, without using the term spatiotemporal. Almost every course in hydrology deals in one chapter with the issue of using spatial correlation and temporal correlation to fill in data. So in this respect, the authors are invited to clearly indicate what innovation is brought by this work to spatiotemporal analysis.

 **Response**:  We agree with the reviewer that a lot has been done in hydrology for spatial and temporal analyses. However, there have been very few studies that address spatial and temporal correlations simultaneously due to the difficulty in parameterizing the spatial and temporal correlations all together. Deep neural networks provide an alternative way to represent such correlations without assuming the explicit form of correlations a priori, which is the innovation our work originally aimed to bring and demonstrate. However, we found there are multiple steps towards accomplishing that goal as it involves merging two types of deep neural networks to represent both the spatial and temporal correlations and evaluate various configurations thoroughly. Related to the comments Reviewer #1 provided, we will be focusing on the temporal component of the data and use LSTM for capturing multi-scale signatures, which we feel is appropriate scope for a technical note. One unique advantage of using LSTM to represent temporal correlations is that we do not pre-assume a correlation form. We will make this clear

in the discussion. We will also add spatio-temporal analyses as our next step to achieve the ultimate goal.

**Reviewer Comment 2.3** — Following point two, it is known that in most of the cases, aquifers with little or no human intervention have low variability. Conventional guidelines and measures in hydrogeological science are typically based on monthly data.

**Response**: It is true that some aquifers with no or little human intervention have low variability, for which monthly data could be sufficient to understand the system behavior. However, anthropogenic activities, in particular, dam operations, have increasingly impacted the river and aquifer systems by altering the exchange patterns between river water and groundwater, and the associated thermal and biogeochemical processes [Song et al., 2018, Shuai et al., 2019, Zachara et al., 2020]. Due to significant, high-frequency (hourly) stage variations caused by dam regulations to meet power generation needs, it is insufficient to use monthly data to understand such systems as have been demonstrated in numerous studies performed at our study site. Our study site is representative of many dam-regulated gravel-bed rivers across the world. Therefore, our study will have broader impacts to many other systems. We will make this point clear in our introduction and conclusion sections.

**Reviewer Comment 2.4** — In the present paper the idea of nonlinear dynamics is mentioned almost everywhere in the introduction and justification of the work. This is somewhat surprising and needs better justification, since groundwater dynamics, in many cases, can be represented with linear models. As it is concluded in this paper results, ARIMA can approximate the system quite well.

**Response**: This comment is related to the earlier comment 2.3 as high-frequency dynamics lead to higher level of nonlinearity in system responses, especially in the specific conductance that is a result of mixing of water from various sources. We have shown that a linear model like ARIMA was not able to capture such nonlinearity, while LSTM could. We will explain this point in the revised manuscript. Please also refer to our response to comment 2.3 for the importance of capturing high-frequency dynamics for many dam-regulated systems, which will also be better articulated in the revised manuscript.

**Reviewer Comment 2.5** — The particular case study presented here shows a relative complex dynamic nature indeed, but it seems it is due to human intervention (however I could be wrong). Can you comment on this and the uncertainties associated?

**Response**: The reviewer is partially right that human intervention contributes to the complexity of system behavior by creating high-frequency flow dynamics. However, the full complexity is a result of interactions between such human-induced variations and the natural heterogeneity of aquifer physical properties [Zachara et al., 2020]. There is significant uncertainty associated with aquifer physical heterogeneity at our study site as revealed by previous studies. We will add these additional discussion about the system complexity in the revised manuscript.

a The human intervention might affect your calculation and therefore, extractions might not be following a random but more human induced behaviour. So data understanding or replicability used in one year might not be the same in another. It would be advisable first to check how much and when extraction took place. Is this data filled in for a long term analysis, or short-term? This question arises since the hourly step is used.

**Response**: Please refer to our response to Comment 2.1 for explaining our use of long-term data versus long-term trend analyses. The reviewer is right that high-frequency flow variations are mainly caused by the dam operations while the seasonal and interannual variabilities are controlled by climatic forcing like precipitation and snow pack in the headwater systems. We will clarify the drivers of the high-frequency and low-frequency variations in the revision. We used multiple years of training data from dry, normal and wet hydrologic years to capture potential operational patterns associated with various conditions. LSTM units include a 'memory cell' that can maintain information in memory for long periods of time. We will look into the memory cells for more explanations.

b If indeed human intervention influence the dynamics of the groundwater system, the logical approach would be to find a variable to represent direct or indirect measurement of extractions.

**Response**: Thanks for the suggestion. Please refer to our response to your earlier comment. We will look into both memory cells and state cells to illustrate how and where the extraction occurs. We will expand our analyses accordingly.

c It is suggested to read the paper by Amaranto et al. (2018) "Semi-seasonal groundwater forecast using multiple data-driven models in an irrigated cropland". J Hydroinformatics, 20 (6): 1227–1246. DOI: `https://doi.org/10.2166/hydro.2018.002` and - Amaranto et al. ( 2019). A spatially enhanced data driven multimodel to improve semiseasonal groundwater forecasts in the High Plains aquifer, USA. Water Resources Research, 55, 5941– 5961. `https://doi.org/10.1029/2018WR024301`

**Response**: Thank you for the paper suggestions. Both of the papers listed above use data-driven approaches to improve groundwater forecasts. The MuMoC framework select neighboring wells to assist groundwater predictions is of our interest. Although the authors used data with coarser temporal resolution (daily or monthly) to make monthly predictions, which is different from our purpose of filling short gaps (up to 3 days) for capturing high-frequency dynamics, the idea of using information from neighboring wells applies to our case. We will include an additional architecture to enable multi-well setup. We will review and discuss these two papers in our revision.

**Reviewer Comment 2.6** — The regional aquifer and geology might play a more significant role in the study, since not only the river but the size and other interventions and hydrometeorological recharges might be correlated.

**Response**: We agree that the regional aquifer and geology play an important role as shown in previous studies performed by our colleagues [Chen et al., 2012, 2013, Zachara et al., 2020]. The aquifer is composed of two distinct geologic formations, a highly permeable formation (Hanford formation, consisting of coarse gravelly sand and sandy gravel) underlain by a much less permeable formation (the Ringold Formation, consisting of silt and fine sand). The dominant hydrogeologic features of the aquifer are defined by the interface between the Hanford and Ringold formations and the heterogeneity within the Hanford formation. The understanding we developed from these earlier studies is that the physical heterogeneity contributes to the different response behaviours at different locations while the river stage

dynamics lead to multi-frequency dynamics in those responses. We will add more information to our system description to help readers understand.

**Reviewer Comment 2.7** — The stations are so close, and the hourly variation appears to be periodic with an amplitude of 4 or 5cm, according to Figure 1 (and on other graphs). It is intriguing, the question I would have is what happens every hour? and if this hourly variation is noise on the measurement device or data? What is the precision of the measurement device? What is the volume of water extracted to reach the variation of 1 cm? Where the recharge water comes from(has this been studied in the past)? Is this 5 cm recharge volume feasible in one hour? Could be the water from the river affecting your measurements (interflow)? It is advisable to present the time series of the river flow. It would be also useful to have a few hydrological balances (note that this is a hydrological journal). The problematic still can be questioned due to its apparent complex dynamics with the river and human intervention (not a typical, natural aquifer).

**Response**: The reviewer is right that the water table elevation difference is small due to the close distance between wells and the highly permeable aquifer material (hydraulic conductivity in the range of 4000-7000 m/d). The rapid change in water table is at first due to pressure wave propagation from the river stage variation, and the recharge water comes from the river or displacement of groundwater from other parts of the aquifer depending on flow directions and locations of interest. Our pressure transducers (Stainless-Steel CS451 by the Campbell Scientific) is accurate enough to capture cm-scale variations. With a full scale (FS) of 102m,and resolution of 0.0035% FS which means with a full scale of 102m, the sensor will be able to detect a 0.375cm change in pressure. The measurement range of the pressure transducer set up in our study site is 0 to 10.2m, the standard accuracy is $\pm 0.1\%$ [Scientific] which leads the accuracy to about 1cm. In this case, the pressure changes with an amplitude of 4 5cm are the actual measurements and our consistent 10-year data has proven this point. In our revision, we will refer to numerous hydrologic modeling studies performed at the site to help readers understand the flow conditions and where the water comes from.

**Reviewer Comment 2.8** — On the model setup, Please explain why you use Mx128.

**Response**: We use 128 (i.e. 128 units for each LSTM layer) because this number of units showed better performance after we experimented with different model architectures with different number of units. We will add this rationale to the manuscript.

**Reviewer Comment 2.9** —
    Page 7, line 10, mentions the supplemental material, but I cannot find it in the paper.

**Response**: The supplemental materials can be found using this link
    https://www.hydrol-earth-syst-sci-discuss.net/hess-2019-196/hess-2019-196-supplement.pdf

**Reviewer Comment 2.10** — Important: choice of (a very complex model) LSTM has to be justified, since it seems AR-type models is enough. Frankly, I don't see the need for complex models like LSTM, but if you have arguments to defend your position, please present them to convince the readers.

**Response**: We are interested in using an LSTM for this problem because LSTMs have had success in predicting values in time-series data without assuming explicit temporal dependence forms . We cite several examples of this on P3 L12. We wanted to explore whether an LSTM would provide improved performance over traditional methods (i.e. ARIMA). As far as we are aware of, there have not been other applications of LSTMs to groundwater well data. Related to our response to other relevant comments, we will make sure we clearly state the advantages of using LSTM in our revision.

Furthermore, there have been several recent applications of LSTMs to hydrology and earth sciences. Kratzert et al. [2018] used LSTMs to predict rainfall-runoff from meteorological observations. Zhang et al. [2018] used LSTMs for predicting and monitoring sewer overflow. Additionally, Fang et al. [2017] used LSTMs to predict soil moisture with high fidelity. We will update our paper (and references) to further explain our interest in applying LSTMs to this hydrological domain.

Old section (P3 L13): This makes LSTMs well suited for the problem at hand, particularly for data where multiple timescales of variability can affect responses [Liu et al., 2016, Song et al., 2017].

New section (P3 L13): This makes LSTMs well suited for the problem at hand, particularly for data where multiple timescales of variability can affect responses [Liu et al., 2016, Song et al., 2017]. Furthermore, LSTMs have been applied to earth science and hydrology domains. Kratzert et al. [2018] used LSTMs to predict rainfall-runoff from meteorological observations. Zhang et al. [2018] used LSTMs for predicting and monitoring sewer overflow. Additionally, Fang et al. [2017] used LSTMs to predict soil moisture with high fidelity.

**Reviewer Comment 2.11** — On page 14, it states that other configurations of LSTM can be further explored; however, it is not clear why this was not done before. Not sure why the selected configuration was just tried to see if it works or not, without any analysis what is the best structure. This relates to comment 8 and 9.

**Response**: We acknowledge that we did not explain this point as well as we could have. We performed hyperparameter searches on: Number of LSTM layers, number of units per LSTM layer, number (and size of) dense layers, activation functions. This was performed for data on one well (399-1-1) with a smaller subset of input and output prediction windows, experimenting with different architecture configurations. However, this was not an exhaustive search of all possible configurations. We will add some additional description to this effect to the manuscript.

**Reviewer Comment 2.12** — I am a bit in confusion how to interpret the statements made in conclusion. The ARIMA is not suited or less suited for filling high frequency (hourly, or short gaps) and more suitable for a long term period (24, 48 and 74 hours). It is suggested we need deep learning for filling high-frequency gaps (of one hour)?. Maybe is good to elaborate on the simplicity of what this translates to, I am not sure if the meaning is right.

**Response**: We acknowledge a potential source of confusion in terms of high-frequency fluctuations in temporal scale and the dominant frequency in the wavelet transform. In our study, we find that the ARIMA method would work well if the dominant frequencies in the wavelet transform are from seasonal cycles, whereas an LSTM could work better if the dynamics are dominated by daily and subdaily (high-frequency) fluctuations. We will better state this in the paper to avoid further confusion.

**Reviewer Comment 2.13** — Not sure if there is an idea of how high is the overall error; in the figure 8, with well 1-15 it seems almost perfect representation (zero error in the validation data for many points). Also in the same well, it appears like high negative correlation up to 128 hours

**Response**:  Well 1-15 has a different pattern than the rest of wells in that most of its higher intensities fall under lower frequency and its SpC correlates well with river stage towards longer periods and less persistent in time. Both ARIMA and LSTM have highly accurate predictions for well 1-15 (i.e., relative errors are about 1%), which is consistent with our conclusions that smooth changes in the observations are easier to be captured. In the wavelet spectral plot in column 3 of figure 8, where log10 of Wavelet Power Spectrum (WPS) is shown, the blue color under 128 hours still represent weak positive values.

**References**

Xingyuan Chen, Haruko Murakami, Melanie S. Hahn, Glenn E. Hammond, Mark L. Rockhold, John M. Zachara, and Yoram Rubin. Three-dimensional bayesian geostatistical aquifer characterization at the hanford 300 area using tracer test data. *Water Resources Research*, 48(6), 2012. doi: 10.1029/2011WR010675. URL `https://agupubs.onlinelibrary.wiley.com/doi/abs/10.1029/2011WR010675`.

Xingyuan Chen, Glenn E. Hammond, Chris J. Murray, Mark L. Rockhold, Vince R. Vermeul, and John M. Zachara. Application of ensemble-based data assimilation techniques for aquifer characterization using tracer data at hanford 300 area. *Water Resources Research*, 49(10):7064–7076, 2013. doi: 10.1002/2012WR013285. URL `https://agupubs.onlinelibrary.wiley.com/doi/abs/10.1002/2012WR013285`.

Kuai Fang, Chaopeng Shen, Daniel Kifer, and Xiao Yang. Prolongation of smap to spatiotemporally seamless coverage of continental u.s. using a deep learning neural network. *Geophysical Research Letters*, 44(21):11,030–11,039, 2017. doi: 10.1002/2017GL075619. URL `https://agupubs.onlinelibrary.wiley.com/doi/abs/10.1002/2017GL075619`.

F. Kratzert, D. Klotz, C. Brenner, K. Schulz, and M. Herrnegger. Rainfall–runoff modelling using long short-term memory (lstm) networks. *Hydrology and Earth System Sciences*, 22(11):6005–6022, 2018. doi: 10.5194/hess-22-6005-2018. URL `https://www.hydrol-earth-syst-sci.net/22/6005/2018/`.

Jun Liu, Amir Shahroudy, Dong Xu, and Gang Wang. Spatio-temporal lstm with trust gates for 3d human action recognition. In *European Conference on Computer Vision*, pages 816–833. Springer, 2016.

Campbell Scientific. Cs451:stainless-steel pressure transducer. URL `https://www.campbellsci.com/cs451`.

Pin Shuai, Xingyuan Chen, Xuehang Song, Glenn E. Hammond, John Zachara, Patrick Royer, Huiying Ren, William A. Perkins, Marshall C. Richmond, and Maoyi Huang. Dam operations and subsurface hydrogeology control dynamics of hydrologic exchange flows in a regulated river reach. *Water Resources Research*, 55(4):2593–2612, 2019. doi: 10.1029/2018WR024193. URL `https://agupubs.onlinelibrary.wiley.com/doi/abs/10.1029/2018WR024193`.

Sijie Song, Cuiling Lan, Junliang Xing, Wenjun Zeng, and Jiaying Liu. An end-to-end spatiotemporal attention model for human action recognition from skeleton data. In *AAAI*, volume 1, pages 4263–4270, 2017.

Xuehang Song, Xingyuan Chen, James Stegen, Glenn Hammond, Hyun-Seob Song, Heng Dai, Emily Graham, and John M. Zachara. Drought conditions maximize the impact of high-frequency flow variations on thermal regimes and biogeochemical function in the hyporheic zone. *Water Resources Research*, 54(10):7361–7382, 2018. doi: 10.1029/2018WR022586. URL `https://agupubs.onlinelibrary.wiley.com/doi/abs/10.1029/2018WR022586`.

John M. Zachara, Xingyuan Chen, Xuehang Song, Pin Shuai, Chris Murray, and C. Tom Resch. Kilometer-scale hydrologic exchange flows in a gravel-bed river corridor and their implications to solute migration. *Water Resources Research*, n/a(n/a):e2019WR025258, 2020. doi: 10.1029/2019WR025258. URL `https://agupubs.onlinelibrary.wiley.com/doi/abs/10.1029/2019WR025258`. e2019WR025258 2019WR025258.

Duo Zhang, Geir Lindholm, and Harsha Ratnaweera. Use long short-term memory to enhance internet of things for combined sewer overflow monitoring. *Journal of Hydrology*, 556:409 – 418, 2018. ISSN 0022-1694. doi: https://doi.org/10.1016/j.jhydrol.2017.11.018. URL `http://www.sciencedirect.com/science/article/pii/S0022169417307722`.

---

## Author Comment (AC3) · 4 Feb 2020

**Response to referee comments**

**Reviewer 3**

**General note:** I was asked by the editor to review this manuscript, although groundwater hydrology is not my area of expertise. However, machine learning is and therefore most of my review will be around the methods and experimental setting used in this manuscript. This manuscript presents an approach for filling gaps in time series of ground water well measurements. Specifically, the authors compare two different methods (LSTM-based and ARIMA) for different gap lengths for six different wells. Although I generally welcome publications that try to make use of deep learning based methods for various applications in earth science, I see various major concerns with the manuscript at hand. Overall, it seems like the authors are not too familiar with the methods they apply (especially the LSTM-based model) and many decisions made seem questionable and lack any justification or explanation. Because of these concerns, I'm not sure if I can recommend this manuscript for publication. If it should be published at all, major revisions are required.

**Response**: Thank you for reviewing and providing the summary. We agree with the reviewer that it is important to demonstrate that we know what we are doing, and we appreciate the reviewer's careful attention to make sure we did our due diligence. The comments are addressed point-by-point.

**Reviewer Comment 3.1** — Model architecture: Coming from the field of machine learning, I was surprised by the creativity of the authors in finding their model architecture. To be honest, I have never seen such a combination of LSTM layers, dense layers and convolutional layers for a time series task and I wonder if the authors know what they are doing. Here is a list of sub points to this major comment:

  a First: Did you perform any hyperparameter search at all to find this architecture? If yes, please give details on the model configurations (in terms of layers) you tried, if not, why not? To propose such an exotic architecture, it is required to see quantitative evidence that this is required and not a much simple LSTM-based model would be better (e.g. single LSTM layer with single dense + dropout layer)

  **Response**: We performed hyperparameter searches on: Number of LSTM layers, number of units per LSTM layer, number (and size of) dense layers, activation functions. This was performed for data on one well (399-1-1) with a smaller subset of input and output prediction windows, experimenting with different architecture configurations.

  b Why do you stack 3 LSTM layers? In theory, a single LSTM layer is turing-complete. Besides probably natural language processing, where the training data consists of million/billion of samples, there is almost always no need to use more than a single LSTM layer. Additionally, since you have very limited training data (2 years of hourly data are just 17520 data points), the size of your LSTMs seem to be exorbitantly large. Especially with 3 LSTM layers.

  **Response**: In response to using multiple LSTM layers, there has been research looking at the benefits of using multiple RNNs/LSTMs in a model in comparison to a single RNN/LSTM [Graves et al., 2013, Pascanu et al., 2013]. Likewise, there has been work in using multiple LSTMs for

action recognition [Zhu et al., 2016], traffic prediction [Du et al., 2017], and vulnerable road users location predictions [Saleh et al., 2017]. As such, we wanted to investigate the potential benefits of using multiple LSTM layers in our problem domain. We will add a paragraph to our manuscript discussing previous uses of stacked LSTMs and some comparisons of single versus multiple in different domains to give context on why we are interested in this model architecture and update our references with the cited articles.

Old paragraph: We have tested the effects of training data on model performance using 2,4 and 6 years data, and found that 4 years of training data led to similar performance to 6 years of training data. Therefore, we are confident that we have enough data to support the selected architecture. We will provide results from a single LSTM layer in supplemental material for comparison.

New paragraph: There has been research looking at the benefits of using multiple RNNs/LSTMs in a model in comparison to a single RNN/LSTM [Graves et al., 2013, Pascanu et al., 2013]. Likewise, there has been work in using multiple LSTMs for action recognition [Zhu et al., 2016], traffic prediction [Du et al., 2017], and vulnerable road users location predictions [Saleh et al., 2017]. As such, we investigate the potential benefits of using multiple LSTM layers in the problem domain of hydrological networks.

c Why the combination of convolutional layers and dense layers after the LSTM? Probably the standard is to have a single dense layer that uses the hidden output of the LSTM to map to your desired target shape. Why do you think so much complexity is needed after the LSTM, since the LSTM should capture the complex temporal dependencies already?

   **Response**: As stated in our response in 3.1a, we performed some hyperparameter searches, experimenting with different architecture configurations which led us to use convolutional and dense layers. We acknowledge that more information on the extensive analysis and experimentation we have performed would be useful in further justifying the choice of model architecture, so we will provide those details in supplemental material.

d Why do you have the convolutional layer at all? If I understand your setting correctly, the convolutional layer can again look at the entire sequence (M x 64, with M the input sequence length). Why is this necessary? The task of the LSTM is to summarize the input sequence and store all the information necessary for predicting the M+1 time step (first step of your N time step long gap) in it's cell state. e. Another point related to the convolutional layer. I see that the filter size was solely chosen to be able to map from a sequence length of M to an output of N (filter size M-N+1). However, are the authors aware of what that means? For example, for predicting the first of the N time steps, the convolutional filter will only look at the first M-N+1 input sequence elements, effectively ignoring what has happened at the time steps preceding the current time step. Why do you want this? It makes absolutely no sense to not include the most informative information (the previous time steps) necessary to predict the next time step.

   **Response**: Yes, the intent was to map from a sequence length of M to an output of N. The reviewer is correct that the convolutional filter does limit the model in ignoring the most recent time steps. As stated in our response to comment 3.1c, we felt our exploration of architectures, including using convolutional layers, resulted in a good architecture. Furthermore, the time steps

immediately preceding the current time are not necessarily the most informative information in the presence of dynamical behavior. However, in response to the reviewers concern, we will train the models without the convolutional layer (filter size M-N+1) and compare the results against the original architecture. We will ensure all of the input sequence will be used in predicting each future time step. A proposed architecture is to first change the last LSTM layer to return the last hidden states (i.e, output size 128). This modifies the M x 128 dense layer to be a one-dimensional 128 size layer. Then, we will remove the M x 64 dense layer and the convolutional layer (now defunct) and change the final dense layer to be size N, whose output will be the N predicted SpC values. The architecture can be modified to return the predicted N values for all three measurements by using three independent dense layers of size N instead of one, which will be concatenated at the end into a N x 3 output. Below are two images of the proposed architectures, one predicting only SpC and one predicting all three measurements.

[Figure]

Figure 1: Modified model architecture without convolutional layer, only predicting SpC measurement

[Figure]

Figure 2: Modified model architecture without convolutional layer, predicting temperature, SpC, and water level measurements

**Reviewer Comment 3.2** — Related work: Since (correct me if I'm wrong) this is not a forecast task, but just filling gaps in historic data records, I wonder if the authors have done some research, which approaches are currently used in the field of deep learning, before proposing their own method. E.g. for gap filling in historic time series, Bi-directional LSTMs are commonly used over normal LSTMs, since they do two sided gap filling (closer to interpolation), compared to the standard LSTM, which basically extrapolates into the future. I would also advise to add some related work section of LSTM-based gap filling into the introduction.

**Response**: The reviewer is correct that the goal is to test gap filling in historical records. For our work, we treat the gaps as a forecasting problem which means we use the historical data as input to predict the values during gap period. Bi-directional architectures have been used for gap-filling. A bi-directional LSTM is another model type applicable to the work, but we felt that a more thorough explanation of an LSTM would be prudent to explore before jumping to that architecture. We will add a paragraph describing deep learning techniques that have been applied to gap filling, including LSTMs.

New Description: There have been several applications of deep learning techniques to fill gaps in time-series data. Ustoorikar and Deo [2008] used genetic programming to fill in gaps of ocean wave heights. Khalil et al. [2001] estimated missing values in monthly streamflow time-series data using neural networks. Berglund et al. [2015] used RNNs and bi-directional RNNs to infer missing values in high-dimensional binary time-series data.

**Reviewer Comment 3.3** — Training setup: There are various points around the model training setup that I see problematic. Some of them might overlap to other points mentioned above or below.

a Input features for any neural network should be normalized to zero mean, unit variance and not to the range of 0 to 1. This will basically bias your network during the start of the training in a wrong way. Maybe as some intuition: Most (all?) activation functions are centered around zero, e.g. the sigmoid function in all gates of the LSTM. With randomly initialized weights (which are normally initialized around 0), using your normalization would bias the entire network to always have pre-activations of larger than zero, and thus sigmoid values close to one. However, what you want is in expectancy to be undecided in the beginning (pre-activation of 0, equals to sigmoid of 0.5). Long story short, you should re-run all experiments with different normalizations, at least for the LSTM.

   **Response**: Thank you for the comment. We will re-run the experiments using the zero mean, unit variance normalization technique and compare those results against the original normalization.

b Results of neural networks are generally affected by some stochasticity, because of the random weight initialization and the randomness of stochastic gradient descent. This requires almost always to train multiple models for the exact same setting with different random initialization (seeds) and to report the average model performance and variations across those repetitions. Otherwise, results might not be reproducible, since you might only be lucky (or unlucky) with your single initialization.

   **Response**: Thank you for the comment. We will re-run the experiments with different initialization seeds.

c In general, you have very few data points for such a large deep learning model, as already stated above. You could either think of ways, how to combine the data of all wells in a single model, or reduce your model size drastically, which is what I would propose here.

   **Response**: We have the ability to combine the data of input from neighboring wells (up to five more) for the large deep learning model, which for 4 years would be approximately 210240 data points. Similar to the reviewer's comments for 3.1a, we can also perform more extensive experimentation on a smaller model (single LSTM layer with single dense + dropout layer).

d I found it very hard to follow your training and testing setup, until late in the paper. E.g. around the number of possible model configurations, and total train-test combinations. I would advise to a sentence at the very beginning of the methods like "We train one model for a single well and evaluate this model on the same well and all other wells."

**Response**: Thank you for the great suggestion. We will add a sentence to the beginning to further clarify our training and testing setup.

e Furthermore, why are models tested out-of-sample, meaning being trained on different wells than evaluated? Is there any idea behind it? Is the idea to learn a model that should be able to fill gaps in time series of any well at any location? If yes, you should probably re-think your entire training setup. If not, I don't see the need for this evaluation, since this is also not done for the ARIMA model.

**Response**: The intent on evaluating models on wells different from the training well was to analyze how well the model does on data from a well it has not seen. However, as noted by the reviewer, this evaluation was not done for the ARIMA model. As such, we will remove this evaluation in order to make the paper more straight forward and less confusing in its comparison of LSTMs and ARIMA. Furthermore, we will redo figure 6 without the additional analysis and remove figure 6f.

**Reviewer Comment 3.4** — LSTM vs ARIMA comparison:

a Why did you perform Hyperparameter search for the ARIMA method and not for the LSTM-based model?

**Response**: A hyperparameter search for the ARIMA approach is performed by using the "auto.arima" function in R automatically. We also performed a hyperparameter search on the architecture of the LSTM models. This includes: the number of LSTM layers, the number of units per LSTM layer, andthe number (and size of) dense layers, and activation functions.

b Why is ARIMA not tested on wells that are not the training well, while the LSTM is?

**Response**: ARIMA is not tested on other wells since the ARIMA model is built dynamically based on the 168 historical hours for each well. We believe the information carried by the ARIMA model is not enough to train other well. Also according to the comment 3e, we will remove the model evaluation on testing (which includes figure 6(f) on the non-training wells to reduce the confusion.

c P12 L6f: How was the best model decided? On training or test period? As of P13 Line 2f it seems like you picked the best model based on the test period results. If this is true, your results are biased and do not represent the true expected results of your methods. You either chose the best model by the training period, or better, have a third independent period (called validation split in machine learning) and pick your model based on the performance in this third data split, which is neither used for training nor for the final model evaluation.

**Response**: The best model for each well was decided on the test period (data from 2011). We have two more years of data (2017-2018) for the wells. We will update the analysis to use data from 2011 as the validation split and the data from 2017-2018 for the final model evaluation.

**Reviewer Comment 3.5** — SpC: Later in the results section, you state that only SpC is of interest and no results for any of the other two variables are presented in this manuscript. This is totally okay, but my question is, why then do you model all three variables? Why not train the model using three inputs (temp, level and SpC) and predict only SpC?

**Response**: Other similar analyses for groundwater table and temperature are done but not shown here because SpC is our primary interest for this study. The reviewer is right that we don't have to predict all three variables. Although retaining all three variables in the model affords the flexibility to fill gaps in the other two variables, it would be too much for this paper to cover. Therefore, we will follow your suggestion to keep only SpC as our predicted output variable.

**Reviewer Comment 3.6** — P 11 L 20: "We also observe that models with a daily 24-hour input window outperform other models with longer input windows as shown in Figure 6 (c)." This statement, figure 6(c) and thus your conclusion in the following sentences and the rest of the paper are misleading. It is completely logical, that the averaged MAPE over all settings for the input sequence length of 24h is the lowest, since this only includes models, where you predicted N=1h, 6h, 12h or 24h (as of table 1: N ¡= M). And as you have seen from all other experiments, filling only small gaps is easier for any model than filling large gaps. So the fact that the 24h input sequence has the smallest error is not due to the 24h input sequence, but due to the short output sequence for M=24h inputs. I would bet that if you train a model with input length 168h and only evaluate for 1h, 6h, 12h and 24h performance should be similar/better than for a 24h input window. It is probably better to remove figure 6(c) or rethink how you can fairly compare the average results over different input sequence length, since the different input sequence length also mean you evaluate them for different gap filling length.

**Response**: Thank you for the feedback. We re-did the analysis done in figure 6c, but limiting the predict output to 1h, 6h, 12h, and 24h. As the reviewer noted, the averaged MAPE for the model with input length of 168h, 7.330167, is similar to that of the 24h input length models (6.394719). This provides a more fair analysis on the model performance based on input length, given the number of model performances averaged per input window is the same. We will redo figure 6(c) to only include models whose predicted outputs are 1h, 6h, 12h, and 24h. Additionally, we will update our analysis of figure 6(c). The updated figure also includes the removal of figure 6(f), as stated in our response to comment 3.4b. The updated figure and caption is shown below.

Old sentence: We also observe that models with a daily 24-hour input window outperform other models with longer input windows as shown in Figure 6 (c). This likely results from an optimal number of memory units for capturing daily and subdaily memories.

New sentence: We compare model performance by input window size, but limiting to models whose predict output is 1h, 6h, 12h, or 24h. As show in Figure 6 (c), models with a daily 24-hour input window have the best performance. However, there is a large amount of overlap of the 95% confidence interval for each input window.

**Minor Comments:**

[Figure]

Figure 6: Gap filling performance for SpC evaluated against multiple model configuration parameters (a-d) or grouped by training wells (e). (a) average MAPE vs. number of years of training data; (b) average MAPE vs. gap lengths; (c) average MAPE vs. input window size M; (d) average MAPE vs. output window size N, but only for models whose predicted output is 1h, 6h, 12h, and 24h; (e) average MAPE aggregated by wells used to train the models. 95% confidence intervals of the averaged MAPE value are shown in shaded area in plots (a) -(d) and as the error bars in (e)

**Reviewer Comment 3.7** — Title: At no point of this manuscript I see the term "spatio-temporal" justified. You are only filling temporal gaps in time gaps of a single well, without any spatial input information (e.g. the input features of the neighboring wells). So I would strongly advise to change all occurrences of the spatio-temporal framing to temporal only or clearly justify what in your work is the spatial component.

**Response**: We agree with reviewer and have changed all instances.

**Reviewer Comment 3.8** — P3 L4: Connor et al. (1994) is not the citation you should cite here for the RNN. Jordan (1986) would be more appropriate. Also the blog post from Olah (2015) is probably misleading here.

**Response**: We thank the reviewer for the comment. We will remove the citation of Olah (2015) and update our citation for the RNN to Jordan (1986).

**Reviewer Comment 3.9** — P3 L11 Ma et al (2015) is definitely not the correct reference here and you should cite the original LSTM paper by Hochreiter & Schmidhuber (1997).

**Response**: We thank the reviewer for the comment. We will update the citation and references accordingly.

**Reviewer Comment 3.10** — P3 L11f. Beside text prediction, text translation, speech recognition and image captioning, LSTMs have also already been applied to earth science and even in hydrology, which might be also/more relevant to mention here.

**Response**: On P2 Line-35, we cite papers using DL in geophysical domain. However, as implied by the reviewer, we will add a brief description of LSTMs applied to earth science and hydrology (this is the same update in response to reviewer comment 2.10)

Old section (P3 L13): This makes LSTMs well suited for the problem at hand, particularly for data where multiple timescales of variability can affect responses [Liu et al., 2016, Song et al., 2017].

New section (P3 L13): This makes LSTMs well suited for the problem at hand, particularly for data where multiple timescales of variability can affect responses [Liu et al., 2016, Song et al., 2017]. Furthermore, LSTMs have been applied to earth science and hydrology domains. Kratzert et al. [2018] used LSTMs to predict rainfall-runoff from meteorological observations. Zhang et al. [2018] used LSTMs for predicting and monitoring sewer overflow. Additionally, Fang et al. [2017] used LSTMs to predict soil moisture with high fidelity.

**Reviewer Comment 3.11** — P 4 L 2 "select" - "selected"

**Response**: Thanks for the catch. It will be modified in the manuscript.

**Reviewer Comment 3.12** — P5 L15: In this entire discussion you mention "highly correlated" (L19), "lower correlations" (L20), "correlates well" (L20) and many more of these statements. Such statements usually required some quantitative measures (e.g. correlation coefficient). Otherwise, what is a high correlation and what low?

**Response**: The correlation intensities are shown in wavelet power spectrum (WPS) figures using squared wavelet coefficients which yield information of the correlation between the signal at certain scale at particular location. A larger amplitude in WPS (e.g., the log10(WPS) is larger than 0.2) indicates a higher correlation which could be represented using the color codes in the figures.

**Reviewer Comment 3.13** — P5 L27 here you state you only investigate 24-, 48-, 72-h gaps. In table 1 you have much longer periods listed as well as in figure 6, while then in figure 7 again only 24, 48, 72. This is a bit inconsistent.

**Response**: Thanks for the comment. We will add more explanation that the periods listed in table 1 and figure 6 are the trained prediction windows for the models. The 24, 48, and 72 hour gaps in figure 7 and stated in P5 L27 are for testing the model performance.

**Reviewer Comment 3.14** — P5 L23 delete "clearly"

**Response**: Agreed. It will be deleted in the manuscript.

**Reviewer Comment 3.15** — P6 L3 What you mean is not a dropout layer, but the combination of a dense layer with additional dropout. Two consecutive dropout layer would mean simply applying dropout again to the result of your previous dropout output. Correctly it would state "followed by dense layer with dropout".

**Response**: Thank you for the comment. We will update the sentence to correctly describe the model.

Old sentence: The DNN architecture is shown in Figure 4, which contains three LSTM layers, followed by two consecutive dropout layers, a convolutional layer, and a final output dense layer

New sentence: The DNN architecture is shown in Figure 4, which contains three LSTM layers, followed by two dense layer with dropout, a convolutional layer, and a final output dense layer

**Reviewer Comment 3.16** — This model architecture is generally described as a stacked LSTM model, given that the LSTM layers are "stacked" on top of each other." This is a tautology. Maybe simply remove this sentence or rephrase it.

**Response**: Thank you for the comment. We will remove the sentence.

**Reviewer Comment 3.17** — P7 L7 "select" - "selected"

**Response**: Thanks for the catch. It will be modified in the manuscript.

**Reviewer Comment 3.18** — P7 L17 This is not called a "sigmoid neural net layer". You could say "A linear layer with sigmoid activation function". At least call it "neural network" not "neural net".

**Response**: Thank you for the comment. We will update the sentence to say "A linear layer with a sigmoid activation function".
Old sentence: Each gate is composed of a sigmoid neural net layer and a pointwise multiplication operation.
New sentence: Each gate is composed of a linear layer with a sigmoid activation function.

**Reviewer Comment 3.19** — P7 L17: The pointwise multiplication is not part of the gate it-self, but how the gate is combined with the cell state.

**Response**: Thank you for the comment. We will update the sentence to distinguish the multiplication from the gate. See the above response (3.18) for the updated sentence

**Reviewer Comment 3.20** — P7 L18 and Fig5: all gates (f,i,o) and the cell and hidden state are vectors and should be written in lower, bold, italics letter and not capital letters

**Response**: Thank you for the comment. We will update the gate letters accordingly.

**Reviewer Comment 3.21** — P7 L 23: "Finally, an output gate (O t ) decides what to output based on the input and previous memory state. The sigmoid layer of the output gate decides what parts of the memory state will be output..." The second sentence is basically a repetition of the first. Consider rephrasing.

**Response**: Thank you for the comment. The instruction of output gate will be rephrased in the manuscript.

**Reviewer Comment 3.22** — Table 1: Any particular reason, why you excluded 96h from the list of possible output window length, since otherwise possible input and output window length seems to be equal?

**Response**: There is no particular reason why 96h is excluded.We are trying to find the best parameters in training model and a lot of combinations need to be tested. Even without the 96hr output window,

we still could obtain the same conclusion from Figure 6d since the MAPE keeps increasing when the output window is greater than 24hr.

**Reviewer Comment 3.23** — P10 L 22 How are the terms (P, D, Q)m combined into equation 2. This needs more explanation.

**Response**: The equation 2 only contains non-seasonal terms (p, d, q) which are number of autoregressive terms, the number of nonseasonal differences and the number of moving-average terms. The terms (P, D, Q) are three additional numbers to represent the seasonal part of an ARIMA.

**Reviewer Comment 3.24** — P11 L 19: In your setting, you always extrapolate. So this statement is not correct.

**Response**: Thank you for the comment. We will correct the statement in the manuscript.

**Reviewer Comment 3.25** — P11 L 32: delete "very"

**Response**: Yes. It is deleted in the manuscript.

**Reviewer Comment 3.26** — LSTM results in general: It would be good to see only insample results at some point. How good does the LSTM perform for the same well it was trained for (as average over the 6 wells or for each well independently).

**Response**: From the reviewers comment on 3.3e, we will redo the LSTM analysis to only include the test results for the models on the same well it was trained for to be in line with the training/testing performed with the ARIMA analysis.

**Reviewer Comment 3.27** — Figure 7: Missing the information that results are only for SpC.

**Response**: Thank you for the comment. We will update the figure to explicitly state the results are for SpC only

**Reviewer Comment 3.28** — The point above applies to the entire section here.

**Response**: Thank you for the comment. We will update the section to explicitly state the results are for SpC only

**Reviewer Comment 3.29** — P12 L15: "It is noted that the optimal..." I would be cautious with such statements, unless you perform similar hyperparameter search for LSTMs as you did for ARIMA.

**Response**: We will update the sentence to limit the scope to our experimental runs.
   Old sentence: It is noted that the optimal input window size M for the LSTM models is smaller than that required by the ARIMA method for all the wells tested, indicating that LSTM models can rely on less input information than the ARIMA models to produce predictions of comparable accuracy.
   New sentence: In the experimental runs for the LSTM models, the input window size M is smaller than that required by the ARIMA method for the wells tested. This indicates the LSTM models can rely on less input information than the ARIMA models to produce predictions of comparable accuracy.

**Reviewer Comment 3.30** — P13 L 8f I do not see this in Figure 8. For me, there is no visible difference (or very hard to detect) in the Arima and LSTM error at any special frequencies. Maybe a better visualization or some quantitative measures would help.

 **Response**: We can provide a zoomed-in plot of the ARIMA and LSTM predictions for well 1-1 in figure 8 (January 2011 to March 2011) that shows this behaviour and provide quantitative measurements

**Reviewer Comment 3.31** — Figure 8. Why are the results now with the ARIMA model and 72 hour inputs and not 168 as in Figure 7?

 **Response**: The intent of figure 8 was to compare the performance of the ARIMA model and LSTM model when only given 72 hours of previous data for filling in gaps of 24 hours. We will update figure 8 to be for the ARIMA models trained with 168 inputs. Additionally, we will update figure 8 to be for the LSTM models mentioned in figure 7, in order to remove testing models on wells it was not trained on.

**Reviewer Comment 3.32** — P14 L 1 Again, I don't see the LSTM outperforming ARIMA from Figure 8 column 3. Not sure how these (also column 4) help here. Maybe it is due to my lack of understanding of the data itself, but I think some quantitative measures are better than these figures. (e.g. a table with some metrics)

 **Response**: Thank you for the comment. We can add a table of the mean relative error and MAPE for the LSTM and ARIMA models for each well in Figure 8, column 1 and 2.

**Reviewer Comment 3.33** — "In general, both LSTM and ARIMA are effective at capturing longer term variability, but LSTM is more effective at capturing high-frequency fluctuations and nonlinearities in the dataset." I don't see any (quantitative) evidence for such a statement.

 **Response**: As mentioned in the previous comment, we can add a table showing the MAPE and mean relative error of the LSTMs and ARIMA models as well as adding an additional figure showing the LSTM capturing the high-frequency fluctuations.

**Reviewer Comment 3.34** — Conclusion: As of everything written above, I think the conclusions need to be entirely rewritten, including possible new results of different model configurations etc. I will not go into more detail here, since I raised many concerns above, that apply similarly to the same statements in the conclusion (e.g. LSTM and ARIMA comparisons etc). Furthermore, you miss to say for which variable you are doing gap filling (SpC only)

 **Response**: Thank you for the comment. We will redo our conclusions section based on the additional analysis we will do. As previously stated, we will explicitly mention we are gap filling for the SpC measurement only.

**References**

Mathias Berglund, Tapani Raiko, Mikko Honkala, Leo Kärkkäinen, Akos Vetek, and Juha T Karhunen. Bidirectional recurrent neural networks as generative models. In C. Cortes, N. D.

Lawrence, D. D. Lee, M. Sugiyama, and R. Garnett, editors, *Advances in Neural Information Processing Systems 28*, pages 856–864. Curran Associates, Inc., 2015. URL `http://papers.nips.cc/paper/5651-bidirectional-recurrent-neural-networks-as-generative-models.pdf`.

X. Du, H. Zhang, H. V. Nguyen, and Z. Han. Stacked lstm deep learning model for traffic prediction in vehicle-to-vehicle communication. In *2017 IEEE 86th Vehicular Technology Conference (VTC-Fall)*, pages 1–5, Sep. 2017. doi: 10.1109/VTCFall.2017.8288312.

Kuai Fang, Chaopeng Shen, Daniel Kifer, and Xiao Yang. Prolongation of smap to spatiotemporally seamless coverage of continental u.s. using a deep learning neural network. *Geophysical Research Letters*, 44(21):11,030–11,039, 2017. doi: 10.1002/2017GL075619. URL `https://agupubs.onlinelibrary.wiley.com/doi/abs/10.1002/2017GL075619`.

A. Graves, A. Mohamed, and G. Hinton. Speech recognition with deep recurrent neural networks. In *2013 IEEE International Conference on Acoustics, Speech and Signal Processing*, pages 6645–6649, May 2013. doi: 10.1109/ICASSP.2013.6638947.

Sepp Hochreiter and Jürgen Schmidhuber. Long short-term memory. *Neural Comput.*, 9(8): 1735–1780, November 1997. ISSN 0899-7667. doi: 10.1162/neco.1997.9.8.1735. URL `https://doi.org/10.1162/neco.1997.9.8.1735`.

M. JORDAN. Attractor dynamics and parallelism in a connectionist sequential machine. *Proc. of the Eighth Annual Conference of the Cognitive Science Society (Erlbaum, Hillsdale, NJ), 1986*, 1986. URL `https://ci.nii.ac.jp/naid/10018634949/en/`.

M Khalil, U.S Panu, and W.C Lennox. Groups and neural networks based streamflow data infilling procedures. *Journal of Hydrology*, 241(3):153 – 176, 2001. ISSN 0022-1694. doi: https://doi.org/10.1016/S0022-1694(00)00332-2. URL `http://www.sciencedirect.com/science/article/pii/S0022169400003322`.

F. Kratzert, D. Klotz, C. Brenner, K. Schulz, and M. Herrnegger. Rainfall–runoff modelling using long short-term memory (lstm) networks. *Hydrology and Earth System Sciences*, 22(11):6005–6022, 2018. doi: 10.5194/hess-22-6005-2018. URL `https://www.hydrol-earth-syst-sci.net/22/6005/2018/`.

Jun Liu, Amir Shahroudy, Dong Xu, and Gang Wang. Spatio-temporal lstm with trust gates for 3d human action recognition. In *European Conference on Computer Vision*, pages 816–833. Springer, 2016.

Razvan Pascanu, Caglar Gulcehre, Kyunghyun Cho, and Yoshua Bengio. How to construct deep recurrent neural networks, 2013.

K. Saleh, M. Hossny, and S. Nahavandi. Intent prediction of vulnerable road users from motion trajectories using stacked lstm network. In *2017 IEEE 20th International Conference on Intelligent Transportation Systems (ITSC)*, pages 327–332, Oct 2017. doi: 10.1109/ITSC.2017.8317941.

Sijie Song, Cuiling Lan, Junliang Xing, Wenjun Zeng, and Jiaying Liu. An end-to-end spatiotemporal attention model for human action recognition from skeleton data. In *AAAI*, volume 1, pages 4263–4270, 2017.

Ketaki Ustoorikar and M.C. Deo. Filling up gaps in wave data with genetic programming. *Marine Structures*, 21(2):177 – 195, 2008. ISSN 0951-8339. doi: https://doi.org/10.1016/j.marstruc.2007.12.001. URL `http://www.sciencedirect.com/science/article/pii/S0951833907000676`.

Duo Zhang, Geir Lindholm, and Harsha Ratnaweera. Use long short-term memory to enhance internet of things for combined sewer overflow monitoring. *Journal of Hydrology*, 556:409 – 418, 2018. ISSN 0022-1694. doi: https://doi.org/10.1016/j.jhydrol.2017.11.018. URL `http://www.sciencedirect.com/science/article/pii/S0022169417307722`.

Wentao Zhu, Cuiling Lan, Junliang Xing, Wenjun Zeng, Yanghao Li, Li Shen, and Xiaohui Xie. Co-occurrence feature learning for skeleton based action recognition using regularized deep lstm networks, 2016. URL `https://www.aaai.org/ocs/index.php/AAAI/AAAI16/paper/view/11989/12149`.

---

## Editor Comment (EC1) · Dimitri Solomatine (Editor) · 9 Feb 2022

The provided reviews have been very useful, with references to learlier research papers. The authors responded to them adequately, showing the ways and directions of revising the paper.

---

## Author Response (AR1)

**Response to the short comments**

**Reviewer 1**

**Reviewer Comment 1.1** — This paper uses long short-term memory (LSTM) neural networks to fill in gaps in spatially distributed time-series data. The performance of the LSTM-based gap-filling method is compared to that of a traditional, popular gapfilling method: autoregressive integrated moving average (ARIMA). Overall, this paper is well written, structured and results seem sufficiently justified and useful. However, this paper is very technical and there is no physical insight beyond just feeding data into a standard code. I think this paper should be published as technical note (not as research article). Several aspects could be further improved in order to having it published in this journal.

**Response**: We thank the reviewer for overall positive assessment of our manuscript. While we emphasized the scientific importance of gap-filling the spatio-temporal data to capture and understand dynamic behaviors of complex systems, we also agree with the reviewer that our primary focus was to introduce a technical method that can fill data gaps by capturing the dynamic features using deep neural network that contains LSTM layers. We are resubmitting this manuscript as a technical note as suggested.

**Reviewer Comment 1.2** — Would you guarantee the LSTM method in your paper can achieve the same excellent performance in other areas of the whole world? Is it possible that the good performance of the LSTM model is just applicable for the case given by the manuscript? The authors should include one more test for another area (maybe not in the text but in the supporting materials).

**Response**: There is unfortunately no guarantee for any model to have the same performance in other applications, which is the case for all data-driven and physics-based models. The performance of data-driven models can be optimized by iterating on various model configurations based on data types and characteristics of the relations between predictors and desired responses, as we have demonstrated in our study case when comparing ARIMA with LSTM-based DNNs. Sometimes the ARIMA works better, and sometimes the DNNs work better, and we believe that we have started down the path of predicting which model will work better for a given case. The LSTM-based DNN model we adopted in our study has the same chain-like nature as other recurrent neural networks, meaning that this architecture lends itself well to sequences, so it will often be a useful (if not the best) approach for dynamic system behaviors (Karim et al., 2017; Malhotra et al., 2015; Kratzert et al., 2018; Malhotra et al., 2016; Wang et al., 2017; Reddy and Prasad, 2018; Lipton et al., 2015). The optimal model configuration and performance we can achieve would be case by case, and our focus of this technical note is to introduce a general method that can be broadly applied to other systems and be evaluated similarly. We have emphasized this aspect of transferrability in our discussion and conclusion sections. We call on the community participation to test the transferrability of this method to other monitoring systems.

**Reviewer Comment 1.3** — LSTM model is only compared to ARIMA. Why not compare LSTM with other widely-used methods (such as Kriging interpolation and Gaussian process)? Furthermore, are the authors familiar with DIEOF (Data Interpolating Empirical Orthogonal Functions) which are proposed by Beckers and Rixen (2003)? I think that DIEFOF is powerful and

useful for filling temporal and spatial gaps in geophysical datasets. Maybe the authors can compare LSTM with DIEOF.

**Response**:  There are many interpolation approaches that are commonly used, including the EOF-based approaches and kriging (based on Gaussian processes) that the reviewer mentioned. We did discuss that kriging and the Gaussian processes are mostly used for spatial interpolation rather than spatio-temporal interpolation. We have added more discussions on the EOF related interpolation methods, such as least squares EOF (LSEOF), data interpolation EOF (DINEOF), and recursively subtracted EOF (REEOF), which are widely used to fill in missing data from geophysical fields such as clouds in sea surface temperature datasets or other satellite-based images with regular gridded domains. However, as discussed by Beckers and Rixen (2003), "For the method to have a chance to work, one needs, for each moment, at least a sufficiently large number of data points (otherwise one should drop the whole picture) and for each spatial point a sufficient amount of data in time (otherwise one should discard the point from the analysis)", which is a challenge for most of the monitoring networks that are sparsely distributed. Therefore, we keep using ARIMA as the benchmark since it is the most commonly used method in time series analysis (a conclusion based on reviewing the hydrological literature) and gap filling. While we acknowledge that we did not (and could not) explore every possible interpolation method, we feel that by choosing such a representative approach is appropriate for assessing the performance of our proposed method without loss of generality. We have also acknowledged this aspect in our introductions, discussion and conclusion sections.

New description of DINEOF has been added in manuscript at P3L1-P3L5: " Empirical Orthogonal Functions (EOF) related interpolation methods, such as least squares EOF(LSEOF), data interpolation EOF (DINEOF), and recursively subtracted EOF (REEOF), are widely used to fill in missing data from geophysical fields such as clouds in sea surface temperature datasets or other satellite-based images with regular gridded domains (Beckers and Rixen, 2003; Beckers et al., 2006; Alvera-Azcárate et al., 2016). However, the requirement of gridded data by the EOF methods limits their use in filling data gaps in irregularly spaced monitoring networks."

**Reviewer Comment 1.4** — The present title "Using Deep Learning to Fill Spatio-Temporal Data Gaps in Hydrological Monitoring Networks" are inaccurate. I suggest new title like "Using Long Short Term Memory Neural Network Model to Fill Spatio-Temporal Data Gaps in Hydrological Monitoring Networks"

**Response**:  Our neural network models contain both LSTM and CNN layers. Therefore, we modified our title to : "Technical note: Using Deep Neural Network Models to Fill Spatio-Temporal Data Gaps in Hydrological Monitoring Networks". Deep neural network is a broader term that allows flexible architecture we are using to include multiple types of layers.

**References**

Alvera-Azcárate, A., Barth, A., Parard, G., and Beckers, J.-M.: Analysis of SMOS sea surface salinity data using DINEOF, Remote Sensing of Environment, 180, 137 – 145, https://doi.org/https://doi.org/10.1016/j.rse.2016.02.044, URL `http://www.sciencedirect.com/science/article/pii/S0034425716300724`, special Issue: ESA's Soil Moisture and Ocean Salinity Mission - Achievements and Applications, 2016.

Beckers, J. M. and Rixen, M.: EOF Calculations and Data Filling from Incomplete Oceanographic Datasets*, Journal of Atmospheric and Oceanic Technology, 20, 1839–1856, https://doi.org/10.1175/1520-0426(2003)020⟨1839:ECADFF⟩2.0.CO;2, URL `https://doi.org/10.1175/1520-0426(2003)020<1839:ECADFF>2.0.CO;2`, 2003.

Beckers, J. M., Barth, A., and Alvera-Azcárate, A.: DINEOF reconstruction of clouded images including error maps. Application to the Sea-Surface Temperature around Corsican Island, Ocean Science Discussions (OSD), 2, https://doi.org/10.5194/os-2-183-2006, 2006.

Karim, F., Majumdar, S., Darabi, H., and Chen, S.: LSTM fully convolutional networks for time series classification, IEEE access, 6, 1662–1669, 2017.

Kratzert, F., Klotz, D., Brenner, C., Schulz, K., and Herrnegger, M.: Rainfall–runoff modelling using Long Short-Term Memory (LSTM) networks, Hydrology and Earth System Sciences, 22, 6005–6022, https://doi.org/10.5194/hess-22-6005-2018, URL `https://www.hydrol-earth-syst-sci.net/22/6005/2018/`, 2018.

Lipton, Z. C., Kale, D. C., and Wetzel, R. C.: Phenotyping of clinical time series with LSTM recurrent neural networks, arXiv preprint arXiv:1510.07641, 2015.

Malhotra, P., Vig, L., Shroff, G., and Agarwal, P.: Long short term memory networks for anomaly detection in time series, in: Proceedings, vol. 89, Presses universitaires de Louvain, 2015.

Malhotra, P., Ramakrishnan, A., Anand, G., Vig, L., Agarwal, P., and Shroff, G.: LSTM-based encoder-decoder for multi-sensor anomaly detection, arXiv preprint arXiv:1607.00148, 2016.

Reddy, D. S. and Prasad, P. R. C.: Prediction of vegetation dynamics using NDVI time series data and LSTM, Modeling Earth Systems and Environment, 4, 409–419, 2018.

Wang, Q., Guo, Y., Yu, L., and Li, P.: Earthquake prediction based on spatio-temporal data mining: an LSTM network approach, IEEE Transactions on Emerging Topics in Computing, 2017.

**Response to referee comments**

**Reviewer 2**

**General Remarks:** The paper presents an interesting use of deep learning with LSTM Networks for infilling groundwater data. The article is timely and tries to make a comprehensive description and explanation of how the Deep learning technique is implemented using statistical and machine learning techniques. The paper is a welcome contribution to the field of groundwater and hydrological earth sciences. However, I cannot recommend publication in the present form due to the comments and questions raised. The paper needs major revision.

 **Response**:  We thank the reviewer for the positive assessment of our manuscript and the constructive comments.  We have addressed all the individual comments to improve our manuscript for possible publication.

**Reviewer  Comment  2.1**  —  The paper states that long-term spatiotemporal changes in subsurface hydrological flow is usually quantified using a network of wells. However this paper does not deal with the long-term trend or analysis. Hourly data is hardly interpreted or used for the long term. Hourly information for sure contains noise that would be advisable to remove for the long term analysis.

 **Response**:  There seems to be some confusion between long-term changes and long-term trends. We have rephrased to clarify that monitoring networks capture the dynamic system behaviors over long time windows, which allows the discovery of signals over a spectrum of time scales as revealed by the spectral analysis we presented in the manuscript. While long-term trends, e.g., low-frequency variations, are usually smoother and could be captured by existing time series analysis method like ARIMA (see our comparison analyses between LSTM-based DNN and ARIMA gap filling results), our focus is on capturing high-frequency dynamics that are important signatures for understanding managed systems. We have clarified those points in our objectives and reiterate in conclusions to avoid confusion.

**Reviewer  Comment  2.2**  —  Observations are mentioned to be spatially sparse, and temporal gaps exist.  Many papers have solved the same type of problem, without using the term spatiotemporal. Almost every course in hydrology deals in one chapter with the issue of using spatial correlation and temporal correlation to fill in data. So in this respect, the authors are invited to clearly indicate what innovation is brought by this work to spatiotemporal analysis.

 **Response**:  We agree with the reviewer that a lot has been done in hydrology for spatial and temporal analyses.  However, there have been very few studies that address spatial and temporal correlations simultaneously due to the difficulty in parameterizing the spatial and temporal correlations all together. Deep neural networks provide an alternative way to represent such correlations without assuming the explicit form of correlations a priori, which is the innovation our work originally aimed to bring and demonstrate. However, we found there are multiple steps towards accomplishing that goal as it involves merging two types of deep neural networks to represent both the spatial and temporal correlations and evaluate various configurations thoroughly. In order to bring the spatial component together with the temporal correlations, also related to a comment raised by Reviewer #1, we added multi-well DNN models to take advantage of information from neighboring wells.  The multi-well DNN models were found

to ourperform their single-well counterparts as shown in the newly added Section 4.3. We emphasized in the introduction and conclusion that the primary advantage of using DNN approach is to address both spatial and temporal correlations without assuming their explicit forms before hand.

**Reviewer Comment 2.3** — Following point two, it is known that in most of the cases, aquifers with little or no human intervention have low variability. Conventional guidelines and measures in hydrogeological science are typically based on monthly data.

**Response**: It is true that some aquifers with no or little human intervention have low variability, for which monthly data could be sufficient to understand the system behavior. However, anthropogenic activities, in particular, dam operations, have increasingly impacted the river and aquifer systems by altering the exchange patterns between river water and groundwater, and the associated thermal and biogeochemical processes (Song et al., 2018; Shuai et al., 2019; Zachara et al., 2020). Due to significant, high-frequency (hourly) stage variations caused by dam regulations to meet power generation needs, it is insufficient to use monthly data to understand such systems as have been demonstrated in numerous studies performed at our study site. Our study site is representative of many dam-regulated gravel-bed rivers across the world. Therefore, our study could have broader impacts to many other systems. We have made this point clear in our study site description section.

We added a paragraph at P4 L30-P5 L6: "The understanding we developed from earlier studies is that the physical heterogeneity contributes to the different response behaviors at different locations while the river stage dynamics lead to multi-frequency dynamics in those responses. The seasonal and annual variations are driven by natural climatic forcing (Amaranto et al., 2019, 2018), whereas the higher-frequency (i.e., daily and sub-daily) fluctuations are primarily induced by operations of the upstream hydroelectric dam operations to meet various demands of human society (Song et al., 2018). Our system is representative of many dam-regulated gravel-bed rivers across the world, where dam operations as a typical anthropologenic activity have significantly altered the hydrologic exchanges between river water and groundwater, as well as the associated thermal and biogeochemical processes (Song et al., 2018; Shuai et al., 2019; Zachara et al., 2020). Note that the multi-frequency variations in data are characterizing the dynamic features of data, which could exist in both short-term and long-term time series data as a result of short-term or long-term monitoring effort."

**Reviewer Comment 2.4** — In the present paper the idea of nonlinear dynamics is mentioned almost everywhere in the introduction and justification of the work. This is somewhat surprising and needs better justification, since groundwater dynamics, in many cases, can be represented with linear models. As it is concluded in this paper results, ARIMA can approximate the system quite well.

**Response**: This comment is related to the earlier comment 2.3 as high-frequency dynamics lead to higher level of nonlinearity in system responses, especially for the specific conductance that is a result of mixing of water from various sources. We have shown that a linear model like ARIMA was not able to capture such nonlinearity, while LSTM-based DNN performed better. We have explained this point in the revised manuscript. Please also refer to our response to comment 2.3 for the importance of capturing high-frequency dynamics for many dam-regulated systems, which was also better articulated in the revised manuscript.

**Reviewer Comment 2.5** — The particular case study presented here shows a relative complex

dynamic nature indeed, but it seems it is due to human intervention (however I could be wrong). Can you comment on this and the uncertainties associated?

**Response**: The reviewer is partially right that human intervention contributes to the complexity of system behavior by creating high-frequency flow dynamics. However, the full complexity is a result of interactions between such human-induced variations and the natural heterogeneity of aquifer physical properties (Zachara et al., 2020). There is significant uncertainty associated with aquifer physical heterogeneity at our study site as revealed by previous studies. We have added these additional discussion about the system complexity in the revised manuscript.

a The human intervention might affect your calculation and therefore, extractions might not be following a random but more human induced behaviour. So data understanding or replicability used in one year might not be the same in another. It would be advisable first to check how much and when extraction took place. Is this data filled in for a long term analysis, or short-term? This question arises since the hourly step is used.

**Response**: Please refer to our response to Comment 2.1 for explaining our use of long-term data versus long-term trend analyses. The reviewer is right that high-frequency flow variations are mainly caused by the dam operations while the seasonal and interannual variabilities are controlled by climatic forcing like precipitation and melting of snow pack in the headwater systems. We have clarified the drivers of the high-frequency and low-frequency variations in the revision. We used multiple years of training data from dry, normal and wet hydrologic years to capture potential operational patterns associated with various conditions. LSTM units include a 'memory cell' that can maintain information in memory for long periods of time.

The model learns to extract the information from the input during the training process, typically by the end of 30 epochs we train the model for. The amount of information maintained by the LSTM is dependent on the size of the input time series. Future work is required to identify what information the LSTM model remembers in the cell, i.e., opening the black box of the DNN model. While this is a very important next step and an active research area, it is beyond the scope of this technical note to describe the relevant method and results imagining the level of effort that is required. After the data gaps are filled, the dataset can be used for both long-term or short-term analyses as we will have long-term dataset with high temporal resolution.

To illustrate what can be done to open the DNN box, we are providing an example where we examined the memory cells of a model with an input window size of 120 hours, output of 1 hour, trained on data from well 1-1. We investigated which of the inputs (SpC, water level, temperature) drives the state of the LSTM units, i.e., which input has the most influence on the LSTM units, for the validation period (year 2011). We divided the data into 8640 input samples of 120 hours each and ran each input through the trained model. We then extracted the hidden state of each unit in the three LSTM layers at each time sample and calculated the Pearson correlation coefficient between the hidden states of all the units and the normalized input. Next, we calculated the percentage of the 8640 samples for each input variable to have the largest positive and negative correlations with the hidden states of the LSTM units in each of the three layers. The results for the top positive and negative correlations are show in table 2.1, from which we see that the water level occurs the most frequent (almost 50% of cases) as the highest positive and negative correlation for the LSTM units in layers 1 and 2. SpC is consistently the second most frequent as the highest correlated in all the layers. SpC occurs most frequently (42.62%) as the highest

negatively correlated with the hidden states in the third LSTM layer. Thus, for this given model, the memory of the LSTM units are most strongly driven by the dynamics of the water level in the input, then followed by the dynamics of the SpC.

| Layer 1 | | | |
|---|---|---|---|
| **Rank** | **SpC (%)** | **Temperature (%)** | **Water Level (%)** |
| Highest Positive | 32.6% | 14.9% | 52.4% |
| Highest Negative | 32.4% | 22.1% | 45.6% |

| Layer 2 | | | |
|---|---|---|---|
| **Rank** | **SpC (%)** | **Temperature (%)** | **Water Level (%)** |
| Highest Positive | 28.5% | 20.2% | 51.3% |
| Highest Negative | 29.6% | 16.3% | 54.2% |

| Layer 3 | | | |
|---|---|---|---|
| **Rank** | **SpC (%)** | **Temperature (%)** | **Water Level (%)** |
| Highest Positive | 28.2% | 20.2% | 51.6% |
| Highest Negative | 42.6% | 17.9% | 39.4% |

Table 2.1: Percent of input samples in 2011 (8640 in total) for each measurement to have the largest positive and negative Pearson correlation coefficient (R) with the hidden states of each LSTM layer

Taking a closer look, as seen in figure 2.1, the top positively correlated LSTM unit with SpC (R = 0.897) closely follows the dynamics of the SpC.

b If indeed human intervention influence the dynamics of the groundwater system, the logical approach would be to find a variable to represent direct or indirect measurement of extractions.

**Response**: Thanks for the suggestion. Please refer to our response to your earlier comment. We have looked into both memory cells and state cells to illustrate how and where the extraction occurs.

c It is suggested to read the paper by Amaranto et al. (2018) "Semi-seasonal groundwater forecast using multiple data-driven models in an irrigated cropland". J Hydroinformatics, 20 (6): 1227–1246. DOI: `https://doi.org/10.2166/hydro.2018.002` and - Amaranto et al. ( 2019). A spatially enhanced data driven multimodel to improve semiseasonal groundwater forecasts in the High Plains aquifer, USA. Water Resources Research, 55, 5941– 5961. `https://doi.org/10.1029/2018WR024301`

**Response**: Thank you for the paper suggestions. Both of the papers listed above use data-driven approaches to improve groundwater forecasts. The MuMoC framework select neighboring wells to assist groundwater predictions is of our interest. Although the authors used data with coarser temporal resolution (daily or monthly) to make monthly predictions, which is different from our purpose of filling short gaps (up to 3 days) for capturing high-frequency dynamics, the idea of using information from neighboring wells applies to our case. We have explored multi-well DNN

[Figure]

Figure 2.1: Well 1-1 normalized measurements over time (2011-04-15 04:00:00 to 2011-04-20 03:00:00), along with the hidden state of the top positvely correlated LSTM unit in the third layer

models to use information from neighboring wells, which led to improved accuracy in gap filling. Please refer to the new added section 4.3: Performance of multi-well DNN models. We have reviewed and discussed these two papers in our revision.

**Reviewer Comment 2.6** — The regional aquifer and geology might play a more significant role in the study, since not only the river but the size and other interventions and hydrometeorological recharges might be correlated.

**Response**: We agree that the regional aquifer and geology play an important role as shown in previous studies performed by our colleagues (Chen et al., 2012, 2013; Zachara et al., 2020). The aquifer is composed of two distinct geologic formations, a highly permeable formation (Hanford formation, consisting of coarse gravelly sand and sandy gravel) underlain by a much less permeable formation (the Ringold Formation, consisting of silt and fine sand). The dominant hydrogeologic features of the aquifer are defined by the interface between the Hanford and Ringold formations and the heterogeneity within the Hanford formation. The understanding we developed from these earlier studies is that the physical heterogeneity contributes to the different response behaviours at different locations while the river stage dynamics lead to multi-frequency dynamics in those responses. We have added more information and re-organized the entire section 2: Study Site and Data Description to better describe our system to help readers understand. The recharge from rainfall is negligible due to the semi-arid climate.

**Reviewer Comment 2.7** — The stations are so close, and the hourly variation appears to be periodic with an amplitude of 4 or 5cm, according to Figure 1 (and on other graphs). It is intriguing, the question I would have is what happens every hour? and if this hourly variation is noise on the measurement device or data? What is the precision of the measurement device? What is the volume of water extracted to reach the variation of 1 cm? Where the recharge water comes from(has this been studied in the past)? Is this 5 cm recharge volume feasible in one hour? Could

be the water from the river affecting your measurements (interflow)? It is advisable to present the time series of the river flow. It would be also useful to have a few hydrological balances (note that this is a hydrological journal). The problematic still can be questioned due to its apparent complex dynamics with the river and human intervention (not a typical, natural aquifer).

**Response**: The reviewer is right that the water table elevation difference is small due to the close distance between wells and the highly permeable aquifer material (hydraulic conductivity in the range of 4000-7000 m/d). The rapid change in groundwater table is at first caused by pressure wave propagation from the river stage variation, and then by recharge water coming from the river or displacement of groundwater from other parts of the aquifer depending on flow directions and locations of interest. The stainless-steel pressure transducer CS451 from campbell scientific (Scientific) was used for water level measurements. The measurement range of the pressure transducer in our study site is 0 to 10.2m with a standard accuracy of $\pm 0.1\%$, which leads to an accuracy of 1cm. In this case, the pressure changes with an amplitude of 4 5cm are the actual measurements and our consistent 10-year data has proven this point. In our revision, we have included the measurement accuracy information and numerous hydrologic modeling studies(Song et al., 2018; Shuai et al., 2019; Zachara et al., 2020) performed at the site to help readers better understand the flow conditions and where the recharging water comes from.

**Reviewer Comment 2.8** — On the model setup, Please explain why you use Mx128.

**Response**: We use 128 (i.e. 128 units for each LSTM layer) because this number of units showed better performance after we experimented with different model architectures with different number of units. We have added this rationale to the manuscript at P10 L5-L6 "Each of the three LSTM layers has 128 units because this configuration outperformed others with more or fewer number of units".

**Reviewer Comment 2.9** — Page 7, line 10, mentions the supplemental material, but I cannot find it in the paper.

**Response**: The supplemental materials can be found using this link
    https://www.hydrol-earth-syst-sci-discuss.net/hess-2019-196/hess-2019-196-supplement.pdf

**Reviewer Comment 2.10** — Important: choice of (a very complex model) LSTM has to be justified, since it seems AR-type models is enough. Frankly, I don't see the need for complex models like LSTM, but if you have arguments to defend your position, please present them to convince the readers.

**Response**: We are interested in using an LSTM-based DNN for this problem because DNNs have had success in predicting values in time-series data without assuming explicit temporal dependence forms. We added several examples of this on P3 L12. We aimed to explore whether an DNN model would provide improved performance over traditional methods (i.e. ARIMA). Our study demonstrated that more complex DNN models were able to capture high-frequency variations in system dynamics, for which a simpler ARIMA model failed to capture. The choice of the DNN architecture was a result of hyperparameter search based on the validation performance of the models. Related to our response to other relevant comments, we restructured the methodology section and added more reasoning for the choice of the DNN architecture and the advantages and necessity of using LSTM-based DNNs in

our revision. For example, We have expanded the introduction section to discuss recent applications of LSTMs to hydrology and earth sciences and the relevance to gap filling problems.

New section (P3 L21-P43): "There have been applications of RNNs and LSTMs emerging in hydrology. For example, Kratzert et al. (2018) used LSTMs to predict watershed runoff from meteorological observations, Zhang et al. (2018) used LSTMs for predicting sewer overflow events from rainfall intensity and sewer water level measurements, and Fang et al. (2017) used LSTMs to predict soil moisture with high fidelity. Compared to a single RNN/LSTM layer, more complex LSTM architectures such as stacked and bidirectional LSTMs, CNN-LSTM or convolutional LSTM have the potential to capture extra features (Graves et al., 2013; Pascanu et al., 2013) as shown in various applications, including action recognition (Zhu et al., 2016) and vulnerable road users location predictions (Saleh et al., 2017). A bidirectional RNN/LSTM works by duplicating the recurrent network into two networks: one responsible for fitting the positive time direction (i.e. the forward states) and the other responsible for the negative time direction (i.e the backwards state)(Schuster and Paliwal, 1997). In general, the input sequence is fed as-is to the forward state and a reversed copy of the input sequence is fed to the backwards state. The bidirectional LSTM can be used in history matching problems.

Our study aims to evaluate the potential of using LSTM layers within a DNN architecture to fill gaps in spatio-temporal environmental time series. We treat the gap filling as a forecasting problem, i.e., we use the historical data as input to predict the missing values in the data gaps. We demonstrate our method using a test case that focuses on understanding the interactions between a regulated river and contaminated groundwater aquifer. We adopt the stacked-LSTM combined with the convolutional layer as our DNN model to understand the interactions between a regulated river and contaminated groundwater aquifer. The DNN-based gap filling method is compared with traditional time series approaches (e.g., ARIMA) to identify situations in which DNNs outperform ARIMA as well as what the optimal configurations might be for this particular application."

**Reviewer Comment 2.11** — On page 14, it states that other configurations of LSTM can be further explored; however, it is not clear why this was not done before. Not sure why the selected configuration was just tried to see if it works or not, without any analysis what is the best structure. This relates to comment 8 and 9.

 **Response**: We acknowledge that we did not explain this point as well as we could have. We performed hyperparameter searches on: Number of LSTM layers, number of units per LSTM layer, number (and size of) dense layers, activation functions. This was performed for data on one well (399-1-1) with a smaller subset of input and output prediction windows, experimenting with different architecture configurations. However, this was not an exhaustive search of all possible configurations. We have dedicated a subsection 3.1.1 on hyperparameter search to make this clear.

**Reviewer Comment 2.12** — I am a bit in confusion how to interpret the statements made in conclusion. The ARIMA is not suited or less suited for filling high frequency (hourly, or short gaps) and more suitable for a long term period (24, 48 and 74 hours). It is suggested we need deep learning for filling high-frequency gaps (of one hour)?. Maybe is good to elaborate on the simplicity of what this translates to, I am not sure if the meaning is right.

 **Response**: We acknowledge a potential source of confusion in terms of high-frequency fluctuations in system states versus short-term or long-term data records/gaps. In the revised manuscript, we made extra effort to distinguish those two concepts to avoid further confusion. In our study, we found ARIMA

to be suitable for time series with less dynamic behavior, while DNNs excel in capturing high-frequency dynamics (daily and subdaily) in time-series observations for various data gaps.

**Reviewer Comment 2.13** — Not sure if there is an idea of how high is the overall error; in the figure 8, with well 1-15 it seems almost perfect representation (zero error in the validation data for many points). Also in the same well, it appears like high negative correlation up to 128 hours.

 **Response**: Well 1-15 has the smallest error as been illustrated in Figures 8 and 9. The mean error shown in testing period in Figure 9 is 0.05%. Please note that the log10 scale of wavelet power spectrum is plotted in Figure 9, where the blue colors represent weak power rather than negative correlation.

**Response to referee comments**

**Reviewer 3**

**General note:** I was asked by the editor to review this manuscript, although groundwater hydrology is not my area of expertise. However, machine learning is and therefore most of my review will be around the methods and experimental setting used in this manuscript. This manuscript presents an approach for filling gaps in time series of ground water well measurements. Specifically, the authors compare two different methods (LSTM-based and ARIMA) for different gap lengths for six different wells. Although I generally welcome publications that try to make use of deep learning based methods for various applications in earth science, I see various major concerns with the manuscript at hand. Overall, it seems like the authors are not too familiar with the methods they apply (especially the LSTM-based model) and many decisions made seem questionable and lack any justification or explanation. Because of these concerns, I'm not sure if I can recommend this manuscript for publication. If it should be published at all, major revisions are required.

**Response**: Thank you for reviewing and providing the summary. We agree with the reviewer that it is important to demonstrate that we know what we are doing, and we appreciate the reviewer's careful attention to make sure we did our due diligence. We have taken major revisions to address all the comments raised by the reviewer, as illustrated by the point-by-point response below.

**Reviewer Comment 3.1** — Model architecture: Coming from the field of machine learning, I was surprised by the creativity of the authors in finding their model architecture. To be honest, I have never seen such a combination of LSTM layers, dense layers and convolutional layers for a time series task and I wonder if the authors know what they are doing. Here is a list of sub points to this major comment:

a First: Did you perform any hyperparameter search at all to find this architecture? If yes, please give details on the model configurations (in terms of layers) you tried, if not, why not? To propose such an exotic architecture, it is required to see quantitative evidence that this is required and not a much simple LSTM-based model would be better (e.g. single LSTM layer with single dense + dropout layer)

**Response**: We performed hyperparameter searches on: Number of LSTM layers, number of units per LSTM layer, number (and size of) dense layers, activation functions. This was performed for data on one well (399-1-1) with a smaller subset of input and output prediction windows, experimenting with different architecture configurations. We have dedicated a subsection 3.1.1 on hyperparameter search.

New description is at P9 L12 to P10 L6: We performed a hyperparameter search to explore different model architecture configurations, i.e., the number of LSTM layers, number of units per LSTM layer, number (and size of) dense layers, and activation functions. The search was performed on well 1-1 only due to computational cost. We chose the optimal DNN architecture using model performance on validation data set of well 1-1 (see Table 1) using MAPE defined in Eq. (1).

The final DNN architecture, as shown in Figure 4, contains three LSTM layers, followed by two dense layers with dropout, a convolutional layer, and a final output dense layer. Stacking three

layers of LSTM was found to yield better performance than a one- or two-layer architecture. Each of the three LSTM layers has 128 units because this configuration outperformed others with more or fewer number of units.

b Why do you stack 3 LSTM layers? In theory, a single LSTM layer is turing-complete. Besides probably natural language processing, where the training data consists of million/billion of samples, there is almost always no need to use more than a single LSTM layer. Additionally, since you have very limited training data (2 years of hourly data are just 17520 data points), the size of your LSTMs seem to be exorbitantly large. Especially with 3 LSTM layers.

**Response**: In response to using multiple LSTM layers, there has been research looking at the benefits of using multiple RNNs/LSTMs in a model in comparison to a single RNN/LSTM (Graves et al., 2013; Pascanu et al., 2013). Likewise, there has been work in using multiple LSTMs for action recognition (Zhu et al., 2016), traffic prediction (Du et al., 2017), and vulnerable road users location predictions (Saleh et al., 2017). As such, we investigated the potential benefits of using multiple LSTM layers in our problem domain. We have added additional sentences in the revision to discuss previous uses of stacked LSTMs and some comparisons of single versus multiple in different domains to give context on why we are interested in this model architecture and updated our references with the cited articles.

New paragraph (P3 L24-L27): Compared to a single RNN/LSTM layer, more complex LSTM architectures such as stacked and bidirectional LSTMs, CNN-LSTM or convolutional LSTM have the potential to capture extra features (Graves et al., 2013; Pascanu et al., 2013) as shown in various applications, including action recognition (Zhu et al., 2016) and vulnerable road users location predictions (Saleh et al., 2017). .

c Why the combination of convolutional layers and dense layers after the LSTM? Probably the standard is to have a single dense layer that uses the hidden output of the LSTM to map to your desired target shape. Why do you think so much complexity is needed after the LSTM, since the LSTM should capture the complex temporal dependencies already?

**Response**: As stated in our response in 3.1a, we performed some hyperparameter searches, experimenting with different architecture configurations which led us to use convolutional and dense layers. We acknowledge that more information on the extensive analysis and experimentation we have performed would be useful in further justifying the choice of model architecture, so we have provided those details in supplemental material.

d Why do you have the convolutional layer at all? If I understand your setting correctly, the convolutional layer can again look at the entire sequence (M x 64, with M the input sequence length). Why is this necessary? The task of the LSTM is to summarize the input sequence and store all the information necessary for predicting the M+1 time step (first step of your N time step long gap) in it's cell state. e. Another point related to the convolutional layer. I see that the filter size was solely chosen to be able to map from a sequence length of M to an output of N (filter size M-N+1). However, are the authors aware of what that means? For example, for predicting the first of the N time steps, the convolutional filter will only look at the first M-N+1 input sequence elements, effectively ignoring what has happened at the time steps preceding the current time step. Why do you want this? It makes absolutely no

sense to not include the most informative information (the previous time steps) necessary to predict the next time step.

**Response**: Yes, the intent was to map from a sequence length of M to an output of N. The reviewer is correct that the convolutional filter does limit the model in ignoring the most recent time steps. As stated in our response to comment 3.1c, we felt our exploration of architectures, including using convolutional layers, resulted in a good architecture. Furthermore, the time steps immediately preceding the current time are not necessarily the most informative information in the presence of dynamical behavior. However, in response to the reviewers concern, we trained models with a single LSTM and dense layer and compared the results against the original architecture (see response in 3.3c).

**Reviewer Comment 3.2** — Related work: Since (correct me if I'm wrong) this is not a forecast task, but just filling gaps in historic data records, I wonder if the authors have done some research, which approaches are currently used in the field of deep learning, before proposing their own method. E.g. for gap filling in historic time series, Bi-directional LSTMs are commonly used over normal LSTMs, since they do two sided gap filling (closer to interpolation), compared to the standard LSTM, which basically extrapolates into the future. I would also advise to add some related work section of LSTM-based gap filling into the introduction.

**Response**: The reviewer is correct that the goal is to test gap filling in historical records. For our work, we treat the gaps as a forecasting problem which means we use the historical data as input to predict the values during gap period. Bi-directional architectures have been used for gap-filling and is another model type applicable to the work if we treat the gap filling as a history matching problem. As we stated in the conclusion section, the bi-directional LSTMs can be explored in future work to keep the scope of this study manageable. We have also added a paragraph describing multiple deep learning techniques that have been applied to gap filling in hydrology, including LSTMs. We have also added a brief description regarding bi-directional LSTMs.

New Description P3 L21-31: There have been applications of RNNs and LSTMs emerging in hydrology. For example, Kratzert et al. (2018) used LSTMs to predict watershed runoff from meteorological observations, Zhang et al. (2018) used LSTMs for predicting sewer overflow events from rainfall intensity and sewer water level measurements, and Fang et al. (2017) used LSTMs to predict soil moisture with high fidelity. Compared to a single RNN/LSTM layer, more complex LSTM architectures such as stacked and bidirectional LSTMs, CNN-LSTM or convolutional LSTM have the potential to capture extra features (Graves et al., 2013; Pascanu et al., 2013) as shown in various applications, including action recognition (Zhu et al., 2016) and vulnerable road users location predictions (Saleh et al., 2017). A bidirectional RNN/LSTM works by duplicating the recurrent network into two networks: one responsible for fitting the positive time direction (i.e. the forward states) and the other responsible for the negative time direction (i.e the backwards state) (Schuster and Paliwal, 1997). In general, the input sequence is fed as-is to the forward state and a reversed copy of the input sequence is fed to the backwards state. The bidirectional LSTM can be used in history matching problems..

**Reviewer Comment 3.3** — Training setup: There are various points around the model training setup that I see problematic. Some of them might overlap to other points mentioned above or below.

a Input features for any neural network should be normalized to zero mean, unit variance and not to the range of 0 to 1. This will basically bias your network during the start of the training in a wrong way. Maybe as some intuition: Most (all?) activation functions are centered around zero, e.g. the sigmoid function in all gates of the LSTM. With randomly initialized weights (which are normally initialized around 0), using your normalization would bias the entire network to always have pre-activations of larger than zero, and thus sigmoid values close to one. However, what you want is in expectancy to be undecided in the beginning (pre-activation of 0, equals to sigmoid of 0.5). Long story short, you should re-run all experiments with different normalizations, at least for the LSTM.

**Response**: Thank you for the comment. We have re-run the experiments using the zero mean, unit variance normalization. In comparing models trained with the original 0 to 1 scaling normalization technique against the zero mean normalization, the different normalization has a mixed result on model performance. As seen in figure 3.1 of this response, only models trained on wells 1-15 and 2-3 gain a notable improvement in performance, with 100% and 51.28% of model configurations tested on the gap lengths gaining an increase in performance, respectively. However, for wells 1-1, 1-10A, 2-2, and 2-5, only 21.15%, 39.74%, 37.82%, and 27.57% of configurations saw an improvement in performance. As such, while some model configurations do benefit (i.e. models trained on well 1-15 and 2-3), a majority of configurations tested on the four gap lengths do not improve in SpC MAPE performance when using zero mean normalization. We therefore kept our original normalization.

b Results of neural networks are generally affected by some stochasticity, because of the random weight initialization and the randomness of stochastic gradient descent. This requires almost always to train multiple models for the exact same setting with different random initialization (seeds) and to report the average model performance and variations across those repetitions. Otherwise, results might not be reproducible, since you might only be lucky (or unlucky) with your single initialization.

**Response**: We have re-run our experiment using three additional initialization seeds. Model results presented are now averaged over the initialization seeds.

New sentence P8 L18: Each model configuration was trained using four different initialization seeds and error metrics were averaged to determine the best configuration. .

c In general, you have very few data points for such a large deep learning model, as already stated above. You could either think of ways, how to combine the data of all wells in a single model, or reduce your model size drastically, which is what I would propose here.

**Response**: We have the ability to combine the data of input from neighboring wells (up to five more) for the large deep learning model, which for 4 years would be approximately 210240 data points. We have used well 1-1 as example to show the comparison of model performance between single-well and multi-well models. We trained a multi-well model using 3 wells (1-1, 1-10A, and 1-16) and compared the performance to the single-well models trained on 1-1. With additional spatial information involved, we have added the extra discussion regarding to this effort under new subsection ("4.3 Performance of multi-well DNN models") on our manuscript.

[Figure]

Figure 3.1: Comparison of models trained using scaling normalization on the data versus models trained using zero-mean, unit-variance normalization per well. Each data point represents the MAPE of a unique model configuration (size of input window, size of output window) tested on a gap length (1 hour, 24 hours, 48 hours, or 72 hours) for 2011 data. The x-axis of each plot is the natural log SpC MAPE of a unique model configuration trained on data normalized with scaling between 0 to 1. The y-axis is the natural log of the SpC MAPE of the same model configuration trained on data normalized via zero-mean. The dashed black line in each plot is the line $y = x$.

Similar to the reviewer's comments for 3.1a, we have also performed more extensive experimentation on smaller models (single LSTM layer with single dense + dropout layer) that only predicts SpC using the same model configurations as the original model (input window, output window, years of data), but limited to data from well 1-1. We compare the models against each other via gap filling on data from 2011. As seen in figure 3.2 of this response, the single LSTM performs better when filling in gaps of 1 hour for both normalization methods. However, our model architecture performs better in comparison for filling in gaps of 24, 48, and 72 hours. As such, the original model architecture is more robust in filling in larger gaps.

d I found it very hard to follow your training and testing setup, until late in the paper. E.g. around the number of possible model configurations, and total train-test combinations. I would advise to a sentence at the very beginning of the methods like "We train one model for a single well and evaluate this model on the same well and all other wells."

**Response**: Thank you for the great suggestion. We have added a sentence to the beginning to

[Figure]

[Figure]

Figure 3.2: Comparison of models trained using the original model architecture versus a single LSTM layer with a single dense layer architecture. The left plot is for models trained on data normalized by scaling between 0 and 1. The right plot is for models trained on data normalized by zero-mean. Each data point represents the MAPE of a unique model configuration (size of input window, size of output window) tested on a gap length (1 hour, 24 hours, 48 hours, or 72 hours) for 2011 data. The x-axis of each plot is the natural log SpC MAPE of a unique model configuration using the original model architecture. The y-axis is the natural log of the SpC MAPE of the same model configuration using the single LSTM architecture. The dashed black line in each plot is the 1:1 line.

further clarify our training, valuation and testing setup. Also, table 1 had been updated to clarify the three independent time periods.

New description at P8 L10-L8: "After evaluating the gain in performance improvement by using increasingly more training data (details provided in the online supplemental materials), we concluded that 4 years of training data (2012-2015) was sufficient for all the models. Validation datasets were used to select the best model hyperparameters (section 3.1.1) and the optimal combination of $M$ and $N$ (section 3.1.2) for gap filling at each well. Another independent period was selected at each well, depending on data availability, to compare the gap filling performance using the DNN and ARIMA methods. The complete set of alternatives we considered for each DNN model configuration is shown in Table 1. Excluding combinations with $M < N$, 1080 unique models (180 models per well) were trained. We used an Adam optimizer (Kingma and Ba, 2014) for training and the mean-squared error as the loss function. The models were trained for 30 iterations (i.e., epochs) over the training data."

e  Furthermore, why are models tested out-of-sample, meaning being trained on different wells than evaluated? Is there any idea behind it? Is the idea to learn a model that should be able to fill gaps in time series of any well at any location? If yes, you should probably re-think your entire training setup. If not, I don't see the need for this evaluation, since this is also not done for the ARIMA model.

   **Response**: The intent on evaluating models on wells different from the training well was to analyze how well the model does on data from a well it has not seen. However, as noted by the reviewer, this evaluation was not done for the ARIMA model. As such, we removed this evaluation in order to make the paper more straight forward and less confusing in its comparison of our DNN model and ARIMA. Furthermore, we have updated Figure 6 without the additional analysis and removed Figure 6f.

**Reviewer  Comment  3.4**  —  LSTM vs ARIMA comparison:

a  Why did you perform Hyperparameter search for the ARIMA method and not for the LSTM-based model?

   **Response**: A hyperparameter search for the ARIMA approach is performed by using the "auto.arima" function in R automatically. We also performed a hyperparameter search on the architecture of the DNN models. This includes: the number of LSTM layers, the number of units per LSTM layer, andthe number (and size of) dense layers, and activation functions, as in the subsection 3.1.1.

b  Why is ARIMA not tested on wells that are not the training well, while the LSTM is?

   **Response**: ARIMA is not tested on other wells since the ARIMA model is built dynamically based on the  168 historical hours for each well. The information carried by the ARIMA model is not enough to train other well. Also according to the comment 3e, we have removed the model evaluation on testing (which includes figure 6(f) on the non-training wells to reduce the confusion.

c P12 L6f: How was the best model decided? On training or test period? As of P13 Line 2f it seems like you picked the best model based on the test period results. If this is true, your results are biased and do not represent the true expected results of your methods. You either chose the best model by the training period, or better, have a third independent period (called validation split in machine learning) and pick your model based on the performance in this third data split, which is neither used for training nor for the final model evaluation.

**Response**: The best model for each well was decided on the data period from 2011 (now labeled as the validation period in our paper). Based on the reviewer's suggestion, we have added a third independent time period (i.e. testing period) used to compare our DNN models to the ARIMA method. So now, we have three datasets: training, validation, and testing. The training period is used to fit the model (data from 2012-2015). The validation period (year 2011 for all wells) was used to determine the optimal model configuration. The testing period is the year 2016 for comparing the model performance against the ARIMA method for all wells except 2-5 and 1-15. The testing time periods for 2-5 and 1-15 are year 2008 and 2017 respectively because there was lack of SpC observations during 2016. Table 1 has been updated to clarify training, validation and testing period.

New description at P8 L12-L15: "Validation datasets were used to select the best model hyperparameters (section 3.1.1) and the optimal combination of $M$ and $N$ (section 3.1.2) for gap filling at each well. Another independent testing period was selected at each well, depending on data availability, to compare the gap filling performance using the DNN and ARIMA methods."

**Reviewer Comment 3.5** — SpC: Later in the results section, you state that only SpC is of interest and no results for any of the other two variables are presented in this manuscript. This is totally okay, but my question is, why then do you model all three variables? Why not train the model using three inputs (temp, level and SpC) and predict only SpC?

**Response**: We performed other similar analyses for groundwater table and temperature, but they are not shown here because SpC is our primary interest for this study and also for space consideration. The reviewer is right that we don't have to predict all three variables. We experimented with a simpler architecture to predict SpC only. Those results are shown in response to comment 3.3c, which showed the simpler architecture performs better on smaller gaps, but our DNN model outperforms on larger hour gaps.

**Reviewer Comment 3.6** — P 11 L 20: "We also observe that models with a daily 24-hour input window outperform other models with longer input windows as shown in Figure 6 (c)." This statement, figure 6(c) and thus your conclusion in the following sentences and the rest of the paper are misleading. It is completely logical, that the averaged MAPE over all settings for the input sequence length of 24h is the lowest, since this only includes models, where you predicted N=1h, 6h, 12h or 24h (as of table 1: N ¡= M). And as you have seen from all other experiments, filling only small gaps is easier for any model than filling large gaps. So the fact that the 24h input sequence has the smallest error is not due to the 24h input sequence, but due to the short output sequence for M=24h inputs. I would bet that if you train a model with input length 168h and only evaluate for 1h, 6h, 12h and 24h performance should be similar/better than for a 24h input window. It is probably better to remove figure 6(c) or rethink how you can fairly compare the average results over

different input sequence length, since the different input sequence length also mean you evaluate them for different gap filling length.

**Response**: Thank you for the feedback. We re-did the analysis in Figure 6 by limiting the predicting output window sizes to 1h, 6h, 12h, and 24h for all input window sizes to provide a fair comparison. In comparing MAPEs across various input window sizes shown in Figure 6 (b), we observe that models with all input windows have comparable median MAPEs, with those of 24, 72, 144 and 168 hours leading to slightly smaller median MAPEs. We are now including all the models correspond to an input window size in boxplots rather than just showing a mean value as in the original manuscript. The input window size of 24 hours led to robust performance in terms of the fraction of small MAPEs, median and outliers on the large MAPE end. As the reviewer noted, the averaged MAPE for the model with input length of 168h, 7.33, is similar to that of the 24h input length models (6.39). In addition, we removed the original figure 6(a) for simplification and moved that analysis to the supplemental material, only showing results with 4 years of data. The updated figure and caption is shown in figure 3.3.

[Figure]

Figure 3.3: Updated Figure 6: Gap filling performance for SpC evaluated against the validation datasets under multiple model configuration parameters (a-c) or grouped by training wells (d). (a) distribution of SpC MAPE vs. tested gap lengths; (b) distribution of SpC MAPE vs. model input window size $M$; (c) distribution of SpC MAPE vs. model output window size $N$; (d) distribution of SpC MAPE aggregated by wells

New sentence P13 L12-L21: As shown in Figure 6(a), model performance deteriorates as the gap length increases, indicating that the DNN-based method tends to lose ground truth information from its input to inform prediction. In comparing MAPEs across various input window sizes shown in Figure 6 (b), we observe that models with all input windows have comparable median MAPEs, with those of 24, 72, 144 and 168 hours leading to slightly smaller median MAPEs. The 144- and 168-hour input windows

also yield lower third quartile of MAPE and fewer outliers on the larger MAPE end, indicating that the memory units in the LSTM layers are capturing important daily to weekly signatures (evident in WPS plots in Figure 2 for all wells except for Well 1-15) for some wells. As shown in 6 (c), daily and subdaily output windows yield comparable median MAPEs, with the 24-hour output window leading to smaller third quartile and fewer large MAPE outliers than its 1-, 6-, and 12-hour counterparts. Overall, an input window of 144 hours and an output windows of 24 hours appear to be a robust model configuration for all wells and gap lengths considered.

**Minor Comments:**

**Reviewer Comment 3.7** — Title: At no point of this manuscript I see the term "spatio-temporal" justified. You are only filling temporal gaps in time gaps of a single well, without any spatial input information (e.g. the input features of the neighboring wells). So I would strongly advise to change all occurrences of the spatio-temporal framing to temporal only or clearly justify what in your work is the spatial component.

**Response**: We have trained multi-well DNN models by including multiple neighboring wells to address both spatial and temporal components in our study. The additional discussion related to this effort has been added to new subsection 4.3. Please also refer to our response to comment 3.3c.

**Reviewer Comment 3.8** — P3 L4: Connor et al. (1994) is not the citation you should cite here for the RNN. Jordan (1986) would be more appropriate. Also the blog post from Olah (2015) is probably misleading here.

**Response**: We thank the reviewer for the comment. We have removed the citation of Olah (2015) and updated our citation for the RNN to Jordan (1986).

**Reviewer Comment 3.9** — P3 L11 Ma et al (2015) is definitely not the correct reference here and you should cite the original LSTM paper by Hochreiter & Schmidhuber (1997).

**Response**: We thank the reviewer for the comment. We have updated the citation and references accordingly.

**Reviewer Comment 3.10** — P3 L11f. Beside text prediction, text translation, speech recognition and image captioning, LSTMs have also already been applied to earth science and even in hydrology, which might be also/more relevant to mention here.

**Response**: On P2 Line-35, we cite papers using DL in geophysical domain. However, as implied by the reviewer, we have added a brief description of LSTMs applied to earth science and hydrology (this is the same update in response to reviewer comment 2.10)

New section (P3 L21-24): There have been applications of RNNs and LSTMs emerging in hydrology. For example, Kratzert et al. (2018) used LSTMs to predict watershed runoff from meteorological observations, Zhang et al. (2018) used LSTMs for predicting sewer overflow events from rainfall intensity and sewer water level measurements, and Fang et al. (2017) used LSTMs to predict soil moisture with high fidelity.

**Reviewer Comment 3.11** — P 4 L 2 "select" -"selected"

**Response**: Thanks for the catch. It has been modified in the revised manuscript.

**Reviewer Comment 3.12** — P5 L15: In this entire discussion you mention "highly correlated" (L19), "lower correlations" (L20), "correlates well" (L20) and many more of these statements. Such statements usually required some quantitative measures (e.g. correlation coefficient). Otherwise, what is a high correlation and what low?

**Response**: We have added the following sentences to clarify:
New sentences (P5 L13-L18):"A larger coherence at a given frequency indicates a stronger correlation at that frequency between the SpC at a well and the river stage. We consider these two variables highly correlated when the coherence is larger than 0.7 (shown in green to red colors in Coherence plots). We found that such high correlations exist at multiple frequencies, from subdaily to daily to yearly, at all the wells close to the river (e.g., 1-1, 1-10A, 2-2, and 2-3), while the higher correlation regimes in wells farther from the river (e.g., 1-15 and 2-5) are shifted towards longer periods at semi-annual and annual frequencies and less persistent in time."

**Reviewer Comment 3.13** — P5 L27 here you state you only investigate 24-, 48-, 72-h gaps. In table 1 you have much longer periods listed as well as in figure 6, while then in figure 7 again only 24, 48, 72. This is a bit inconsistent.

**Response**: We didn't include the comparison for gap length of 1 hour because both methods were highly accurate in filling such small data gaps. We added this explanation on P14 L1-2 to be consistent.

**Reviewer Comment 3.14** — P5 L23 delete "clearly"

**Response**: Agreed. It has been deleted in the manuscript.

**Reviewer Comment 3.15** — P6 L3 What you mean is not a dropout layer, but the combination of a dense layer with additional dropout. Two consecutive dropout layer would mean simply applying dropout again to the result of your previous dropout output. Correctly it would state "followed by dense layer with dropout".

**Response**: We have updated the sentence to correctly describe the model.
New sentence at P9 L3 to L4: The final DNN architecture, as shown in Figure 4, contains three LSTM layers, followed by two dense layers with dropout, a convolutional layer, and a final output dense layer

**Reviewer Comment 3.16** — This model architecture is generally described as a stacked LSTM model, given that the LSTM layers are "stacked" on top of each other." This is a tautology. Maybe simply remove this sentence or rephrase it.

**Response**: We have removed the sentence.

**Reviewer Comment 3.17** — P7 L7 "select" - "selected"

**Response**: Thanks for the catch. It has been modified in the manuscript.

**Reviewer Comment 3.18** — P7 L17 This is not called a "sigmoid neural net layer". You could say "A linear layer with sigmoid activation function". At least call it "neural network" not "neural net".

**Response**: We have been updated the sentence to say "A linear layer with a sigmoid activation function" in the manuscript.

New sentence (P11-L6): Each gate is composed of a linear layer with a sigmoid activation function.

**Reviewer Comment 3.19** — P7 L17: The pointwise multiplication is not part of the gate it-self, but how the gate is combined with the cell state.

**Response**: We have updated the sentence to distinguish the multiplication from the gate. See the previous response (3.18) for the updated sentence.

**Reviewer Comment 3.20** — P7 L18 and Fig5: all gates (f,i,o) and the cell and hidden state are vectors and should be written in lower, bold, italics letter and not capital letters

**Response**: We have updated the gate letters accordingly.

**Reviewer Comment 3.21** — P7 L 23: "Finally, an output gate (O t ) decides what to output based on the input and previous memory state. The sigmoid layer of the output gate decides what parts of the memory state will be output..." The second sentence is basically a repetition of the first. Consider rephrasing.

**Response**: The second sentence in the instruction of output gate has been removed in the revised manuscript.

**Reviewer Comment 3.22** — Table 1: Any particular reason, why you excluded 96h from the list of possible output window length, since otherwise possible input and output window length seems to be equal?

**Response**: We were not able to complete the training for all cases correspond to the output window of 96 hours within allocated computing hours. In the analyses with other output window sizes, we found the performance kept deteriorating when the output window size exceeded 24 hours. We also constrained all the analyses in the revised manuscript to not exceed 24 hours so that we have the same pool of models to compare across other parameters, see our response to comment 3.6. Therefore, our conclusions are not impacted by missing the 96hr output window, and we didn't take extra computational resources to finish training those cases.

**Reviewer Comment 3.23** — P10 L 22 How are the terms (P, D, Q)m combined into equation 2. This needs more explanation.

**Response**: Equation 2 only contains non-seasonal terms (p, d, q), i.e., autoregressive terms, nonseasonal differences and moving-average terms. The model with seasonal terms are much more complicated, so we added the equations with seasonal terms in online supplemental material with a note added to the revised manuscript on P12 L5.

**Reviewer Comment 3.24** — P11 L 19: In your setting, you always extrapolate. So this statement is not correct.

 **Response**: The statement has been removed in revised manuscript.

**Reviewer Comment 3.25** — P11 L 32: delete "very"

 **Response**: Yes. It has been deleted in the manuscript.

**Reviewer Comment 3.26** — LSTM results in general: It would be good to see only insample results at some point. How good does the LSTM perform for the same well it was trained for (as average over the 6 wells or for each well independently).

 **Response**: From the reviewers comment on 3.3e, we have redone the LSTM analysis to only include the test results for the models on the same well it was trained for, which is consistent with the training/testing performed with the ARIMA analysis.

**Reviewer Comment 3.27** — Figure 7: Missing the information that results are only for SpC.

 **Response**: We have updated the figure caption to explicitly state the results are for SpC only

**Reviewer Comment 3.28** — The point above applies to the entire section here.

 **Response**: We have updated the section to explicitly state that the results are for SpC only

**Reviewer Comment 3.29** — P12 L15: "It is noted that the optimal..." I would be cautious with such statements, unless you perform similar hyperparameter search for LSTMs as you did for ARIMA.

 **Response**: We have updated the sentence to limit the scope to our experimental runs.
   New sentence (P13 L30 - P14 L1): We observe that DNN models require less or equal input information than that required by the ARIMA method for the wells tested.

**Reviewer Comment 3.30** — P13 L 8f I do not see this in Figure 8. For me, there is no visible difference (or very hard to detect) in the Arima and LSTM error at any special frequencies. Maybe a better visualization or some quantitative measures would help.

 **Response**: We revised the description to show a better linkage between the time when larger relative errors occur and the time window when higher-frequency variations show more energy in WPS plots. Please refer to P16 L17-27: "the time windows of high relative errors are found to approximately co-locate with the time when high-frequency (daily and subdaily) signals are gaining more power. The difference between the DNN and ARIMA models tend to be amplified during those time windows. Wells 1-1, 1-10A, and 2-2 share similar seasonal patterns in WPS, with the highest intensity bin above 1024 hours across February to July. Their average WPSs all show peaks around daily and subdaily frequencies. Well 2-3 has its greatest energy between 16 to 256 hours from January to July. Well 2-5 has low intensities of variability at daily and subdaily frequencies with the low-frequency variations (monthly and seasonal) dominating the Jan to March time frame. For well 1-15, one of its strongest intensities is above 2048 hours across the entire year, and the other strong intensities are narrow bands between

16 to 256 hours. In general, both DNN and ARIMA are effective at capturing low-frequency variability (monthly and seasonal). Although DNN is more effective at capturing high-frequency (daily and subdaily) fluctuations and nonlinearities in the datasets, it may also lead to overly dynamic predictions when the training data contain more significant high-frequency signatures than the system behavior to be predicted."

**Reviewer Comment 3.31** — Figure 8. Why are the results now with the ARIMA model and 72 hour inputs and not 168 as in Figure 7?

 **Response**:
    Thank you for catching the mistake. The revised figure (Figure 9 in the revised manuscript) now shows the ARIMA models with the same parameters (input window size, output window size) that were used in the original Figure 7 (Figure 8 in the revised manuscript). These optimal ARIMA parameters are shown in the new Figure 7 to provide more information.

**Reviewer Comment 3.32** — P14 L 1 Again, I don't see the LSTM outperforming ARIMA from Figure 8 column 3. Not sure how these (also column 4) help here. Maybe it is due to my lack of understanding of the data itself, but I think some quantitative measures are better than these figures. (e.g. a table with some metrics)

 **Response**:  Please refer to our response to comment 3.30. Columns 3 and 4 were meant to show when the higher-frequency variations are gaining more power during the testing time period, which were found to colocate with the time window larger relative errors occur. Hope the more detailed explanation provided in the revise text P16 L17-27 helps with understanding. We haven't found a better way to quantitatively relate magnitude of error and the composition of flow variations.

**Reviewer Comment 3.33** — "In general, both LSTM and ARIMA are effective at capturing longer term variability, but LSTM is more effective at capturing high-frequency fluctuations and nonlinearities in the dataset." I don't see any (quantitative) evidence for such a statement.

 **Response**:  Please refer to our responses to comments 3.30 and 3.32. The purpose of the original Figure 8 (Figure 9 in revised manuscript) is to show LSTM-based DNN was able to better match the observed behavior during time windows when high-frequency fluctuations are significant. Hope the revise text explains that better.

**Reviewer Comment 3.34** — Conclusion: As of everything written above, I think the conclusions need to be entirely rewritten, including possible new results of different model configurations etc. I will not go into more detail here, since I raised many concerns above, that apply similarly to the same statements in the conclusion (e.g. LSTM and ARIMA comparisons etc). Furthermore, you miss to say for which variable you are doing gap filling (SpC only)

 **Response**:  Agreed. We have updated our conclusions section based on the additional analysis we have done, including the multi-well model, zero-mean normalization and different seeding experiments. As previously stated, we have explicitly mentioned we are analyzing gap filling results for the SpC only.

10 The performance of ~~gap filling using an LSTM framework is evaluated using test datasets with synthetic data gaps created by assuming the observations were missing for a given time window (i.e., gap length), such that the mean absolute percentage error can be calculated against true observations. Such test datasets also allow us to examine how well the original nonlinear dynamics are captured in gap-filled time series beyond the error statistics. The performance of the LSTM-based gap-filling method is compared to that of a traditional, popular gap-filling method:~~ the DNN-based gap filling method was evaluated

15 against a traditional autoregressive integrated moving average (ARIMA)  method in terms of both error statistics and capturing nonlinear, dynamic patterns in wells that exhibit various dynamics signatures. Although the ARIMA models yield better error statistics,  they fail to capture abrupt changes or high-frequency (daily and subdaily) variations in system states that are typical characteristics of a complex dynamic system. The DNN-based models excel in capturing both high-frequency

20 and low-frequency (monthly and seasonal) dynamics that are present in time series  at all wells, although they may also lead to overly dynamic predictions as guided by the training data. The DNN is shown to improve the predictive ability by taking 
[revised manuscript text omitted]

35 LSTM have the potential to capture extra features (Graves et al., 2013; Pascanu et al., 2013) as shown in various applications,

including action recognition (Zhu et al., 2016) and vulnerable road users location predictions (Saleh et al., 2017). A bidirectional RNN/LSTM works by duplicating the recurrent network into two networks: one responsible for fitting the positive time direction (i.e. the forward states) and the other responsible for the negative time direction (i.e the backwards state) (Schuster and Paliwal, 199 . In general, the input sequence is fed as-is to the forward state and a reversed copy of the input sequence is fed to the backwards state. The bidirectional LSTM can be used in history matching problems.

[revised manuscript text omitted]
  (output window length)  that follow the input window. The performance of both methods were evaluated using testing datasets for the selected wells.

**3.1     DNN Models for Gap Filling**

We designed a DNN architecture to train models of  an input size of $M$  time steps and an output size

20    of $N$ time steps to fill gaps of various lengths in groundwater well measurements. The

top of each other.  input and predicted output at a time step contain the following three  measurements from a single well or multiple wells: water level (m), temperature ($^\circ C$), and SpC ($mS/cm$), leaving the model to generalize nonlinear connections among them.  Assuming the observations from $W$ ($W \geq 1$) wells are used to fill in data gaps, the input size of the model is then $M \times 3W$. Similarly, the model output can be those three variables in the next $N$ time steps for one or more wells. Using multiple wells as input  allows the DNN model to account for both the temporal and spatial correlations in the data to improve gap-filling performance. Wells were selected based on adequate data availability and their distances from the river. While the DNN model can be used to fill in gaps in all three variables, we focused our analyses on filling gaps in SpC because of its importance to reveal river water and groundwater mixing. Same set of analyses can be performed on water level and temperature.

We explored different DNN model architectures that contain a single or multiple LSTM layers for each desired combination of $M$ and $N$ at each well under various lengths of gaps with different amount of training data (2, 4 and 6 years). Training data for the DNN models were created by finding data segments of $M + N$ hours that have no missing values, i.e., no gaps in the data, for all three measurements over a specified monitoring window. The well data were then preprocessed by normalizing all measurements to fall between 0 and 1 using different scaling factors for each variable, as temperature measurements are on a scale of $10^1$, SpC is on a scale of $10^{-1}$, and water level is on a scale of $10^2$. After evaluating the gain in performance improvement by using increasingly more training data (details provided in the online supplemental materials), we concluded that 4 years of training data (2012-2015) was sufficient for all the models. Validation datasets were used to select the best model hyperparameters (3.1.1) and the optimal combination of $M$ and $N$ (3.1.1) for gap filling at each well. Another independent testing period was selected at each well, depending on data availability, to compare the gap filling performance using the DNN and ARIMA methods. The complete set of alternatives we considered for each DNN model configuration is shown in Table 1. Excluding combinations with $M < N$, 1080 unique models (180 models per well) were trained. We used an Adam optimizer (Kingma and Ba, 2014) for training and the mean-squared error as the loss function. The models were trained for 30 iterations (i.e., epochs) over the training data. Each model configuration was trained using four different initialization seeds and error metrics were averaged to determine the best configuration.

In addition to the DNN models trained for the single-well setup, we also trained multi-well models that used observations from wells 1-1, 1-10A, and 1-16A to fill in data gaps for well 1-1. We explored the same set of configuration parameters shown in Table 1 for single-well models in multi-well models. We then compared the gap filling performance of the multi-well DNN with the single-well DNN model for well 1-1. The multi-well models were not explored for the other wells due to lack of neighboring wells in close proximity.

To evaluate the accuracy of the trained DNN models in filling SpC data gaps during the validation and testing processes, we assumed that synthetic gaps of various lengths (e.g., 1, 24, 48, and 72 hours, referred to as gap scenarios hereafter) exist in the validation or testing dataset of a well. Then a DNN model configured with input and output windows of $M$ and $N$ on a single or multiple wells is given the first $M$ hours of data from the time series preceding the occurrence of a gap (assuming no missing values in these $M$ hours) to fill in the first missing value in the gap by taking the first value of the predicted $N$

hours. This gap-filled value is then treated as if it was observed when repeating this procedure to fill in the gap of the next hour. This sliding window moves forward hour by hour until the entire gap of the data is filled. The accuracy of the gap-filling model is evaluated by calculating the mean absolute percentage error (MAPE; %) between the SpC values that are filled in (i.e., predicted) and that were observed:

$$MAPE = 100 \times \frac{1}{n} \sum_{t=1}^{n} \left| \frac{\text{Prediction} - \text{Observation}}{\text{Observation}} \right|,$$
(1)

where $n$ is the number of data points being missing.

Table 1. Parameters used in training single-well DNN models[a].

| Parameter | Values |
|---|---|
| Training wells | 1-1, 1-10A, 1-15, 2-2, 2-3, 2-5 |
| Synthetic gap length (hours) | 1, 24, 48, 72 |
| Model input window ($M$ hours) | 24, 48, 72, 96, 120, 144, 168 |
| Model output window ($N$ hours) | 1, 6, 12, 24, 48, 72, 120, 144, 168 |
| Training period | 2012-2015 |
| Validation period[b] | 2011 |
| Testing Period[c] | 2008 for well 2-5; 2017 for well 1-15; 2016 for all other wells |

[a] no models were trained for combinations with $M < N$.

[b] used to select the best DNN model configurations and hyperparameters.

[c] used to evaluate performance of DNN vs ARIMA.

**3.1.1    Hyperparameter search for the optimal DNN architecture**

We performed a hyperparameter search to explore different model architecture configurations, i.e., the number of LSTM layers, number of units per LSTM layer, number (and size of) dense layers, and activation functions. The search was performed on
10   well 1-1 only due to computational cost. We chose the optimal DNN architecture using model performance on validation data set of well 1-1 (see Table 1) using MAPE defined in Eq. (1).

The final DNN architecture, as shown in Figure 4, contains three LSTM layers, followed by two dense layers with dropout, a convolutional layer, and a final output dense layer. Stacking three layers of LSTM was found to yield better performance than a one- or two-layer architecture. Each of the three LSTM layers has 128 units  because this configuration outperformed

others with more or fewer number of units. The output from the last LSTM layer  of size $M$ x 128  is fed into two consecutive dense layers  where every input neuron is connected to every output neuron with a weight matrix and bias vector.  The output from the second dense layer of size $M$ x 64 is fed into a convolutional layer with 24 filters of size $M$-$N$+1, reducing the output size to $N$ x 24. Finally, a dense layer is applied to yield a model output of our desired size,  $N \times 3$ for a single well or multiple of that when the model is designed to fill in data gaps in multiple wells. The detailed structures of the LSTM layers, dense layer, and convolutional layer are provided in the supplemental material. Dropout, i.e., randomly disables a selected fraction of neurons, was used in the dense layers as the regularization technique to enhance robust model performance and prevent overfitting (Hinton et al., 2012). We adopted a dropout rate of 0.3 after testing a set of alternatives (0, 0.1, 0.2, 0.3 and 0.4).

[revised manuscript text omitted]

30   evaluated using the same MAPE metric shown in Eq. (1) during the testing period listed in Table 1 and compared with the DNN-based methods.

**4 Results and Discussion**

**4.1 Performance of single-well DNN models**

 We evaluated the accuracy of  DNN models in filling gaps of various lengths of the SpC measurements in the year 2011 following the steps described in section  3.1.1. MAPEs were
5  summarized in boxplots for different DNN model configuration parameters, as shown in Figure 6. Each MAPE boxplot was drawn from a group of models with one parameter (corresponding to each x-axis) fixed at the given value while all the other parameters, including  the training wells and gap scenarios, cycle through their possible combinations.
10  ~~be seen in Figure 6 (a), using more training data improves the model performance on average, consistent with a standard observation in machine learning applications that model performance is highly dependent on the amount of training data available. When the amount of training data increases from 2 to 4 and 6 years, the MAPE drops slightly with consistent variability across all combinations. We therefore conclude that 2 years of data is sufficient to train the LSTM models we need for gap filling, although 4 years of training data would be better~~Although we attempted to train models with output window
15  sizes greater than 24 hours, these models performed noticeably worse than those with output windows less than or equal to 24 hours (results shown in the online supplemental materials). Thus, our analyses here focus on models with output window less than or equal to 24 hours.

 As shown in Figure 6 (a), model performance deteriorates as the gap length increases, indicating that the DNN-based method tends to lose ground truth information from its
20  input to  inform prediction. In comparing MAPEs across various input window sizes shown in Figure 6 (b), we observe that models with all input windows have comparable median MAPEs, with those of 24, 72, 144 and 168 hours leading to slightly smaller median MAPEs. The 144- and 168-hour input windows also yield lower third quartile of MAPE and fewer outliers on the larger MAPE end, indicating that the memory units
25  in the LSTM layers are capturing important daily to weekly signatures (evident in WPS plots in Figure 2 for all wells except for Well 1-15) for some wells. As shown in 6 (c) daily and subdaily output windows yield comparable median MAPEs, with the 24-hour
30  output window leading to smaller third quartile and fewer large MAPE outliers than its 1-, 6-, and  12-hour counterparts. Overall, an input window of 144 hours and an output windows of 24 hours appear to be a robust model configuration for all wells and gap lengths considered.

 The performance of  single-well DNN models varied among the wells as shown in Figure 6 ( d). The DNN models for well 1-15  lead the performance with the smallest MAPEs, while those for well 2-2 is grouped by wells being tested, as shown in Figure 6 (f), all models can perform very well in filling in gaps for well 1-15 and reasonably well for,, while all models appear to have difficulty in filling gaps for well 2-2. When selecting the optimal LSTM configuration of 24-hour input window and 12-hour output window using 4 years of training data, the models trained on well 2-3 perform the best in filling gaps of various lengths for all the 6 wells.~~ performed comparably overall, with more large MAPE outliers for well 1-10A.

[Figure]

**Figure 6.** Gap filling performance for SpC evaluated against the validation datasets under multiple model configuration parameters (a-c) or grouped by training  wells (d). (a)  distribution of  SpC MAPE vs. tested gap lengths; (b)  distribution of SpC MAPE vs. model input window size $M$; (c)  distribution of SpC MAPE vs. model output window size $N$; (d)  distribution of SpC MAPE aggregated by wells95% confidence intervals of the averaged MAPE value are shown in shaded area in plots (a) -(d) and as the error bars in (e) and (f).

**4.2     Single-well DNN and  ARIMA comparisons**

 The single-well DNN-based gap filling approach was compared to the ARIMA approach using relative errors calculated for each data point that  was assumed to be missing in the testing data by setting n=1 in Eq. (1)  for MAPE. Best model configurations determined on the validation dataset (i.e., data from year 2011), as described in sections 3.1.1 and 3.2, were used in comparing the two approaches. Figure 7 illustrates the input and output windows selected as the best model configurations  for DNN and ARIMA methods. The output window $N$ of an ARIMA model is the same as the length of the gap it is built to fill. We observe that DNN models require less or equal input information than that required by the ARIMA method for the wells tested. None of the optimal output window sizes exceeds 24 hours for the DNN models. We only compared two methods for gap lengths of 24, 48 and 72 hours because both methods were highly accurate in filling small gaps such as one hour.

[Figure]

**Figure 7.** Best input and output windows for DNN and ARIMA models for filling gaps of various lengths at each well.

Figure 8 shows the interquartile ranges of relative errors under different gap lengths for all individual wells, each bounded by its 25th to 75th percentiles. Relative errors were used to show overestimations or underestimations by both approaches. The horizontal dotted lines represent the ±5% relative error range  that are typical measurement errors of the SpC sensors deployed at the site.  The ARIMA models tend to perform better than the

 DNN models in terms of error statistics. For both approaches, the relative errors increase as the gap length increases as expected, especially so for well 1-1 when the gap lengths are 48 and 72 hours. While all the relative errors yielded from the ARIMA method are within the $\pm5\%$ measurement error range, there are a few cases using DNN leading to relative errors outside the typical observational error range with longer gaps for wells 1-1, 1-10A and 2-2. The relative errors in the ARIMA

5 models tend to distribute symmetrically on both sides of 0%, whereas errors in the  DNN models appear to skew toward the negative side for  all wells except for well 2-5. For well 1-15, the relative errors for all three gap lengths are very close to 0.  Wells 1-1 and 2-2  have larger relative errors over the testing

10  period using both approaches.

[Figure]

**Figure 8.** Summary of SpC relative errors for filling gaps of various lengths (i.e., 24, 48, and 72 hours) for best  DNN and ARIMA models tested for each well.

In addition to the error statistics, it is also important to examine how well a gap-filling method  captures the

15 desired dynamic patterns in the gap-filled time series. Therefore, the SpC time series reproduced by the gap-filling methods  during the testing period (2016 for wells 1-1, 1-10A, 2-2, 2-3; 2017 for well 1-15; 2008 for well 2-5) with 24-hour synthetic gaps are evaluated against the real time series. Model configurations are the same as those used in error

statistics comparison (Figure 7). As shown in the first and second columns of Figure 9, the ARIMA approach (column 1) can capture the smooth changes in the observations but not abrupt changes that occur over a short time window (i.e., at higher frequency); these occur in all wells except 1-15. The spikes in errors during those rapid changes were not captured in Figure 7 as they are outside of the interquartile ranges of the relative errors. This is an indication that ARIMA fails to capture  high-frequency (daily and subdaily) dynamics and nonlinear trends despite having smaller  error quartiles. The DNN approach, on the other hand, is able to better  capture such dynamics in some wells (e.g., wells 1-15, 2-3 and 2-5). However, the DNN approach appears to overestimate the high-frequency (daily and subdaily) fluctuations in some wells near the river (i.e., wells 1-1, 1-10A, and 2-2), which contributed to less desirable relative errors distributed between the first and third quartiles (7). This is likely caused by the variability in dynamics signatures among the training, validation and test periods. For well 1-15, which  exhibits less dynamic behavior  in SpC, both gap filling methods  perform well in terms of both the relative errors and capturing  the dynamic patterns.

To further investigate how the relative performance of  the two gap-filling methods depends on the inherent dynamics in each time series,  spectral analyses for the testing SpC  datasets were performed using the same wavelet decomposition method for the multi-year analyses (shown earlier in Figure 2). As shown in Figure 9, the time windows of high relative errors are found to approximately co-locate with the time when high-frequency (daily and subdaily) signals are gaining more power. The  difference between the DNN and ARIMA models tend to be amplified during those time windows. Wells 1-1, 1-10A,  and 2-2 share similar seasonal patterns in WPS, with the highest intensity bin above 1024 hours  across February to July. Their average WPSs all show peaks around daily and subdaily frequencies. Well 2-3 has its greatest energy between 16 to 256 hours from January to July. Well 2-5 has low intensities of variability at daily and subdaily frequencies with the low-frequency variations (monthly and seasonal) dominating the Jan to March time frame. For well 1-15,  one of its strongest intensities is above 2048 hours across the entire year, and the other strong intensities are narrow bands between 16 to 256 hours. In general, both  DNN and ARIMA are effective at capturing  low-frequency variability (monthly and seasonal). Although DNN is more effective at capturing high-frequency (daily and subdaily) fluctuations and nonlinearities in the datasets, it may also lead to overly dynamic predictions when the training data contain more significant high-frequency signatures than the system behavior to be predicted.

There is also significant difference in computational cost between the DNN and ARIMA methods for gap filling. ARIMA requires very little computational resources: the auto.arima function in R requires approximately 40 seconds for fit and

validate a model using one year of data on a personal computer with a 3.00 GHz CPU. Conversely, training and  validating a single DNN model takes approximately 20-30 minutes on dual NVIDIA P100 12GB PCI-e based GPUs.

**4.3 Performance of multi-well DNN models**

5    We evaluated the predictive ability of the multi-well DNN models in filling gaps of various lengths in the SpC data at well 1-1 by comparing the performance against their single-well counterparts. Well 1-1 was chosen because of data availability in nearby wells (wells 1-10A and 1-16A). Moreover, both the ARIMA and single-well DNN methods had difficulty in capturing its dynamic patterns as discussed in Section 4.2. Similar to the single-well DNN model for well 1-1, the multi-well DNN models also predict the three variables for the well 1-1 only. We adopted the same DNN architecture from the single-well models and

10  trained the same set of alternatives considering input window sizes and output window sizes for various gap lengths as listed in Table 1. Only output window sizes smaller than 24 hours were considered as learnt from the single-well models. The same training and validation periods were adopted to select the optimal combination of $M$ and $N$. Results were summarized in Figure 10, where each boxplot was generated in the same manner as in Figure 6.
   Compared to the single-well DNN models, the multi-well DNN models significantly improve the gap filling accuracy at well

15  1-1 with longer gaps (48 and 72 hours) while perform comparably with smaller gaps (i.e., 1 and  24 hours), as shown in Figure 10 (a). The multi-well DNN models reduce the fraction of larger MAPEs under all the input window sizes (figure 10 (b)) and all output windows (figure 10 (c)). Further one to one comparisons between different $M$ and  $N$ combinations are provided in figure 11 under different gap scenarios. Natural log scales were on both axes for better separations in data points. All the points below the 1:1 line represent cases where a multi-well DNN outperforms a single-well

20  DNN. The percentage of points below the 1:1 line increases with the gap lengths: 17.9%, 35.7%, 64.3%, and 82.1% for gaps of 1, 24, 48, and 72 hours, respectively. Therefore, including spatial information from neighbouring wells could potentially increase the chance of successes in filling gaps under more challenging circumstances, such as more complex dynamic patterns and longer data gaps.

**5   Conclusion**

25  In this study, we implemented  a DNN-based gap filling method to account for spatio-temporal correlations in  monitoring data. We extensively evaluate the new method in filling data gaps in SpC measurements that are often used to indicate groundwater and river water interactions along river corridors. We optimized a DNN architecture that contains stacked LSTM, convolutional, and dense layers to take advantage of a 10-year spatially distributed multi-variable time series dataset collected by a groundwater monitoring well network

30   for filling SpC data gaps. A primary advantage of using DNN is the ability to incorporate spatio-temporal correlations and

nonlinearity in model states without assuming an explicit form of correlations or nonlinear functions in advancing system states as a priori. We compared the performance of single-well DNN-based gap-filling method  with a traditional gap-filling method, ARIMA, to evaluate how well a DNN can capture multi-frequency dynamics. We also trained DNN models that take input from multiple wells to predict responses at one well. The multi-well DNN models were compared with single-well models to assess the improvement in gap filling performance by including additional spatial correlation from neighboring wells.

In general, both  DNN and ARIMA were highly accurate in filling small data gaps (i.e., 1 hour). They were reasonably effective at filling in gaps of 24, 48, and 72 hours. The relative errors  were mostly within the range of instrument measurement error.  Both models captured the long-term trends in data  (i.e., low-frequency variations at the monthly or seasonal time scales), except during some time windows with highly dynamic fluctuations.  The ARIMA method was found to be suitable for time series with less dynamic behavior.  DNNs excel in dealing with high-frequency dynamics (daily and subdaily) or nonlinearities, although they  require more training data and computational  resources. The DNN approach also appeared to overestimate the high-frequency (daily and subdaily) fluctuations in some wells near the river (i.e., wells 1-1, 1-10A, and 2-2), which was likely caused by the variability in dynamics signatures among the training, validation and test periods. Availability of sufficient training data is critical for the success of  DNN-based methods, as is with any DNN-based learning method. Extrapolating the DNN models to conditions beyond those in the training data remains as a major challenge.

Wavelet analysis could provide useful insights to the dynamic signatures of the data and the change in composition of their important frequencies over time, which can serve as a prior basis for selecting an appropriate gap-filling method. For example, the ARIMA method would work well if the dynamics are dominated by seasonal cycles, while more sophisticated approaches like  DNN-based methods could work better if there is evidence of weekly, daily and subdaily fluctuations.  Depending on the mixture of  high- and low-frequency variability inherent in the time series, different  DNN architecture and configurations can be  explored and evaluated  through hyperparameter searches with respect to LSTM layers, dense layers and activation functions to achieve better performance in capturing more complex dynamics.  We also demonstrated that incorporating spatial information from neighboring stations in DNN could contribute to performance improvement under challenging scenarios with dynamic system behaviours with longer data gaps (multiple days). The optimal DNN model configuration and performance that could be achieved would vary case by case. The bidirectional LSTM can be explored to formulate the gap filling as a history matching problem and evaluate the value of observed data in future time window relative to missing data for filling the data gaps. 
[revised manuscript text omitted]

Pascanu, R., Gulcehre, C., Cho, K., and Bengio, Y.: How to Construct Deep Recurrent Neural Networks, 2013.

Pfeifer, P. E. and Deutrch, S. J.: A three-stage iterative procedure for space-time modeling phillip, Technometrics, 22, 35–47, 1980.

Reichstein, M., Camps-Valls, G., Stevens, B., Jung, M., Denzler, J., Carvalhais, N., and Prabhat: Deep learning and process understanding for data-driven Earth system science, Nature, 566, 195–204, https://doi.org/10.1038/s41586-019-0912-1, http://www.nature.com/articles/s41586-019-0912-1, 2019.

Saleh, K., Hossny, M., and Nahavandi, S.: Intent prediction of vulnerable road users from motion trajectories using stacked LSTM network, in: 2017 IEEE 20th International Conference on Intelligent Transportation Systems (ITSC), pp. 327–332, https://doi.org/10.1109/ITSC.2017.8317941, 2017.

Schmidhuber, J.: Deep learning in neural networks: An overview, Neural Networks, 61, 85 – 117, https://doi.org/https://doi.org/10.1016/j.neunet.2014.09.003, http://www.sciencedirect.com/science/article/pii/S0893608014002135, 2015.

Schuster, M. and Paliwal, K. K.: Bidirectional recurrent neural networks, IEEE Transactions on Signal Processing, 45, 2673–2681, 1997.

[revised manuscript text omitted]

**Figure 9.** Columns 1 and 2 show time series of model predictions (in red) from ARIMA and DNN methods, respectively, assuming 24-hour synthetic gap in the SpC data, compared with observations (in black) and the relative errors (in blue). The best model configurations were used for all models. The testing data come from year 2016 for wells 1-1, 1-10A, 2-2, and 2-3, from year 2017 for well 1-15 and from 2008 for well 2-5. Column 3 is the spectrogram of each well and column 4 is the WPS averaged over for the corresponding year.

[Figure]

**Figure 10.** Comparing performance between single-well (well 1-1) and multi-well DNN models (wells 1-1,  1-10A and 1-16A) for filling  SpC data gaps for well 1-1 during the testing period (year 2011). (a) distribution of SpC MAPE vs.  tested gap lengths; (b) distribution of SpC MAPE vs. model  input  window size $M$; (c) distribution of SpC MAPE vs.  model  output window size $N$; (d) distribution  of  SpC MAPE aggregated by wells used to train the models

[Figure]

**Figure 11.** Comparing performance between single-well (well 1-1) and multi-well DNN models (wells 1-1, 1-10A and 1-16A) for filling SpC data gaps of various lengths for well 1-1 during the testing period (year 2011). The subplots (a)-(d) correspond to a gap length of 1, 24, 48 and 72 hours, respectively. Each data point represents a unique model configuration (size of input window, size of output window). The x-axis and y-axis are the natural log of SpC MAPE for single-well and multi-well DNN models, respectively. The dashed black line in each plot is the 1:1 line.

---

## Referee Report (RR1)

As in my previous review, I will mostly concentrate on the methodology of this manuscript, than on the data or context of this study in groundwater literature.

I think that the revised manuscript overall improved by a lot over the original submission in terms of clarity and structure. The authors included information on the hyperparameter search (and made the search space larger than before) and chose the best model based on independent validation data. Although I am surprised by the result of the hyperparameter search, I guess this is something we have to accept.

However, there is one point around the model architecture that I still do not see satisfiably answered (and largely ignored in the manuscript), see Comment 1 + Comment 2.

Comment 1:
In my last review (Reviewer Comment 3.1d), I questioned the use of the convolutional layer. As the layer is used currently, the most recent time steps are ignored for doing the gap filling. To be more concrete: Given an input sequence of length M with N consecutive time steps, which should be gap filled. The first of the N time steps is immediately following the last time step of M. In most autoregressive tasks, the immediate preceding time steps are the most important features, especially with time series of high temporal frequencies. However, due to the choice of the convolutional layer and the filter size, the first day of N is only predicted by the M-N+1 first time steps of M, ignoring probably the most important information. The ARIMA model however, does see these time steps (and performs much better than the LSTM). From the hyperparameter search as described in Section 3.1.1. It does not seem as if the convolution layer at all was optimized. And even if the authors decide to keep this architecture, I think this is a critical point to include into the paper and to explain their decision. I could imagine that people who see this (that the most recent days are ignored in an autoregressive task) will ask why. The answer of the authors in their rebuttal ("*Furthermore, the time steps immediately preceding the current time are not necessarily the most informative information in the presence of dynamical behavior*") might be true, but should definitely be tested as well as discussed in the manuscript.

Comment 2:
This is very related to the comment above: The authors argued in their answer to Reviewer Comment 3.1.d that the (one) reason for the convolutional layer is to map from a sequence with M time steps to a sequence of N time steps. I don't know how this slipped my eye in my previous review, but an important question is "Why do you even map to N?".
On Page 9 L1ff. you say you actually only map M to 1 and, then move M by one time step (integrating the last prediction into the shifted input sequence M) to predict the next time step and so one. So why is the LST-based model not trained to do exactly this? This setting is the most common LSTM setting (called sequence-to-one), and you would simply use the LSTM output at the last time step, to predict the next time step. During inference (= gap filling) you would do exactly what you do now: passing one sequence of M time steps through the model, get the prediction for the M+1 time step, shift M by one time step and include the previous prediction, pass the new sequence again to get the prediction for the M+2 time step, and so on.

The convolutional filter is also not, what makes you model account for spatial correlations (related to the answer of the authors to reviewer comment #1.4), since the LSTM can already account for those correlations. So the framing of the manuscript can remain unchanged.

Comment 3:
I can not follow the conclusion in L7 P 15ff, especially that "*ARIMA cap capture [...] but not changes that occur over a short time window (i.e., at higher frequencies)*". As the authors note themselves, ARIMA is better in every error statistic. It is argued that the LSTM does better at higher frequencies and it is pointed to Figure 9, the first two columns. The figures are small so it might be hard to see, but from what I can see, I don't see the LSTM being better in any well at any point during the entire period. The blue line, which shows the relative error, seems to be always worse for the LSTM, also during periods with higher variance. At this point, I can't see any evidence that backs the statement of the authors and I think, additionally to these plots, some quantification (using some metrics) are needed to support the statement that the LSTM has some advantage over the ARIMA model.

Comment 4:
P18 L11: "significantly" I agree that the improvement seems obvious, however, the use of significant should always be supported by the result of a significance test. Otherwise, maybe rephrase this sentence.

Comment 5:
Isn't it possible to train a multi-well ARIMA(X) model as well? This would be an interesting benchmark for the experiment in Section 4.3, since in the single site the ARIMA model showed superior performance. If the LSTM would be better in the multi-well setting, this would certainly be an interesting result.

Comment 6:
The two sentences in P19 L3ff seem to contradict each other. "*DNNs excel in dealing with high-frequency dynamics (daily and subdaily) or nonlinearities, although they require more training data and computational resources. The DNN approach also appeared to overestimate the high-frequency (daily and subdaily) fluctuations in some wells near the river (i.e., wells 1-1, 1-10A, and 2-2), which was likely caused by the variability in dynamics signatures among the training, validation and test periods.*". They "*excel*" but "*also appear[ed] to overestimate*".

---

## Referee Report (RR2)

**3rd Revision of "Technical note: Using Long Short-term Memory Models to Fill Data Gaps in Hydrological Monitoring Networks" by Ren et al.**

I am happy to see that the authors tested a simpler model architecture, as suggested in previous reviews. The paper has become shorter and more concise, which is great. It is worth remembering that this is intended as a technical note, rather than a full paper, but there are still some open questions from my side.

1.  Conceptually, I don't understand why the manuscript is framed as gap filling, when the setup of the model matches the common setup of forecasting (i.e. having historic observations + additional inputs to predict the next time step). From my point of view, gap filling is a task that is performed on historic data records where observations of both sides of the gap are available. This raises the question why you would not make use of this additional information, since interpolating between two points is most likely easier than predicting into the future. If the framing of the manuscript should indeed be focused around gap filling, I think it might be worth discussing the decision for running a gap filling model in this forecasting setup.
2.  Why are ARIMA and LSTM not treated equally? That is, why does the ARIMA model predict all missing values at once, while the LSTM predicts only one time step at a time (and is then re-run for the next time step with the previous prediction filled into the input sequence). I think both models could be set up similarly and I wonder if you ever tested this and if the results suggested that this is the optimal setting for both models.
3.  Looking at Figure 6: If you perform hyper parameter tuning and one model (almost) constantly picks the largest (or smallest) value, usually you should increase the search range, as this indicates that eventually even larger (or smaller) parameters would be better. Looking at e.g. the input window length of the ARIMA model.
4.  To my understanding, part of this technical note is the benchmarking/comparison of two models, LSTMs and ARIMA. As such, I think this paper is still missing a statistical analysis of the modeling results. The entire discussion is currently focussed around a few plots and a textual description of what one can see in these figures. However, to a certain degree I would argue that this analysis/interpretation is rather subjective. I think it would benefit the paper to have a table that compares both models on a range of different metrics, including statistical tests of e.g. the robustness/significance of the results. Right now, I wonder what the takeaway message of this paper is. I would argue that it was probably known that both models, LSTMs and ARIMA, are generally capable of time series forecasting. If I would be a user with similar data or a similar problem, what is the additional knowledge that I can gain from reading this paper?

Minor comments:

- Figure 4: I think this figure is misleading to someone unfamiliar with LSTMs. You actually drew a fully connected network, rather than a recurrent (sequential) neural network. As of now, it seems like all inputs go into the LSTM at once (no time steps are visible in this figure), and the outputs of all time steps (since on the left side the timesteps are top to bottom) are used to predict the output. Figure (b) is actually the more correct depiction of the LSTM and I don't understand why both visualizations are needed.

- P 3, L 32 "*at the 300 Area of the U.S. Department of Energy Hanford site*". What is the 300 for?

- P 8, L 25: "*The well data were then pre-processed by normalizing all measurements via zero-mean and unit variance for each variable*". You do not normalize *"via"* zero-mean and unit variance, you rather normalize *to* zero mean and unit variance. Normalizing via zero-mean and unit-variance sounds like you subtract a mean of zero and divide by a variance of one, which is hopefully not what you have done.

- P9 L 7ff You use plural for "multi-well models" throughout this passage but you only trained one multi-well model to predict at well 1-1, or?

- P17 L 17: Which "auto.arima" function?

---

## Author Response (AR2)

**Response to referee comments**

**Reviewer 1**

**General note:** The updated paper is has improved, and I happy with the answers to the comments and most of the changes performed.

However, on the abstract mainly, I still suggest some minor updates to reflect clearly and better the work. In general, there are so many places where the author misleads readers with terms like dynamic data, dynamic parameter, dynamic high-frequency data, complex dynamic, highly challenging situations, the great advantages, multiple dominant modes and others. It is challenging to follow with all these overused terms.

The source of the dynamics that lead to the high variability in the groundwater levels is not known, and therefore, trying to fill in data is indeed a challenge for research. There is not needed to oversell the complexity, and instead of helping the paper, it makes it cumbersome to read and very conflictive for people in the area. Terminology should be correctly used in a scientific publication.

**Response**: We appreciate the reviewer's overall positive assessment of our revised manuscript and the suggestion to improve the abstract. We have edited the abstract to make the important messages clear without overselling the complexity as the reviewer suggested. Please also see below our one-to-one responses to minor comments in the abstract for details.

**Minor comments are in the abstract**

**Reviewer Comment 1.1** — What are data set with high-frequency dynamics? I believe you want to highlight the dynamics of a phenomenon reflected in data? I am not sure to understand the high-frequency term here?

**Response**: We reworded this to be more specific: This 10-year-long dataset contains hourly temperature, specific conductance, and groundwater table elevation measurements from 42 wells with various lengths of gaps.

**Reviewer Comment 1.2** — The sentence: "We selected a location at the U.S. Department of Energy's Hanford site to demonstrate and evaluate the new method" not sure which new method?

**Response**: The new method here refers to the LSTM-based gap filling method. The abstract has been extensively revised to address various comments, and the phrase "new method" has been removed.

**Reviewer Comment 1.3** — It is well known that wells data usually are spatially distributed (no two wells in the same place), so in many studies, the research address them as in a more precise way just saying temporal data from 42 wells.

**Response**: We have revised this sentence to be "This 10-year-long dataset contains hourly temperature, specific conductance, and groundwater table elevation measurements from 42 wells with various lengths of gaps. "

**Reviewer Comment 1.4** — The sentence: "that monitor the dynamic and heterogeneous hydrologic exchanges between ". I would suggest making it more clear. Same as in point "a" of this comments.

**Response**: The long sentence has been rewritten to make it clearer. New Sentences:" In this study, we explore the ability of deep neural networks to fill in gaps in a spatially distributed time-series dataset from a well network deployed at the U.S. Department of Energy's Hanford site that monitors the dynamic and heterogeneous hydrologic exchanges between the Columbia River and its adjacent groundwater aquifer. This 10-year-long dataset contains hourly temperature, specific conductance, and groundwater table elevation measurements from 42 wells with various lengths of gaps"

**Reviewer Comment 1.5** — The sentence: "capturing nonlinear, dynamic patterns in wells that exhibit various dynamics signatures", it seems trying to oversell the work. Dynamics everywhere?

**Response**: We added "river corridor" before the "wells" to refine the scope in river corridor where the dynamics are present in all wells. We also changed "nonlinear, dynamic patterns" to "temporal patterns".

**Reviewer Comment 1.6** — The sentence: "Although the ARIMA models yield better error statistics, they fail to capture abrupt changes or high-frequency (daily and subdaily) variations in system states that are typical characteristics of a complex dynamic system." I guess there was a mathematical way to verify this statement? Maybe would be nice to elaborate in this line if this was just visual, or what was used to assess the so-called high-frequency variations?

**Response**: The high-frequency (hourly and subdaily) variations are assessed from the WPS spectrum plots in the second column of Figure 7. We have explained that concept in multiple places in the manuscript, and the sentence has been revised to "Our study demonstrates that the ARIMA models yield better average error statistics, yet they tend to have larger errors during time windows with abrupt changes or high-frequency (daily and subdaily) variations.".

**Reviewer 2**

**General note:** As in my previous review, I will mostly concentrate on the methodology of this manuscript, than on the data or context of this study in groundwater literature. I think that the revised manuscript overall improved by a lot over the original submission in terms of clarity and structure. The authors included information on the hyperparameter search (and made the search space larger than before) and chose the best model based on independent validation data. Although I am surprised by the result of the hyperparameter search, I guess this is something we have to accept. However, there is one point around the model architecture that I still do not see satisfiably answered (and largely ignored in the manuscript), see Comment 1 +Comment 2.

**Response**: We thank the reviewer for the constructive comments to improve our manuscript and for the overall positive assessment on our revision. We have taken extensive work to address the comments on the convolutional layer and the overall LSTM architecture. We have achieved improved performance with a much simpler architecture: a single-layer LSTM model without the convolutional layer. We provide the details in the one-to-one responses to your detailed comments below.

**Reviewer Comment 2.1** — In my last review (Reviewer Comment 3.1d), I questioned the use of the convolutional layer. As the layer is used currently, the most recent time steps are ignored for doing the gap filling. To be more concrete: Given an input sequence of length M with N consecutive time steps, which should be gap filled. The first of the N time steps is immediately following the last time step of M. In most autoregressive tasks, the immediate preceding time steps are the most important features, especially with time series of high temporal frequencies. However, due to the choice of the convolutional layer and the filter size, the first day of N is only predicted by the M-N+1 first time steps of M, ignoring probably the most important information. The ARIMA model however, does see these time steps (and performs much better than the LSTM). From the hyperparameter search as described in Section 3.1.1. It does not seem as if the convolution layer at all was optimized. And even if the authors decide to keep this architecture, I think this is a critical point to include into the paper and to explain their decision. I could imagine that people who see this (that the most recent days are ignored in an autoregressive task) will ask why. The answer of the authors in their rebuttal ("Furthermore, the time steps immediately preceding the current time are not necessarily the most informative information in the presence of dynamical behavior ") might be true, but should definitely be tested as well as discussed in the manuscript.

**Response:**

We agree with the reviewer that the application of the convolutional layer limits the use of most recent time steps in predicting the future states. As a result and based on the Reviewer comment 2.2 (and previous comments), we have updated the model to a single LSTM layer and a dense layer to forecast N=1 time step immediately following the input time window. This removes the convolutional layer and ensures the model is taking advantage of the most recent time steps. With the simplified model, we have updated the hyperparameter search to include the number of LSTM units and the learning rate. In addition, we drop the temperature measurement as input. Please refer to the new Figure 4 in the manuscript for the updated architecture.

We compared the performance between the best models using the original and the new simplified architectures. Figure 2.1 and Figure 2.2 show the model performance comparisons in different metrics between the two LSTM architectures on the validation dataset from year 2011 for single-well models by setting N=1. As seen in Figure 2.1, the single-layer LSTM models outperform the original architecture on MAPE for all wells except 1-15, where both perform comparably on gaps of 24, 48, and 72. We see a similar improvement in relative error distributions (Figure 2.2).

Figure 2.1: Comparison of the 2011 SpC MAPE scores of the best single LSTM model per gaplength of each well against the best original model per gap-length. The results for the new single LSTM are shown in red, and the results for the original architecture are shown in green.

Additionally, we trained multi-well models with the simplified architecture using the same procedure described in section 4.3 and compared them to the single-well modified architectures. Similar to the single-well models, the new multi-well models perform better than the original architecture across all gap-lengths by MAPE (Figure 2.3) and relative error distribution (Figure 2.4).

We modified the description on hyperparameter search accordingly in the manuscript under section 3.1.1.

**Reviewer Comment 2.2** — This is very related to the comment above: The authors argued in their answer to Reviewer Comment 3.1.d that the (one) reason for the convolutional layer is to map from a sequence with M time steps to a sequence of N time steps. I don't know how this slipped my eye in my previous review, but an important question is "Why do you even map to N?". On Page 9 L1ff. you say you actually only map M to 1 and, then move M by one time step (integrating the last prediction into the shifted input sequence M) to predict the next time step and so one. So why is the LSTM-based model not trained to do exactly this? This setting is the most common LSTM setting (called sequence-to-one), and you would simply use the LSTM output at the last time step, to predict the next time step. During inference (= gap filling) you would do exactly what you do now: passing one sequence of M time steps through the model, get the prediction for the M+1 time

---

## Author Response (AR3)

**Response to referee comments**

**Reviewer 1**

I am happy to see that the authors tested a simpler model architecture, as suggested in previous reviews. The paper has become shorter and more concise, which is great. It is worth remembering that this is intended as a technical note, rather than a full paper, but there are still some open questions from my side.

**Reviewer Comment 1.1** — Conceptually, I don't understand why the manuscript is framed as gap filling, when the setup of the model matches the common setup of forecasting (i.e. having historic observations + additional inputs to predict the next time step). From my point of view, gap filling is a task that is performed on historic data records where observations of both sides of the gap are available. This raises the question why you would not make use of this additional information, since interpolating between two points is most likely easier than predicting into the future. If the framing of the manuscript should indeed be focused around gap filling, I think it might be worth discussing the decision for running a gap filling model in this forecasting setup.

**Response**:
    We thank the reviewer for the positive assessment on the work we have done to improve the manuscript in our last revision. We framed our manuscript as gap filling because that is the purpose of our work. We recognize that multiple approaches can be applied for this purpose using the information before the gaps only or involving data from both sides of the gaps, as we described in the introduction, including interpolation, extrapolation, regression, predictive models and so on. In this technical report, we explore a forward forecasting approach for filling in the data gaps for a fair comparison between the LSTM models and the ARIMA models because ARIMA does not use observations after the gap for its estimations. We explicitly state this in the introduction as suggested by a reviewer in previous revisions.
    We also recognize that Bi-directional LSTMs could be used to frame the gap filling as an interpolation problem as we discussed in the conclusion as a potential research topic for the future because non-trivial effort is needed to build, train and evaluate the bi-directional LSTMs.

**Reviewer Comment 1.2** — 2. Why are ARIMA and LSTM not treated equally? That is, why does the ARIMA model predict all missing values at once, while the LSTM predicts only one time step at a time (and is then re-run for the next time step with the previous prediction filled into the input sequence). I think both models could be set up similarly and I wonder if you ever tested this and if the results suggested that this is the optimal setting for both models.

**Response**: We have revised the ARIMA predictive settings to match those of LSTM models for both single-well and multi-well models by following reviewer's suggestions, i.e., they predict the next time step immediately after the input window and the input window is sliding hour-by-hour to fill the entire length of a gap. We also searched a wider range of input windows for the ARIMA models for optimal performance per reviewer's another comment below. Please see the details for the response to comment 1.3. These changes did lead to slightly improved performance for ARIMA in terms of reducing the relative error outliers although all the conclusions on relative performance between LSTM and ARIMA stay unchanged. The new results have been updated for Figures 7, 8, 9 and 10 in the revision.

**Response to referee comments**

**Reviewer 2**

The paper explores in relative novel tools from Deep Learning LSTM models for reconstructing time series data. The potential of the paper describes well a setup and have improved significantly from the previous version. Now, it is more clear the problem, motivation and complexity in the modelling system presented. However, I still find the formulation somehow unfair in introducing spatiotemporal information. The comparison with ARIMA is 1D, but the LSTM has been fed with Spatial information (2D) so would question the idea of fair input information in both models. Now from the perspective of time series, the work is valuable and shows clear contribution on the trade-offs in complexity and usefulness of the LSTM and ARIM for filling gap (in special cases). It is to highlight that the concept here is as a technical note, and the case study is not so common and the very complex in nature. This added to the large availability of data, makes it an important problem to solve with LSTM, and this is an important aid to the area of Deep Learning and appllication cases of LSTM. I think is worth to share in this journal.

Aside from the above, maybe I would like to comment on small issues that I hope can be commented or updated if the paper is published.

**Response**:
In Section 4.3, we did train a multi-well ARIMA model and a multi-well LSTM model. Both types of multi-well models used data from wells 1-1, 1-10A, and 1-16A to estimate the SpC for well 1-1. We compared the multi-well models to the their single-well counterparts in Figure 9, along with a statistical analysis in Table 3 that has been added in the revision.

**Minor comments are in the abstract**

**Reviewer Comment 2.1** — Even the process followed an Hyperparameter optimization, it is important to describe the ranges and sequence of such pipeline of optimal steps.

**Response**: We used a grid-search approach to find the optimal LSTM configuration for a given gap-length at each well. This involved iterating over all combinations of input time window size (M), the number of units (U) in the LSTM layer, and the learning rate (L) listed in Table 1 for each well. We have updated section 3.1.1 to specifically state this.

Old statement: "We performed a hyperparameter search to explore different LSTM model configu-rations, including the input time window size $M$, the number of units $(U)$ in the LSTM layer, and the learning rate $(L)$ at each well."

New statement: "We used a grid-search approach to explore different LSTM model hyperparameter configurations to find the best model for a given gap length at each well. This involved iterating over all combinations of input time window size $(M)$, the number of units $(U)$ in the LSTM layer, and the learning rate $(L)$ listed in Table 1 for each well"

**Reviewer Comment 2.2** — The overall LSTM input output structure for all optimals might be better understood in a table, where the performance and AIC are shown.

**Response**:
We have added a table in the online supplemental material (Table S1) showing the optimal LSTM configuration for a given gap length at each well, along the the models MAPE score for the gap length,

the AIC for the model on the validation set, and range of AIC scores for all models for a given gap length and well on the validation set. The table is shown below (Table 2.1).

We have also added a similar table in the online supplemental material for the optimal multi-well LSTM configurations (Table S3). The table is also shown below (Table 2.2),

Table 2.1: The best LSTM configurations and performance for a given gap length at each well based on the validation data set (2011): the input time window size ($M$), the number of units ($U$) in the LSTM layer, the learning rate ($L$), the SpC MAPE score, the Akaike Information Criterion (AI) for the model on the validation set, and range of AIC scores for all models for a given gap length and well on the validation set.

| Well | Gap Length | M | U | L | MAPE | AIC | AIC Min | AIC Max |
|---|---|---|---|---|---|---|---|---|
| 1-1 | 1 | 96 | 128 | 1e-3 | 0.189 | $1.31 \times 10^4$ | $-1.13 \times 10^5$ | $2.63 \times 10^4$ |
| | 6 | 96 | 128 | 1e-3 | 0.701 | $3.50 \times 10^4$ | $-9.08 \times 10^4$ | $4.19 \times 10^4$ |
| | 12 | 72 | 32 | 1e-4 | 1.24 | $-8.11 \times 10^4$ | $-8.11 \times 10^4$ | $5.40 \times 10^4$ |
| | 24 | 72 | 32 | 1e-4 | 1.95 | $-7.33 \times 10^4$ | $-7.33 \times 10^4$ | $6.50 \times 10^4$ |
| | 48 | 144 | 32 | 1e-5 | 2.85 | $-6.77 \times 10^4$ | $-6.80 \times 10^4$ | $7.50 \times 10^4$ |
| | 72 | 24 | 64 | 1e-5 | 3.67 | $-3.89 \times 10^4$ | $-6.42 \times 10^4$ | $8.08 \times 10^4$ |
| 1-10A | 1 | 120 | 128 | 1e-3 | 0.19 | $4.75 \times 10^4$ | $-7.82 \times 10^4$ | $5.58 \times 10^4$ |
| | 6 | 120 | 128 | 1e-3 | 0.685 | $6.24 \times 10^4$ | $-6.29 \times 10^4$ | $7.39 \times 10^4$ |
| | 12 | 120 | 128 | 1e-3 | 1.1 | $6.69 \times 10^4$ | $-5.82 \times 10^4$ | $8.07 \times 10^4$ |
| | 24 | 144 | 128 | 1e-3 | 1.63 | $7.23 \times 10^4$ | $-5.22 \times 10^4$ | $8.70 \times 10^4$ |
| | 48 | 168 | 64 | 1e-5 | 2.16 | $-2.35 \times 10^4$ | $-4.90 \times 10^4$ | $9.14 \times 10^4$ |
| | 72 | 168 | 64 | 1e-5 | 2.39 | $-2.22 \times 10^4$ | $-4.63 \times 10^4$ | $9.37 \times 10^4$ |
| 1-15 | 1 | 48 | 32 | 1e-3 | 0.0163 | $-1.31 \times 10^5$ | $-1.32 \times 10^5$ | $1.93 \times 10^4$ |
| | 6 | 48 | 32 | 1e-3 | 0.0521 | $-1.04 \times 10^5$ | $-1.10 \times 10^5$ | $2.68 \times 10^4$ |
| | 12 | 48 | 32 | 1e-3 | 0.109 | $-1.02 \times 10^5$ | $-1.02 \times 10^5$ | $4.18 \times 10^4$ |
| | 24 | 96 | 128 | 1e-4 | 0.229 | $3.26 \times 10^4$ | $-9.25 \times 10^4$ | $5.70 \times 10^4$ |
| | 48 | 144 | 64 | 1e-3 | 0.372 | $-5.27 \times 10^4$ | $-8.50 \times 10^4$ | $7.15 \times 10^4$ |
| | 72 | 144 | 64 | 1e-3 | 0.506 | $-4.98 \times 10^4$ | $-8.08 \times 10^4$ | $8.00 \times 10^4$ |
| 2-2 | 1 | 168 | 32 | 1e-3 | 0.49 | $-6.06 \times 10^4$ | $-7.54 \times 10^4$ | $6.52 \times 10^4$ |
| | 6 | 168 | 32 | 1e-3 | 1.62 | $-4.86 \times 10^4$ | $-5.66 \times 10^4$ | $7.75 \times 10^4$ |
| | 12 | 144 | 128 | 1e-5 | 3.07 | $8.00 \times 10^4$ | $-4.95 \times 10^4$ | $8.47 \times 10^4$ |
| | 24 | 144 | 128 | 1e-5 | 4.42 | $8.34 \times 10^4$ | $-4.54 \times 10^4$ | $9.15 \times 10^4$ |
| | 48 | 144 | 128 | 1e-5 | 6.61 | $8.69 \times 10^4$ | $-4.19 \times 10^4$ | $9.63 \times 10^4$ |
| | 72 | 144 | 128 | 1e-5 | 7.52 | $8.66 \times 10^4$ | $-4.01 \times 10^4$ | $9.85 \times 10^4$ |
| 2-3 | 1 | 168 | 64 | 1e-3 | 0.142 | $-8.62 \times 10^4$ | $-1.16 \times 10^5$ | $2.28 \times 10^4$ |
| | 6 | 168 | 64 | 1e-3 | 0.369 | $-6.95 \times 10^4$ | $-9.87 \times 10^4$ | $4.64 \times 10^4$ |
| | 12 | 168 | 64 | 1e-3 | 0.663 | $-5.97 \times 10^4$ | $-8.78 \times 10^4$ | $6.02 \times 10^4$ |
| | 24 | 24 | 64 | 1e-5 | 1.09 | $-5.59 \times 10^4$ | $-7.92 \times 10^4$ | $7.59 \times 10^4$ |
| | 48 | 48 | 64 | 1e-5 | 1.7 | $-4.82 \times 10^4$ | $-7.18 \times 10^4$ | $8.95 \times 10^4$ |
| | 72 | 48 | 64 | 1e-5 | 2.28 | $-4.35 \times 10^4$ | $-6.72 \times 10^4$ | $9.51 \times 10^4$ |
| 2-5 | 1 | 120 | 32 | 1e-3 | 0.109 | $-1.15 \times 10^5$ | $-1.18 \times 10^5$ | $2.51 \times 10^4$ |
| | 6 | 144 | 64 | 1e-4 | 0.332 | $-6.72 \times 10^4$ | $-9.56 \times 10^4$ | $5.49 \times 10^4$ |
| | 12 | 144 | 64 | 1e-4 | 0.586 | $-5.70 \times 10^4$ | $-8.46 \times 10^4$ | $6.89 \times 10^4$ |
| | 24 | 144 | 64 | 1e-4 | 0.999 | $-4.90 \times 10^4$ | $-7.66 \times 10^4$ | $8.08 \times 10^4$ |
| | 48 | 120 | 64 | 1e-5 | 1.39 | $-4.51 \times 10^4$ | $-7.12 \times 10^4$ | $8.92 \times 10^4$ |
| | 72 | 120 | 64 | 1e-5 | 1.77 | $-4.16 \times 10^4$ | $-6.76 \times 10^4$ | $9.19 \times 10^4$ |

Table 2.2: The best multi-well LSTM configurations and performance for a given gap length based on the validation data set (2011): the input time window size ($M$), the number of units ($U$) in the LSTM layer, the learning rate ($L$), the SpC MAPE score, the Akaike Information Criterion (AI) for the model on the validation set, and range of AIC scores for all models for a given gap length on the validation set.

| Gap Length | M | U | L | MAPE | AIC | AIC Min | AIC Max |
|---|---|---|---|---|---|---|---|
| 1 | 144 | 128 | 1e-3 | 0.154 | $4.56 \times 10^4$ | $-8.40 \times 10^4$ | $6.11 \times 10^4$ |
| 6 | 144 | 128 | 1e-3 | 0.63 | $6.27 \times 10^4$ | $-6.68 \times 10^4$ | $7.13 \times 10^4$ |
| 12 | 24 | 32 | 1e-4 | 1.14 | $-5.97 \times 10^4$ | $-5.97 \times 10^4$ | $8.08 \times 10^4$ |
| 24 | 24 | 32 | 1e-4 | 1.75 | $-5.43 \times 10^4$ | $-5.43 \times 10^4$ | $8.89 \times 10^4$ |
| 48 | 96 | 128 | 1e-5 | 2.69 | $7.90 \times 10^4$ | $-4.84 \times 10^4$ | $9.46 \times 10^4$ |
| 72 | 96 | 128 | 1e-5 | 3.15 | $8.09 \times 10^4$ | $-4.74 \times 10^4$ | $9.74 \times 10^4$ |

**Reviewer Comment 1.3** — Looking at Figure 6: If you perform hyper parameter tuning and one model (almost) constantly picks the largest (or smallest) value, usually you should increase the search range, as this indicates that eventually even larger (or smaller) parameters would be better. Looking at e.g. the input window length of the ARIMA model.

**Response**:

We have expanded the search range of the ARIMA model input window from 192 to 504 hours in an increment of 24 hours for all wells to identify optimal input windows to make sure the optimal input window is not at the upper or lower bound of the search range, as shown in the updated Figure 6. In general, larger optimal windows are resulted, which has consequently led to improved performance. We also noticed that the performance improvement for input windows larger than 288 hours is marginal in terms of the MAPE for all gap lengths across the wells we tested.

**Reviewer Comment 1.4** — To my understanding, part of this technical note is the benchmarking/comparison of two models, LSTMs and ARIMA. As such, I think this paper is still missing a statistical analysis of the modeling results. The entire discussion is currently focussed around a few plots and a textual description of what one can see in these figures. However, to a certain degree I would argue that this analysis/interpretation is rather subjective. I think it would benefit the paper to have a table that compares both models on a range of different metrics, including statistical tests of e.g. the robustness/significance of the results. Right now, I wonder what the takeaway message of this paper is. I would argue that it was probably known that both models, LSTMs and ARIMA, are generally capable of time series forecasting. If I would be a user with similar data or a similar problem, what is the additional knowledge that I can gain from reading this paper?

**Response**:

We have added two table comparing the LSTM and ARIMA models on several statistics: one comparing the performance of the LSTM and ARIMA models filling in gap lengths of 24 hours (same models in Figure 8) that is added to the main paper in section 4.1 (Table 1.1 in this response, Table 2 in the revision), and another comparing the two approaches for all gap lengths that is added to the supplemental material (Table 1.2 in this response, Table S2 in the supplemental material). In addition, we have added a table of statistics for the models in Figure 9 to section 4.3, comparing the single and multi-well ARIMA and LSTM models (Table 1.3 in this response, Table 3 in the revision)

We have also added the following statistical analysis of the two models in Section 4.1:

"In addition to the relative errors, we calculated the MAPE, Root Mean Squared Error (RMSE), Nash–Sutcliffe model efficiency coefficient (NSE) [Nash and Sutcliffe, 1970], and Kling-Gupta Efficiency (KGE) [Gupta et al., 2009] for the best LSTM and ARIMA model per gap length to compare the two approaches. Table 1.1 compares the performance of the LSTM and ARIMA models filling in gap lengths of 24 hours. The table for all gap lengths is in the online supplemental material (Table S2).

NSE is a metric used to assess the predictive skills and accuracy of hydrological models. Values range from $-\infty$ to 1, where 1 indicates a perfect model fit, 0 indicates that the model has the same predictive power as the mean of the observations, and less than 0 indicates that the model is a worse predictor than the mean of the observations. NSE is calculated on the SpC predictions by the following equation:

$$NSE = 1 - \frac{\sum_{t=1}^{n} (\mathrm{P_t} - \mathrm{O_t})^2}{\sum_{t=1}^{n} (\mathrm{O_t} - \mu(\mathrm{O}))^2}, \tag{1}$$

where $n$ is the total number of synthetic missing data points during the evaluation period, $\mathrm{P_t}$ and $\mathrm{O_t}$ are the predicted and observed SpC values at time $t$, and $\mu(\mathrm{O})$ is the mean observed SpC value.

KGE is another goodness-of-fit metric used to evaluate hydrological models by combining the three components of NSE of model errors (i.e. correlation, bias, ratio of variances or coefficients of variation) in a more balanced way. It has the same range of values as NSE, where 1 indicates a perfect model fit. KGE is calculated on a model's SpC predictions by the following equations:

$$KGE = 1 - \sqrt{(r-1)^2 + (\alpha - 1)^2 + (\beta - 1)^2} \tag{2}$$

$$r = \frac{\mathrm{cov}(\mathrm{O}, \mathrm{P})}{\sigma(\mathrm{O}) * \sigma(\mathrm{P})} \tag{3}$$

$$\alpha = \frac{\sigma(\mathrm{P})}{\sigma(\mathrm{O})} \tag{4}$$

$$\beta = \frac{\mu(\mathrm{P})}{\mu(\mathrm{O})}, \tag{5}$$

where $\mathrm{cov}$ is the covariance, $\sigma$ is the standard deviation, and $\mu$ is the arithmetic mean.

The LSTM and ARIMA models yielded comparable average metrics at all the wells for the gap length of 24 hours, as can be seen in Table 1.1. The NSE and KGE resulted from both models are close to 1 for all the wells with negligible differences between the two models. The difference in MAPE and RMSE is also small, with relatively more notable differences for wells 2-2 and 2-3, where the ARIMA models resulted in lower MAPE and RMSE."

In addition, we updated the analysis of the multi-well models in section 4.3 of the revision as follows:

"The boxplots of relative errors yielded from the single-well and multi-well models using both approaches are provided in Figure 9 for comparison. Additionally, we include performance metrics comparing the single and multi-well models in Table 3. Additional spatial information seems to exacerbate the relative errors by the ARIMA models, except for large gaps (e.g., 72 hours). The LSTM approach, on the contrary, benefits from the information carried by the neighboring wells to fill in those larger gaps, while the performance for small gaps stay unchanged. The aggregated performance metrics in Table 3 show slightly improved metrics for multi-well ARIMA models for gaps smaller than 24 hours compared to the single-well models, while the turning point in relative performance is at 12 hours for the LSTM models. The deteriorated performance metrics of the multi-well LSTM models at the larger gap lengths are consistent with their larger inter-quartile ranges as revealed by the boxplots of relative errors in Figure 9. However, the multi-well LSTM and ARIMA models can reduce the occurrence of large relative errors for larger gaps, providing more robust gap-filling under those circumstances."

**Minor comments are in the abstract**

**Reviewer Comment 1.5** — Figure 4: I think this figure is misleading to someone unfamiliar with LSTMs. You actually drew a fully connected network, rather than a recurrent (sequential) neural network. As of now, it seems like all inputs go into the LSTM at once (no time steps are visible in this figure), and the outputs of all time steps (since on the left side the timesteps are top

Table 1.1: Comparison of single-well LSTM and ARIMA models on 24-hour synthetic gap in the SpC data on the test set for each well. The models are the same ones used in Figure 8. The calculated statistics are: MAPE, Root Mean Squared Error (RMSE), Nash–Sutcliffe model efficiency coefficient (NSE), and Kling-Gupta Efficiency (KGE). The T-Score and P-Value are calculated on the relative errors of the two models per well.

| Well | Model Type | MAPE | RMSE | NSE | KGE | T-Score | P-Value |
|------|-----------|------|------|-----|-----|---------|---------|
| 1-1 | LSTM | 1.38 | $8.33 \times 10^{-3}$ | 0.991 | 0.988 | 19.1 | $1.00 \times 10^{-80}$ |
| | ARIMA | 1.36 | $8.98 \times 10^{-3}$ | 0.989 | 0.994 | | |
| 1-10A | LSTM | 1.37 | $8.07 \times 10^{-3}$ | 0.986 | 0.968 | $-24.6$ | $1.48 \times 10^{-131}$ |
| | ARIMA | 1.5 | $9.60 \times 10^{-3}$ | 0.98 | 0.987 | | |
| 1-15 | LSTM | 0.259 | $1.88 \times 10^{-3}$ | 0.989 | 0.982 | $-48.9$ | 0.00 |
| | ARIMA | 0.119 | $1.18 \times 10^{-3}$ | 0.996 | 0.997 | | |
| 2-2 | LSTM | 2.97 | $1.87 \times 10^{-2}$ | 0.922 | 0.962 | 48.1 | 0.00 |
| | ARIMA | 2.23 | $1.64 \times 10^{-2}$ | 0.939 | 0.967 | | |
| 2-3 | LSTM | 2.15 | $1.63 \times 10^{-2}$ | 0.945 | 0.965 | 21.6 | $4.69 \times 10^{-102}$ |
| | ARIMA | 1.72 | $1.48 \times 10^{-2}$ | 0.954 | 0.971 | | |
| 2-5 | LSTM | 0.929 | $6.86 \times 10^{-3}$ | 0.976 | 0.988 | $-9.6$ | $9.22 \times 10^{-22}$ |
| | ARIMA | 0.866 | $7.45 \times 10^{-3}$ | 0.971 | 0.977 | | |

to bottom) are used to predict the output. Figure (b) is actually the more correct depiction of the LSTM and I don't understand why both visualizations are needed.

**Response**: We have modified Figure 4 to make it clear that the middle layer (yellow and green) are individual LSTM units and not a fully connected network. The point of Figure 4(a) is to show the overall structure of the model and Figure 4(b) shows how the input data is fed into an individual LSTM unit.

**Reviewer Comment 1.6** — P 3, L 32 "at the 300 Area of the U.S. Department of Energy Hanford site". What is the 300 for?

**Response**:
Hanford's 300 Area is a U.S. Department of Energy (DOE) site where the fuel manufacturing operations occur. It is a numbered naming convention for the site. Details can be seen https://www.hanford.gov/page.cfm/300

**Reviewer Comment 1.7** — P 8, L 25: "The well data were then pre-processed by normalizing all measurements via zero-mean and unit variance for each variable". You do not normalize "via" zero-mean and unit variance, you rather normalize to zero mean and unit variance. Normalizing via zero-mean and unit-variance sounds like you subtract a mean of zero and divide by a variance of one, which is hopefully not what you have done.

**Response**: We have updated the line to say "normalizing all measurements to zero mean and unit variance" to clarify.

**Reviewer Comment 1.8** — P9 L 7ff You use plural for "multi-well models" throughout this passage but you only trained one multi-well model to predict at well 1-1, or?

**Response**:  Yes, only one type of multi-well LSTM model is trained to predict at well 1-1 due to data availability. It is clarified in the text. "In addition to the LSTM models trained for the single-well setup, we also trained multi-well LSTM models that used observations from wells 1-1, 1-10A, and 1-16A to fill in data gaps for well 1-1"

**Reviewer Comment 1.9** — P17 L 17: Which "auto.arima" function?

**Response**:  The function we used in our study applies the Hyndman-Khandakar algorithm developed by Hyndman and Khandakar [2008] that minimizes the Akaike Information Criterion (AIC) to obtain an optimized ARIMA model. The following details have been added to section 3.2:

[revised manuscript text omitted]

---

## Author Response (AR4)

**Editor**

Editor Comment 1.1 — Please address all comments of referees 1 and 3. Referee 3 makes comments which has been made earlier also: 1) the need to show what is novel (what is contribution to science) in this paper using LSTM in hydrological predictions if compared to other papers of this kind; 2) in this gap filling problem you are not using the "future" data (which are known) - please explain why or update your method to use it.

**Response**: Both reviewers' comments have been carefully addressed. We summarized the novel contribution in our conclusion section and highlighted in our abstract. The key take-away messages are: The LSTM method is able to account for both low-frequency (seasonal) and high-frequency (weekly, daily and subdaily) dynamics in data, while the traditional ARIMA type of methods focus on capturing the seasonal dynamics in data. It is also important to note that ARIMA methods fail to capture abrupt changes that are present in highly dynamic data. Wavelet analysis can be performed to understand whether high-frequency dynamics exist, which can then guide whether LSTM or other deep neural nets are needed. We also demonstrated that LSTM makes it easier to include both spatial and temporal correlations in spatially distributed data and to regress on other co-located data. LSTM may require more training data, which is another important factor when making decision. Given the importance of irregularly distributed monitoring data to understand environmental systems and to inform decision making, our work is showcasing the power of emerging deep learning methods in filling data gaps, thus improves the value of costly monitoring data.

We ran more than 400 individual models using both LSTM and ARIMA approaches to answer the following questions: (1) how is gap filling performance impacted by the length of gaps? (2) how does the amount of training data impact the model performance? (3) how important is the choice of the input time window? (4) how does the dynamics signature of the data impact the performance of both models? (4) How much value can measurements at neighboring add to the performance improvement? The answers to these questions are highly valuable in practical applications. Our discussion paper has been cited 16 times on google scholar in the past 2 years, which also reflects how much the research community values our work.

The models developed using ARIMA and LSTM in our paper were predictive models which did not explicitly include the future information. However, the future information has been implicitly used in training the spatial-temporal correlations in LSTM models. As we discussed in our response to the comments of Reviewer 3, this is a common practice in gap filling research. We added relevant statements in the introduction to further clarify. We also discuss in our conclusion section that bidirectional LSTM can be a natural choice for incorporating future information. However, developing, training and evaluating Bi-LSTM is not a trivial task, and it will overload the current paper. Thus, we would be happy to pursue this as a future publication.

**Reviewer 1**

**Reviewer Comment 1.1** — The paper has improved significantly, yet in my opinion still the text does fail to explain data and equations were it should and some paragraphs leave uncertainty. However, the equations and answers of details appear in another section which makes it very difficult to follow. I think as it is, I suggest a las English review to see that the explanation of basic concepts are mentioned where they should. For the rest, that paper is a valuable contribution and has all information required for publication.

**Response**: Thanks for the positive assessment on the work we have done to improve the manuscript in the previous revisions. To further address the reviewer's comment related to clear explanation, we worked with a professional editor for a thorough edit on the English. The equations/metrics have been moved to the section 3.1 with the descriptions as suggested by the reviewer. The track changes version has been submitted as well.

**Reviewer 3**

**Reviewer Comment 3.1** — This paper presents a method to fill data gaps in hydrological monitoring networks based on LSTM. It is of significance in hydrological applications. However, the strategy used for gap filling is the same as that for hydrological predictions. In other words, the method predicts "future" based on previous observations. In my opinion, data gap filling is different from prediction in that the observations occurring both before and after gaps can be used in gap filling. This study only uses the observations occurring before gaps, which is in essence prediction.

**Response**: The models developed using ARIMA and LSTM in our paper were predictive models which did not explicitly include the future information. However, the future information has been implicitly used in training the spatial-temporal correlations in LSTM models. Framing gap filling as predictive problems is a common practice when machine learning methods are used for filling gaps in time series data (see examples in Kandasamy et al. [2013], Körner et al. [2018], Chen et al. [2020], Zhao et al. [2020], Sarafanov et al. [2020], Contractor and Roughan [2021]). We added relevant statements in the introduction to further clarify. We also discuss in our conclusion section that bidirectional LSTM can be a natural choice for incorporating future information. However, developing, training and evaluating Bi-LSTM is not a trivial task, and it will overload the current paper. Thus, we would be happy to pursue this as a future publication.

**Reviewer Comment 3.2** — The evaluation metrics of the prediction models seem very good. The performance of time series prediction models largely depends on the amplitude of data variation. It is recommended to present the SpC variations with different time intervals (1, 6, 12, 24, 48, and 72hours). Also, the values of RMSE are affected by the magnitude of data. It is helpful to present the mean value of SpC observations.

**Response**: Thank you for the suggestions. Mean and variance of the SpC from model testing period in different time intervals have been presented in two versions of boxplots: with and without outliers. The boxplot without outlier aims to reveal the distribution of majority data points (99.3% of all the data has been used in the boxplot). Figure3.1 shows the mean and variance of SpC in 24-hr duration for modeled wells. We notice that for well 1-15, both mean and variance are relative small where the predictive models have a better performance. The variance of well 2-3 contains large mount of extreme values where both tested approaches have a worse performance. The boxplots for other time intervals are also attached at the end of this response letter.

We did also include other metrics like NSE and KGE, which take into account of the magnitude of variability in data, to better compare across different data variability.

**Reviewer Comment 3.3** — There are not outliers in the boxplot of Figure S4.

**Response:**

Figure S4 is intended to exclude outliers (approximately 0.7% of data points) because such because these extreme values significantly zoom out the plot scale and make it almost impossible to see the range in the majority of data. This is explained in the figure caption.

Figure 3.1: Boxplot of mean and variance of SpC at model testing period in 24-hour duration.

**References**

- Siyong Chen, Xiaoyan Wang, Hui Guo, Peiyao Xie, and Abuobaida M Sirelkhatim. Spatial and temporal adaptive gap-filling method producing daily cloud-free ndsi time series. IEEE Journal of Selected Topics in Applied Earth Observations and Remote Sensing, 13:2251–2263, 2020.
- Steefan Contractor and Moninya Roughan. Efficacy of feedforward and lstm neural networks at predicting and gap filling coastal ocean timeseries: Oxygen, nutrients, and temperature. Frontiers in Marine Science, 2021.
- Sivasathivel Kandasamy, Frederic Baret, Aleixandre Verger, Philippe Neveux, and Marie Weiss. A comparison of methods for smoothing and gap filling time series of remote sensing observations–application to modis lai products. Biogeosciences, 10(6):4055–4071, 2013.
- Philipp Körner, Rico Kronenberg, Sandra Genzel, and Christian Bernhofer. Introducing gradient boosting as a universal gap filling tool for meteorological time series. Meteorologische Zeitschrift, 27(5):369–376, 2018.
- Mikhail Sarafanov, Eduard Kazakov, Nikolay O Nikitin, and Anna V Kalyuzhnaya. A machine learning approach for remote sensing data gap-filling with open-source implementation: An ex-